

# Skill of a global forecasting system in seasonal ensemble streamflow prediction

Naze Candogan Yossef [1,2], Rens van Beek[1], Albrecht Weerts[2,3], Hessel Winsemius[2], Marc F. P. Bierkens[1]

[1]Faculty of Geosciences, Utrecht University, Utrecht, 3508 TC, The Netherlands
[2]Deltares, Delft, 2600 MH, The Netherlands
[3]Department of Environmental Sciences, Wageningen University, 6708 PB, The Netherlands

*Correspondence to*: Naze Candogan Yossef (ncandogan@hotmail.com)

**Abstract.** In this study we assess the skill of seasonal streamflow forecasts with the global hydrological
forecasting system FEWS-World which has been set up within the European Commission 7th Framework Programme Project Global Water Scarcity Information Service (GLOWASIS). FEWS-World incorporates the global hydrological model PCR-GLOBWB. We produce ensemble forecasts of monthly discharges for 20 large rivers of the world, with lead times of up to 6 months, forcing the system with bias-corrected seasonal meteorological forecast ensembles from the ECMWF and with probabilistic
meteorological ensembles obtained following the ESP procedure. Here, the skill from the ESP ensembles, which contain no actual information on weather, serves as a benchmark to assess the additional skill that may be obtained using ECMWF seasonal forecasts. We use the Brier Score to quantify the skill of the system in forecasting high and low flows, defined as discharges higher than the 75$^{th}$ and lower than the 25$^{th}$ percentiles for a given month respectively. We determine the theoretical skill
by comparing the results against model simulations and the actual skill in comparison to discharge observations. We calculate the ratios of actual to theoretical skill in order to quantify the percentage of the theoretical skill that is achieved. The results suggest that the skill of ECMWF S3 forecasts is close to that of the ESP forecasts. While better meteorological forecasts could potentially lead to an improvement in hydrological forecasts, this cannot be achieved yet using the ECMWF S3 dataset.

## 1. Introduction

Reliable seasonal streamflow forecasts potentially have many benefits including disaster relief, management of hydropower reservoirs, water supply, agriculture and navigation. Seasonal hydrological forecasting on a global scale could be especially valuable for developing regions, where effective hydrological forecasting systems are scarce. Furthermore, global seasonal forecasts provide spatially
consistent predictions of streamflow anomalies. These may supply information to disaster management organizations operating at global scale to prepare for response as well as to the international water and energy markets about the regional availability of water and hydropower in the coming months.

Approaches to seasonal streamflow forecasting can be divided into two categories, empirical/statistical methods and numerical/dynamical methods. Empirical/statistical methods use statistical techniques (e.g.,
simple correlation, multiple regression, linear or quadratic discriminant analysis, canonical correlation





analysis, and neural networks etc.) to find statistically significant relationships between atmospheric/oceanic indicators and river flow on the basis of historical observations. While statistical forecasts are quite successful in some regions of the world and in some seasons, in many cases the available records are too short to accurately capture climatic variability. Moreover, forecasts derived

5 from past climate do not include anthropogenic or other long-term changes in the climate, such as global warming; and statistical methods do not explain the underlying physical mechanisms. Although statistical methods are the more widely developed and reliable methods that are used for most current operational seasonal forecasts, dynamical modelling is thought to hold the greatest potential for future improvement in reliable seasonal streamflow forecasting (Zwiers and von Storch, 2004).

Dynamical model experiments involve the integration of General Circulation Models (GCMs) which model atmospheric, oceanic and land surface interactions and processes as a set of dynamic equations. In contrast to medium-range Numerical Weather Prediction (NWP), which is based on atmospheric-only integrations, seasonal forecasting by GCMs is based on coupled ocean-atmospheric integrations, where

both atmospheric and oceanic components of the Earth's system are taken into account. The main source of predictability for climate forecasting at seasonal scale is the long-term predictability of the oceanic circulation and its large impact on the global atmospheric circulation. The most important cause of seasonal climate variability is the ENSO (El Niño Southern Oscillation) cycle, which is the large-scale fluctuation of ocean temperatures, rainfall, atmospheric circulation, vertical motion and air pressure

centered over the tropical Pacific but affecting other ocean basins as well. Similarly, unusually warm or cold sea surface temperatures (SST) in other tropical oceans, the extent and thickness of snow cover and the amount of soil moisture can have a persistent influence on the atmospheric circulation (Persson and Grazzini, 2007). Due to the chaotic nature of the atmospheric-oceanic system, model runs made with small, random perturbations in the input data may produce a wide range of difference in the output.

Therefore, GCMs are run multiple times with slightly different sets of initial conditions, producing a set of output data called an ensemble. The hydrological output from the land surface scheme of a GCM may be used as streamflow forecasts. Alternatively, the meteorological forecast ensemble by a GCM may be used as input to a hydrological model which produces streamflow forecast ensembles, as we do in this research.

This paper investigates the skill of seasonal streamflow forecasts for 20 of the largest rivers in the world with the global hydrological forecasting system FEWS-World, which has been setup within the European Commission 7th Framework Programme Project Global Water Scarcity Information Service (GLOWASIS). FEWS-World incorporates the global hydrological model PCR-GLOBWB. The

35 capability of global hydrological models to predict streamflow was demonstrated previously by several studies such as the WaterGap (Alcamo et al., 2003; Döll et al., 2003), LaD (Milly and Schmakin, 2002), VIC (Nijssen et al., 2001), WBM (Vörösmarty et al., 2000; Fekete et al., 2002), Macro-PDM (Arnell, 1999; 2004), and PCR-GLOBWB (Sperna-Weiland et al., 2010; van Beek et al., 2011). Candogan Yossef et al. (2012) assessed the skill of the global hydrological model PCR-GLOBWB in reproducing past

discharge extremes for 20 large rivers of the world, as a first step towards developing a global seasonal





hydrological forecasting system and assessing its skill. The study quantified skill in deterministic hindcast mode, using the ERA 40 reanalysis by the European Centre for Medium-range Weather Forecasts (ECMWF). This preliminary assessment by Candogan Yossef et al. (2012) concluded that the prospects for seasonal forecasting with PCR-GLOBWB or comparable models are positive. Since actual

probabilistic meteorological forecast ensembles were not used, the assessment did not include errors in the meteorological forcing.

However, in an actual forecasting setup, the predictive skill of a hydrological forecasting system is affected not only by errors in model structure and parameterization and initial conditions such as soil

moisture, groundwater and snow, but also by meteorological forcing errors. Skill of seasonal hydrological forecasts can thus be improved by better meteorological forecasts on the one hand and by better estimation of initial hydrologic states through assimilation of independent hydrological observations on the other hand. The improvement in the overall predictability that may be attained depends on the relative importance of these two sources of uncertainty, which varies considerably among

hydrological systems according to location, season and lead time (Bierkens and van den Hurk, 2007; Bierkens and van Beek, 2009; Shukla and Lettenmaier, 2011; Shukla et al., 2011). Candogan Yossef et al. (2013) assessed the roles of initial conditions (IC) and meteorological forcing (MF) in the skill of the global seasonal streamflow forecasting system FEWS-World, based on the ESP/revESP procedure outlined by Wood and Lettenmaier (2008). This study showed the potential for improvement in the skill

of streamflow forecasts by a better estimation of IC or a more accurate MF input per region and per time of the year. The current paper aims to assess the total skill of hydrological forecasts, as affected by errors in model structure, in the estimation of IC as well as in the actual meteorological forecasts that are used to force the model.

The remaining part of this paper is set up as follows. Section 2 describes the global seasonal hydrological forecasting system, FEWS-World, the global hydrological model PCR-GLOBWB and the meteorological forcing data. Section 3 describes the hydrological simulations, the skill measures and the uncertainty analysis. Results are presented in Section 4, followed by discussion in Section 5 and conclusions in the last section.

**2. Materials and Methods**

**2.1. Global hydrological forecasting system FEWS-World**

FEWS-World is a global hydrological forecasting system configured within the forecasting environment Delft-FEWS. Delft-FEWS is an open shell for data handling, managing and guiding forecasting processes (Werner et al., 2013). It is used by a large number of operational forecasting centres and agencies around

35 the world for various purposes such as forecasting hydrological storm surges, river flows, reservoir management and water quality. FEWS-World has been built as part of the GLOWASIS project. The FEWS-World system consists of a Master Controller, a Postgres database and 18 forecasting shells (i.e. computational cores) for efficient handling of ensemble forecasts and data processing. Within FEWS-





World several workflows have been setup for running the global hydrological model PCR-GLOBWB using the precipitation, temperature and potential evaporation fields from the ERA Interim/Land GPCP-corrected dataset (Balsamo et al., 2011). Further descriptions of the meteorological forcing datasets are given in Section 2.2.

PCR-GLOBWB (PCRaster Global Water Balance) simulates the terrestrial part of the global water cycle (van Beek et al., 2011; van Beek and Bierkens, 2009). It is coded in the high-level computer language PCRaster for constructing environmental models (Wesseling et al., 1996). The model is fully distributed and operates on a regular grid with a cell size of 0.5 x 0.5° on a daily time step. Meteorological forcing

is assumed to be constant over the grid cell. Sub-grid variability of hydrological processes is taken into account in the representation of short and tall vegetation, open water, different soil types, saturated area, surface runoff, interflow and groundwater discharge.

PCR-GLOBWB calculates the water balance for every grid cell by tracking the transfer of water between

the atmosphere and the cell, through stores within each cell, and laterally, as discharge, from one cell to the downstream neighbour. The model calculates the storages and fluxes of water, simulates the generation of runoff and its propagation as discharge through the river network. Precipitation falls either as snow or rain depending on atmospheric temperature. It can be intercepted by vegetation and added to the finite canopy storage, which is subject to open water evaporation. Snow is accumulated when the

temperature is lower than 0°C and melts when it is higher. Snow melt is added to rain and throughfall; it is either stored in the available pore space in the snow cover, or it infiltrates into the top soil layer. Part of this water is transformed into surface runoff and the remainder infiltrates into the soil through two vertically stacked soil layers and an underlying groundwater layer. Water is exchanged between these layers following Darcy's law and the resulting soil moisture is subject to evapotranspiration. The

remaining water contributes to lateral drainage as interflow from the soil layers or baseflow from the groundwater reservoir. The total drainage, consisting of surface runoff, interflow and baseflow is routed through the drainage network of rivers, lakes, wetlands and reservoirs based on DDM30 (Döll and Lehner, 2002), using the kinematic wave approach. An extensive description of PCR-GLOBWB can be found in van Beek and Bierkens (2009).

**2.2. Meteorological forcing data**

The meteorological variables required to force PCR-GLOBWB are daily values of precipitation, evapotranspiration and temperature. In the absence of direct estimates of actual evapotranspiration, the model can be forced with values of reference potential evapotranspiration, calculated from temperature, radiation, cloud cover, vapour pressure and wind speed.

We force PCR-GLOBWB with two different datasets. The first one is the ERA-Interim/Land dataset (Balsamo et al., 2015). This is a global meteorological dataset, which is a combination of the ERA-Interim reanalysis (Dee et al., 2011) and Global Precipitation Climatology Project (GPCP) monthly rainfall observations (Huffman and Bolvin, 2011; Huffman et al., 2009). ERA-Interim is the most robust





global atmospheric reanalysis produced by the ECMWF. It is an 'interim' reanalysis initially started from year 1989; later extended back to the year 1979; and continues to be updated forward in time. ERA-Interim reanalysis was produced as a part of the next-generation extended reanalysis intended to replace ERA-40. The GPCP is part of the Global Energy and Water Cycle Experiment (GEWEX) of the World Climate Research program (WCRP). The GPCP provides global precipitation estimates by merging infrared and microwave satellite estimates with rain gauge data from more than 6000 stations. Monthly values of potential evaporation have been estimated from ERA-Interim, using fields of temperature, radiation, cloud cover, vapour pressure and wind speed, by application of the Penman-Monteith equation (Monteith, 1981; Penman, 1948) for a reference grass canopy, according to the FAO methodology (Allen et al., 1998). Reference potential evaporation is multiplied by a monthly crop factor to obtain land cover specific potential evaporation in PCR-GLOBWB.

The second dataset that we use to force the model is the forecast ensemble from the S3 seasonal forecast archives of the ECMWF covering the period 1981-2010. ECMWF S3 seasonal forecasts are run in ensemble mode on a fully coupled ocean-atmosphere model. They are run on the 1st of every month as the initial date, integrated forward for six months. Verifications show that the skill of forecasts in regions and seasons known to have a teleconnection with the El Niño is much higher than during neutral conditions. ECMWF seasonal forecast system has been shown to be superior to statistical systems in forecasting the onset of El Niño or La Niña. But once an event has started statistical systems have comparable skill. The dynamical model is also better than the statistical models in forecasting the sea surface temperature (SST) in the Atlantic Ocean and the Indian Ocean. In many parts of the tropics, where changes such as those associated with El Niño can have a large impact on global weather patterns, a substantial part of the year-to-year variation in seasonal-mean rainfall and temperature is predictable. In mid-latitudes, the level of predictability is lower, and Europe in particular is a difficult area to predict. Seasonal forecasts start to show signs of systematic model errors after about ten days into the forecast. The ECMWF does not introduce any artificial terms in the equations to reduce the drift. Rather, a daily bias-correction based on quantile-quantile transformation is applied on each forecast. To account for drift, the bias correction varies per forecast month. As a result, there are 12 bias correction datasets each with a length equal to a seasonal forecast. The bias correction dataset was provided by the ECMWF (Dutra, personal communication) within the GLOWASIS project.

### 2.3 Streamflow forecast runs

PCR-GLOBWB is run at a daily time-step to produce two sets of streamflow forecast ensembles, as well as the control simulation run. The first forecast run follows the ESP procedure using the ERA-Interim/Land dataset as basis for the meteorological input. The second forecast run uses actual ECMWF S3 seasonal forecasts as meteorological input. The model spin-up is carried out over the period 1979-1984 using ERA-Interim/Land dataset. Subsequently, the hydrological states at the end of this 5-year spin-up are used as initial states for the control run. The control run is a single simulation covering the 30 years historical period from 1981 to 2010. Daily discharge values are aggregated into monthly totals. Monthly aggregation provides a more appropriate forecast at the seasonal scale and a proxy of the





underlying distribution. Hydrologic states, as well as monthly discharge totals are saved at the end of each month. These states are used as initial conditions (ICs) for running the ESP as well as the ECMWF S3 seasonal forecasts.

The ESP forecast ensemble is produced with the ESP workflow within Delft-FEWS. Input ensembles of the meteorological forcing are created from the 32-year input data series (1979-2010). PCR-GLOBWB model runs are initialized on the 1st day of each month for the period 1981-2010 using the stored ICs. This results in 360 ESP runs, each run containing 32 members.

The ECMWF S3 streamflow forecast ensemble is produced by forcing the model with bias corrected meteorological input dataset from the ECMWF S3 seasonal forecast archive, containing 11 ensemble members for each forecast and covering the period 1981-2010. 12 monthly forecast over the 30 years period results in 30x12=360 runs, with 11 ensemble members for each run. Both the ESP and ECMWF S3 runs are carried out in batch using the FEWS-World forecasting system. Each run spans 6 months and
produces an ensemble of 11 monthly discharge values for 6 lead times.

### 2.4 Skill assessment

Skill assessment of probabilistic ensemble forecasts involves the comparison of forecasted probabilities to observed event frequencies. For the ESP approach and the ECMWF S3 seasonal meteorological forecasts, we quantify the theoretical as well as the actual skill. To calculate the theoretical skill, we
compare the streamflow forecast ensembles to the results of the control simulation; and for the actual skill we compare them to observed discharge records (GRDC). We apply the Brier Score (BS) to all four comparisons. The meteorological datasets used in the calculation of BS are clarified in Table 1.

The BS is commonly used for the verification of meteorological probabilistic forecasts. It is preferred
for being a proper score, i.e., being optimized for forecasts that correspond to the best judgement of the forecaster. It is also a highly compressed score, i.e., it directly accounts for forecast probabilities without necessitating a contingency table for each probability threshold (Bartholmes et al., 2008; Ferro, 2007). In this study we use two probability thresholds corresponding to the 25[th] and 75[th] percentiles for high and low flows respectively. Values below the 25[th] percentile of a given month of the year are considered low
flows and those above the 75[th] percentile are considered high flows. The thresholds are calculated separately for forecasted values and observed values. In other words, we classify a forecasted value as high flow if it exceeds the 75[th] percentile of all forecasted values for the same month of the year and low flow if it is below the 25[th] percentile. Similarly, an observed value is classified as high flow if it exceeds the 75[th] percentile of all observed values for the same month of the year and low flow if it is below the
25[th] percentile. This approach eliminates any systematic bias in the simulations compared to the observations. In this way we are able to assess the skill in forecasting the occurrence of flows that are higher or lower than usual for a given month.

The BS values for a given month and lead time are calculated following Eq. (1):



$$BS = \frac{1}{N}\sum_{t=1}^{N}(p_t - o_t)^2 \tag{1}$$

where,

5    N is the number of forecasting instances,

p is the forecasted probability

o is the observed probability

The range of the BS is (0, 1), 0 being the best value for a perfect forecast and 1 the worst.

We calculate the BS values in 20 large global basins separately for the 12 months of the year and for all 6 lead times. When calculating the BS for a given month and a given lead time, we use the forecast ensembles that predict the total monthly discharge generated during that given month. In other words, we use the discharge ensembles resulting from the simulations which start at time $t_0$ and end at time $t_n$

with a lead time of n months, where $t_0$ is prior to the end of the given forecast month by n months. Thus, for the month of May and for 1 month lead time, $n = 1$, $t_0$ is the 1st of May and $t_n$ is the 31st of May. For 2 months lead time, $n = 2$, $t_0$ is the 1st of April and $t_n$ is again the 31st of May.

We determine the theoretical skill $BS_{theo}$ as well as the actual skill $BS_{act}$ by comparing the results against

model simulations for the reference period, as well as the actual skill $BS_{act}$ against discharge observation records. For $BS_{theo}$ we compare the discharge ensembles resulting from the forecast run (p, the forecasted probability) against the discharge values resulting from the control run (o, the observed probability). For $BS_{act}$ the observed probability values derive from the actual observation records.

We express the percentage of the theoretical skill that is attained as the actual skill following Eq. (2):

$$\%BS_{act} = \frac{BS_{act}*100}{BS_{theo}} \tag{2}$$

In order to quantify the added skill obtained by using ECMWF S3 seasonal meteorological forecasts

compared to the reference ESP forecast, we calculate the Brier Skill Scores (BSS) relative to $BS_{ref}$ following Eq. (3):

$$BSS = 1 - \frac{BS}{BS_{ref}} \tag{3}$$

The range of the BSS is $(-\infty, 1)$ and the best value for a perfect forecast is 1. When the BSS is equal to 0, the forecast skill is equal to that of the reference forecast. Here, a skill of zero or less implies that the seasonal forecasts provide no additional information compared to the random generated climatology of the ESP forecast run.





## 3. Results

### 3.1 Skill scores

We present the results of the skill assessment in 20 score tables for 20 rivers (Tables 2-21). The first 8 parts of each table show the BS and BSS for the four cases of actual and theoretical skill, for low and

high flows, i.e., the $25^{th}$ and the $75^{th}$ percentiles. Tables present the scores for the 12 months of the year and for 6 lead times.

The tables are color-coded for easier visual inspection. Values are highlighted in blue where the skill of the ECMWF S3 forecasts is considerably higher than that of the ESP forecast, and in yellow where it is

considerably lower. Since the best value for BS is 0, higher skill corresponds to a lower BS. Where the difference between the BS values of the ECMWF S3 and ESP forecasts are larger or equal to 0.05, the value is highlighted in light blue or light yellow; where it is larger or equal to 0.1, it is highlighted in dark blue or dark yellow.

### 3.2 Theoretical vs. actual skill

In Tables 2-21, the last two parts of each table show the percentage of theoretical skill of the ESP and ECMWF S3 forecasts which is attained as actual skill. These two parts present the percentages for the 12 months of the year and for 6 lead times, for low and high flows respectively.

### 3.3 Overview of the basins with added skill

We provide a global overview of the basins where added skill is obtained using ECMWF S3

meteorological forecast input compared to the skill obtained using the ESP input. The locations of improved skill are presented on four world maps for the four cases of actual and theoretical skill, for low and high flows, i.e., the $25^{th}$ and the $75^{th}$ percentiles (Fig 1). The maps indicate the number of months per year with skilful forecasts at each location, as well as the maximum lead-time for which the skill is retained.

## 4. Discussion of Results

In this section we discuss the results for several larger basins in the context of prevailing hydroclimatic conditions.

### 4.1 Tropical, monsoon-dominated basins

As can be seen in Fig. 1(a), results indicate that in the Amazon basin the theoretical skill of the ECMWF

S3 forecasts is quite high for predicting lower flows than usual for the given month. In Table 2 for the Amazon, the color-coded first part which presents the theoretical skill for low flow shows that most of the BS values are coloured blue. This indicates that the ECMWF S3 forecasts are significantly more skilful than the ESP forecasts, i.e., the difference between the BS values is higher than 0.05. For lead-times of 1 and 2 months, the skill improvement is larger, as can be seen on the first two columns, which

are coloured mostly dark blue, indicating a difference between BS values higher than 0.1.





The results for high flows are very different than those for low flows, as can be seen in Fig. 1b, as well as the third part of Table 2. Most BS values of the ECMWF S3 are very close to the ESP, with only a few significantly different values, and these indicate lower skill, indicated by the yellow colour.

The results are also different for the actual skill as can be seen in Fig. 1c and Fig. 1d. Both for low and high flows (the fifth and seventh parts of the table), the skill of the ECMWF S3 is either very close to the ESP or lower, as can be seen again by the yellow colour. The average actual skill attained by the ECMWF S3 forecasts over the year and the 6 lead times is 50% of the theoretical skill in forecasting low flows

and 57% in high flows (the last two parts of Table 2). These percentages increase with increasing lead time, starting from 21% for low flows at a lead time of 1 month, and rising to 68% at a lead time of 6 months. There are considerable differences in the actually attained percentage of theoretical skill between months as well.

Candogan Yossef et.al (2012) showed that hydrological forecasting skill in the Amazon basin is dominated by initial conditions for lead times of 1-2 months, and even up to 4 months for forecasting the discharge during the Southern hemisphere spring, from August until November. Initial conditions are especially important during high flow conditions (March, April and May) (Paiva et. al, 2012) and the recession period (June, July, August), when the increased groundwater storage plays an important role.

Moreover, in large basins such as the Amazon where long travel times are involved, the knowledge of surface water conditions several months ahead is an important source of forecast skill. Meteorological forcing starts to play a more important role beyond 1-2 months lead times throughout the rest of the year. The present study shows however, that by using ECMWF S3 seasonal forecasts the biggest skill improvement over the ESP procedure can be attained at lead times of 1-2 months, but less at longer lead

25 times when meteorological forcing plays a more important role on the skill. For lead times beyond 1-2 months an improvement in skill during most of the year still exists, but it should be noted that this improvement is observed only in the theoretical skill in forecasting low flows.

The results for the other tropical South American basin that we study, Parana, shows a somewhat similar

pattern to the Amazon, in the sense that the theoretical skill of ECMWF S3 in forecasting low flows is higher than ESP in some cases, whereas for high flows it is mostly lower (See Table 3). In contrast, the actual skill of ECMWF S3 in forecasting both high and low flows in the Parana is quite different than that in the Amazon. The percentage of theoretical skill attained by the actual skill of ECMWF S3 forecasts is much lower than in the Amazon. Averaged over the months of the year and different lead

35 times, it is 27% and 25% for low and high flows respectively. Notwithstanding, comparing the actual skill of the ECMWF S3 forecasts to the ESP, we see several months and lead-times where the actual skill is significantly improved by using ECMWF S3 forecasts, especially for forecasting high flows at longer lead times and during the first half of the year. For shorter lead times and for the second half of the year however, the actual skill of ECMWF S3 in forecasting high flows is significantly less than ESP. In

forecasting low flows, skill is also mostly reduced by using ECMWF S3 forecasts.





Another monsoon-dominated tropical river, the Brahmaputra in the Indian sub-continent shows a similar pattern to the Parana. In Table 4, we see again a significant improvement in the actual skill for forecasting high flows at longer lead times during the first half of the year. Just like the Parana, the skill is significantly lower at shorter lead times during the second half of the year. In contrast, the actual skill for forecasting low flows is significantly lower at longer lead times, and higher at a lead time of 1 month. The theoretical skill of ECMWF S3 in the Brahmaputra for forecasting both high and low flows is either very close to that of the ESP or lower. The percentage of the theoretical skill of ECMWF S3 that is actualized varies considerably for high and low flows, as well as over the year and the range of lead times. The averages are 24% and 34% for low and high flows respectively, ranging from as low as 2% for low flow forecasts in January to as high as 125% for high flow forecasts in April. The BS values for April high flows at all lead times are higher for actual skill calculations where the forecasted discharges are compared to actual discharge records, than the theoretical skill where they are compared to model simulations. Indeed, it was shown by Candogan Yossef et. al., (2012) that the skill of the ESP is below the climatology from April to September even for lead times of 1 month. The forecast skill in the Brahmaputra is strongly dominated by MF during the monsoon season for all lead times. The actual skill of ECMWF S3 during these months at a lead time of 1 month is significantly lower than ESP. This means the apparent potential for improvement in hydrological forecasts at short lead times by using ECMWF S3 seasonal meteorological forecasts cannot be realized at the moment.

In the two large rivers of China, the Yangtze and the Yellow River there exists a potential for improving forecasts beyond 1 month lead time through better MF during the high flow period (See Table 5 and 6). This period extends from May to October in the Yellow River and from April to September in the Yangtze. (Candogan Yossef et. al., 2012). Our results for the actual skill in forecasting high flows show that this opportunity may be partly realized in both rivers. The added skill of ECMWF S3 over ESP in forecasting higher than usual discharges during the high flow periods at longer lead times may aid the estimation of increased probability of flooding at lead times of 4-6 months. Moreover, the actual skill of ECMWF S3 is also better than ESP in forecasting low flows at short lead times during some months of the high flow periods, especially for the Yellow River. This may help a better estimation of the probability of less than expected discharges during high flow periods, at 1-2 month lead times.

The actual skill of ECMWF S3 forecasts in the Yangtze capture on average 23% of the theoretical skill for low flows, and 25% for high flows. These numbers are 22% and 26% in the Yellow River for low and high flows respectively. In both rivers, for both high and low flows, a significant pattern emerges in the ratios of actual to theoretical skill. The percentage of theoretical skill actualized is considerably higher during wet periods than during dry periods.

Similar to the Yellow River and the Yangtze, also in the Mekong basin forecast skill during the wet period from July to October is dominated by MF beyond 1 month lead time. However, the results for the Mekong are different from those for the Chinese basins. Added skill of ECMWF S3 over ESP in





forecasting higher than usual discharges during the wet periods can be seen not at longer lead times, but only at a lead-time of 1 month, as can be seen in Table 7. This may aid better estimation of flood probability at short notice. Beyond 1 month, ECMWF S3 forecasts are either less skilled or not significantly different than ESP. ECMWF S3 forecasts of lower than usual discharges during either the
wet or dry periods are less skilful than ESP at short lead times, but there are some months of improved skill at long lead times.

The percentages of theoretical skill of ECMWF S3 forecasts that is actualized in the Mekong are 37% and 60% for low and high flows respectively. During the high flow period from July to October, the
actual skill in forecasting higher than usual discharges reaches more than 80% of the theoretical skill.

### 4.2 Arctic basins:

In arctic basins, snowpack, ice and groundwater processes have a long memory, causing the forecast skill to be dominated by ICs for lead times up to 6 months (Candogan Yossef et. al., 2013). The North American arctic rivers Mackenzie and Nelson, as well as the Asian Ob and Lena are ice-bound for a
significant part of the year and peak discharges follow snowmelt. The skill of ESP forecasts is already quite high in these arctic rivers as would be expected for basin with such a large memory. Tables 8-11 show that the ECMWF S3 forecasts for these rivers are not significantly more skilful than the ESP. During May-June, which is the beginning of the high flow season in arctic rivers, one might expect some improvement in skill with ECMWF S3 forecasts over the ESP due to the temperature effect determining
the onset of snowmelt. However, there is no significant increase in the skill of ECMWF S3 forecasts over the ESP forecasts, not even during the beginning of the high flow season. ECMWF S3 forecasts perform very similar to ESP, and even worse in some cases. Especially the actual skill of ECMWF S3 forecasts in the arctic basins in Asia is considerably lower than that of the ESP forecasts.

The ratios of actual skill to theoretical skill are not very low in the arctic basins in general. Low ratios would be expected in areas where the model has large errors associated with snow and glaciers and consequent errors in the timing of peak discharges. In the river Ob for instance, where the discharge peaks in June, the percentage of skill actualized reaches 60-70%, so it may be concluded that the timing of the model is well approximated.

### 4.3 Temperate regions:

The ECMWF S3 forecasts are in general not significantly more skilful than ESP in the temperate European basins, Rhine, Danube and Volga as can be seen in Tables 12-14. There are some cases with improvement in the skill in forecasting flows lower than usual, especially in the theoretical skill. However, for high flows the ECMWF S3 forecasts perform worse than the ESP. In the Rhine basin,
where improvement in forecast skill depends on better climate forecasts, using the ECMWF S3 forecasts does not provide an improvement over the ESP. In the Danube and the Volga, we see an improvement in the theoretical skill in forecasting low flows during winter months. In the Danube and especially the Volga basins snowmelt and groundwater processes play a bigger role than the Rhine. Low flows during





winter months are actually dominated by the groundwater processes rather than the meteorological forcing. Nevertheless, this is where we see a consistent improvement in skill by using the ECMWF S3 forecasts. For high flows on the other hand, ECMWF S3 forecasts perform worse, both theoretically and actually.

The percentage of theoretical skill that is actualized is in general quite high for the European basins, but lower in temperate basins of North America. In the Columbia River forecasts are dominated by the ICs due to snow and the skill of ESP forecasts are already high. Using ECMWF S3 forecasts does not bring a significant improvement (See Table 15).

In the St. Lawrence River, peak flows are fed by spring and summer snowmelt accompanied by rain. Candogan Yossef et. al. (2013) concluded that the forecasting skill in spring and summer months depends largely on the snowpack accumulated during the previous winter months, dominating seasonal forecasts up to 6 months ahead. These findings are in disagreement with the results of Shukla and Lettenmaier (2011), which show that ESP forecasts initialized from December to April are skilful only for 1-2 months lead times. As it was mentioned in Candogan Yossef et. al. (2013), the disagreement is probably due to errors in one or both models in the estimation of snow accumulation. The results of the present study confirm the importance of ICs on the one hand. Table 16 shows that the theoretical skill of ECMWF S3 forecasts is considerably lower than the ESP in the St. Lawrence, especially for forecasting higher flows than usual during the summer months. On the other hand, the actual skill of the ECMWF S3 forecasts in forecasting lower than usual summer flows is significantly higher than the ESP, for 2, 3 and 4 months lead-times. This finding supports the conclusion of Shukla and Lettenmaier (2011) which emphasizes the importance of MF beyond 1-2 months lead times. Additionally, the fact that the ratio of actual skill to theoretical skill in St. Lawrence is rather on the low side may be an indication of errors in our model in representing the snow processes.

For the southeastern US rivers, the results of Candogan Yossef et. al. (2013) as well as those of Shukla and Lettenmaier (2011) show that skill due to ICs diminishes after 1-2 months lead time and that forecasts would benefit most from improvements in MF throughout the year. However, the results of the present study show that in general this potential improvement cannot be realized for the Mississippi by using ECMWF S3 forecasts. The skill of ECMWF S3 forecasts is similar to the ESP in most cases, as can be seen in Table 17, and it is lower than ESP in more case than it is higher, with no apparent pattern.

### 4.4 Semi-arid regions:

Candogan Yossef et. al. (2013) concluded that the relative importance of ICs is the lowest in this continent and any improvement of hydrological forecasts depends on better climate forecasts. The results of the present study for the Murray basin show that the theoretical skill of ECMWF S3 forecasts are significantly higher in some cases, but lower in other cases, with no apparent pattern (See Table 18). The actual skill of ECMWF S3 forecasts is lower than ESP in most cases. Also, the ratios of actual to theoretical skill are quite low in this basin for both high and low flows.



Similarly, in the semi-arid African basins of the Orange River and the Zambezi, where the knowledge of MF plays a very important role in the forecast skill, the performance of ECMWF S3 forecasts is worse compared to the ESP in most cases. Tables 19 and 20 show that the actual skill of ECMWF S3 in particular is lower than ESP in these basins. In contrast, in the Nile basin, the ICs dominate the forecast skill, resulting in high skills of ESP forecasts throughout the year assuming that the release strategy of the Aswan reservoir is known (Candogan Yossef et. al., 2013). The results of the present study show that the theoretical skill of ECMWF S3 cannot surpass the already high skill of the ESP (See Table 21). Actually, forecasts with ECMWF S3 perform considerably worse. The actual skill of the ESP forecasts in the Nile, however, is very low, due to the large effect of the reservoir operations. In fact, the ratio of actual to theoretical skill is the lowest by far in this basin. With such a low actual skill of ESP forecasts despite the dominance of IC's, comparison of ECMWF S3 skill to ESP is not very meaningful. Our results of actual skill in both high and low flows in the Nile appear to be very erratic indeed.

**5. Conclusions**

We assessed the skill of seasonal streamflow forecasts with the global hydrological forecasting system FEWS-World, setup within the GLOWASIS project. Global hydrological model PCR-GLOBWB was run with the ESP procedure as well as with ECMWF S3 bias-corrected seasonal meteorological forecast ensembles. We produced ensemble forecasts of monthly discharges for 20 large rivers of the world, with lead times of up to 6 months. We quantified the skill of ESP and ECMWF S3 in forecasting high and low flows by using the BS. We determined the theoretical skill by comparing the results against model simulations, as well as the actual skill by comparing against discharge observations. We quantified the added skill that may be obtained using ECMWF S3 meteorological forecast input compared to the skill obtained using the ESP input, by applying the BSS on the two ensembles. We also calculated the ratios of actual to theoretical skill, to quantify the percentage of the theoretical skill that is actualized.

We analysed these results in the context of prevailing hydroclimatic conditions. This analysis suggests that the skill varies considerably according to location, season and lead time. The conclusions can be summarized as follows:

- In general, the skill of the ECMWF S3 forecast run is close to that of the ESP forecast run.
- There are basins where the ECMWF S3 forecast performs significantly better than the ESP during certain periods of the year and at certain lead times.
- However, there are more cases where the skill of the ECMWF S3 forecast run is in fact lower than the skill of the ESP.
- In most cases, the apparent potential for improvement in seasonal hydrological forecasts by using better meteorological forecasts cannot be realized as yet with the model PCR-GLOBWB and the ECMWF S3 dataset.
- As more accurate global hydrological models and more skilful seasonal meteorological forecasts become available in the future, further studies will be needed to assess the




improvement in seasonal hydrological forecasts, as well as the effect of meteorological forecast quality vs. model errors on the hydrological forecasts.

**Acknowledgement:**

The forecast system, used in this research, has been set up in the 7th Framework Programme Project Global Water Scarcity Information Service (GLOWASIS). We acknowledge the 7th Framework Programme of the European Commission for the financial support. Furthermore, we acknowledge the European Centre for Medium-ranged Weather Forecasts for making available the ERA Interim – GPCP dataset and the ECMWF S3 seasonal forecasts ensemble used in this study as meteorological forcing.

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



| | theoretical (BS$_{theo}$) | | Actual (BS$_{act}$) | |
|---|---|---|---|---|
| | forecasted (p) | observed (o) | forecasted (p) | observed (o) |
| BS | ECMWF S3 | ERA 40 | ECMWF S3 | GRDC |
| BS$_{ref}$ | ESP | ERA 40 | ESP | GRDC |

**Table 1: Meteorological datasets used for calculating BS.**



| BS-theo-25 | LT=1 | LT=2 | LT=3 | LT=4 | LT=5 | LT=6 |
|---|---|---|---|---|---|---|
| Jan | 0.0945 | 0.0969 | 0.1322 | 0.1855 | 0.1838 | 0.2271 |
| Feb | 0.0536 | 0.2190 | 0.1510 | 0.1325 | 0.1921 | 0.2083 |
| Mar | 0.0617 | 0.1100 | 0.2171 | 0.1861 | 0.1798 | 0.1701 |
| Apr | 0.0961 | 0.0904 | 0.1496 | 0.1430 | 0.1972 | 0.1602 |
| May | 0.1143 | 0.0959 | 0.1039 | 0.1981 | 0.1548 | 0.1938 |
| Jun | 0.0964 | 0.0576 | 0.1562 | 0.1559 | 0.2263 | 0.2143 |
| Jul | 0.0620 | 0.0983 | 0.0848 | 0.1281 | 0.1592 | 0.1479 |
| Aug | 0.0094 | 0.0887 | 0.1047 | 0.1534 | 0.1857 | 0.2242 |
| Sep | 0.0000 | 0.0218 | 0.0821 | 0.0769 | 0.1187 | 0.1603 |
| Oct | 0.0377 | 0.0306 | 0.0730 | 0.0953 | 0.0807 | 0.1033 |
| Nov | 0.0835 | 0.1353 | 0.1176 | 0.1220 | 0.1565 | 0.1628 |
| Dec | 0.0204 | 0.0540 | 0.1675 | 0.1468 | 0.1344 | 0.1725 |

| BSS-theo-25 | LT=1 | LT=2 | LT=3 | LT=4 | LT=5 | LT=6 |
|---|---|---|---|---|---|---|
| Jan | 0.6035 | 0.5030 | 0.3498 | 0.2415 | 0.1824 | 0.1135 |
| Feb | 0.7420 | -0.1163 | 0.1377 | 0.2582 | -0.0666 | -0.0962 |
| Mar | 0.7241 | 0.5020 | -0.0078 | 0.0686 | 0.1169 | 0.1796 |
| Apr | 0.5905 | 0.5441 | 0.2439 | 0.3450 | 0.0367 | 0.2280 |
| May | 0.4650 | 0.5065 | 0.4497 | -0.0250 | 0.3155 | 0.0396 |
| Jun | 0.6213 | 0.7259 | 0.2436 | 0.1836 | -0.2316 | -0.0644 |
| Jul | 0.7755 | 0.5360 | 0.5793 | 0.3876 | 0.2068 | 0.2878 |
| Aug | 0.9663 | 0.5967 | 0.5164 | 0.2769 | 0.1501 | -0.1339 |
| Sep | 1.0000 | 0.9026 | 0.6335 | 0.6631 | 0.5115 | 0.3338 |
| Oct | 0.8642 | 0.8779 | 0.7467 | 0.6836 | 0.7170 | 0.6140 |
| Nov | 0.7421 | 0.4027 | 0.5031 | 0.5517 | 0.4528 | 0.4197 |
| Dec | 0.9147 | 0.7780 | 0.3435 | 0.3892 | 0.4940 | 0.3550 |

| BS-theo-75 | LT=1 | LT=2 | LT=3 | LT=4 | LT=5 | LT=6 |
|---|---|---|---|---|---|---|
| Jan | 0.1339 | 0.1223 | 0.1562 | 0.2249 | 0.2040 | 0.2431 |
| Feb | 0.0425 | 0.1215 | 0.1884 | 0.1872 | 0.2277 | 0.2169 |
| Mar | 0.1198 | 0.1237 | 0.1424 | 0.1835 | 0.1912 | 0.1753 |
| Apr | 0.0510 | 0.1256 | 0.1641 | 0.1609 | 0.1972 | 0.1795 |
| May | 0.1006 | 0.1521 | 0.1603 | 0.1841 | 0.1499 | 0.1736 |
| Jun | 0.1278 | 0.1229 | 0.1234 | 0.1675 | 0.1513 | 0.1777 |
| Jul | 0.0523 | 0.1399 | 0.1548 | 0.1333 | 0.1725 | 0.1926 |
| Aug | 0.0179 | 0.0576 | 0.1410 | 0.1253 | 0.1837 | 0.1529 |
| Sep | 0.0336 | 0.0132 | 0.0410 | 0.0994 | 0.1105 | 0.1157 |
| Oct | 0.0355 | 0.1017 | 0.0975 | 0.0981 | 0.1523 | 0.1614 |
| Nov | 0.0667 | 0.0758 | 0.1499 | 0.1868 | 0.1507 | 0.1893 |
| Dec | 0.0388 | 0.1634 | 0.2281 | 0.2716 | 0.2788 | 0.2300 |

| BSS-theo-75 | LT=1 | LT=2 | LT=3 | LT=4 | LT=5 | LT=6 |
|---|---|---|---|---|---|---|
| Jan | -0.2548 | 0.2532 | 0.1343 | -0.1560 | -0.0121 | -0.2327 |
| Feb | 0.2654 | 0.2177 | 0.0730 | 0.1520 | 0.0194 | 0.0297 |
| Mar | -0.2522 | 0.0932 | 0.2404 | 0.0722 | 0.0444 | 0.1737 |
| Apr | 0.3381 | -0.0486 | -0.0177 | 0.1299 | 0.0110 | 0.1216 |
| May | -0.8484 | 0.0728 | 0.0895 | -0.0325 | 0.2243 | 0.1314 |
| Jun | -0.1924 | 0.0824 | 0.1541 | -0.1249 | 0.0349 | -0.0073 |
| Jul | 0.4072 | -0.2330 | -0.0079 | 0.1141 | -0.0091 | -0.0591 |
| Aug | -1.4529 | 0.2865 | -0.3369 | 0.1743 | -0.1441 | 0.1487 |
| Sep | -1.8243 | 0.3850 | 0.1449 | 0.0055 | 0.2318 | 0.2686 |
| Oct | 0.4421 | -0.3032 | 0.0420 | 0.2386 | -0.0676 | 0.0663 |
| Nov | -3.9383 | 0.1711 | -0.1027 | -0.2042 | 0.0463 | -0.0815 |
| Dec | 0.2629 | -0.1166 | -0.2159 | -0.2984 | -0.3455 | -0.1473 |

| BS-act-25 | LT=1 | LT=2 | LT=3 | LT=4 | LT=5 | LT=6 |
|---|---|---|---|---|---|---|
| Jan | 0.2820 | 0.2500 | 0.2480 | 0.2330 | 0.3060 | 0.3310 |
| Feb | 0.1850 | 0.2550 | 0.2330 | 0.2140 | 0.2110 | 0.2710 |
| Mar | 0.2190 | 0.2420 | 0.2780 | 0.2020 | 0.2210 | 0.1920 |
| Apr | 0.3020 | 0.2120 | 0.2370 | 0.1980 | 0.2500 | 0.2010 |
| May | 0.3020 | 0.2600 | 0.1950 | 0.2360 | 0.2280 | 0.2660 |
| Jun | 0.3150 | 0.2170 | 0.2170 | 0.2590 | 0.2510 | 0.2450 |
| Jul | 0.3590 | 0.3170 | 0.2360 | 0.2740 | 0.2380 | 0.2800 |
| Aug | 0.2760 | 0.3010 | 0.2440 | 0.2690 | 0.2890 | 0.2670 |
| Sep | 0.2670 | 0.2040 | 0.2580 | 0.2650 | 0.2760 | 0.2820 |
| Oct | 0.3040 | 0.2180 | 0.2730 | 0.3380 | 0.2810 | 0.3280 |
| Nov | 0.3440 | 0.2630 | 0.2450 | 0.2800 | 0.2780 | 0.2540 |
| Dec | 0.3290 | 0.2660 | 0.3010 | 0.2860 | 0.3040 | 0.3540 |

| BSS-act-25 | LT=1 | LT=2 | LT=3 | LT=4 | LT=5 | LT=6 |
|---|---|---|---|---|---|---|
| Jan | -0.183415 | -0.282282 | -0.21945 | 0.047337 | -0.361122 | -0.291848 |
| Feb | 0.109216 | -0.299797 | -0.330205 | -0.197899 | -0.171694 | -0.42598 |
| Mar | 0.020716 | -0.095526 | -0.290618 | -0.010995 | -0.085374 | 0.074202 |
| Apr | -0.286211 | -0.069575 | -0.197671 | 0.092906 | -0.22116 | 0.031188 |
| May | -0.413167 | -0.338338 | -0.033184 | -0.221348 | -0.008011 | -0.318324 |
| Jun | -0.237284 | -0.285156 | -0.050794 | -0.356141 | -0.36618 | -0.21677 |
| Jul | -0.300528 | -0.49566 | -0.170097 | -0.309879 | -0.185562 | -0.348213 |
| Aug | 0.005779 | -0.368465 | -0.12717 | -0.26763 | -0.322915 | -0.350163 |
| Sep | -0.02464 | 0.086594 | -0.151665 | -0.161478 | -0.135644 | -0.17179 |
| Oct | -0.093674 | 0.12936 | 0.052687 | -0.12204 | 0.014688 | -0.225661 |
| Nov | -0.062826 | -0.161328 | -0.034983 | -0.028653 | 0.02787 | 0.094688 |
| Dec | -0.377147 | -0.093619 | -0.17973 | -0.189698 | -0.14419 | -0.323944 |

| BS-act-75 | LT=1 | LT=2 | LT=3 | LT=4 | LT=5 | LT=6 |
|---|---|---|---|---|---|---|
| Jan | 0.2975 | 0.2727 | 0.2753 | 0.2562 | 0.2793 | 0.3121 |
| Feb | 0.3121 | 0.2427 | 0.2573 | 0.2750 | 0.2841 | 0.2921 |
| Mar | 0.2895 | 0.2397 | 0.2697 | 0.2713 | 0.2163 | 0.2066 |
| Apr | 0.3479 | 0.2650 | 0.2551 | 0.2700 | 0.2348 | 0.2422 |
| May | 0.3733 | 0.2733 | 0.2512 | 0.2249 | 0.2468 | 0.2425 |
| Jun | 0.3036 | 0.2689 | 0.2281 | 0.2360 | 0.2444 | |
| Jul | 0.3251 | 0.3096 | 0.2639 | 0.2424 | 0.2270 | 0.2898 |
| Aug | 0.1391 | 0.2138 | 0.2284 | 0.2444 | 0.2256 | |
| Sep | 0.1003 | 0.1344 | 0.1623 | 0.2146 | 0.2196 | 0.2127 |
| Oct | 0.1689 | 0.1986 | 0.1642 | 0.1890 | 0.2190 | 0.2099 |
| Nov | 0.2000 | 0.1970 | 0.2165 | 0.2898 | 0.2901 | 0.2680 |
| Dec | 0.3540 | 0.2240 | 0.2463 | 0.2898 | 0.3333 | 0.3331 |

| BSS-act-75 | LT=1 | LT=2 | LT=3 | LT=4 | LT=5 | LT=6 |
|---|---|---|---|---|---|---|
| Jan | -0.2736 | -0.5124 | -0.3641 | -0.2000 | -0.2794 | -0.4431 |
| Feb | 0.0963 | -0.2508 | -0.3122 | -0.3316 | -0.3326 | -0.2663 |
| Mar | -0.2641 | -0.2575 | -0.1593 | -0.4054 | -0.0436 | 0.0969 |
| Apr | -0.2196 | -0.4878 | -0.4502 | -0.2310 | -0.2214 | -0.1713 |
| May | -0.1858 | -0.1533 | -0.3775 | -0.1660 | -0.2100 | -0.1454 |
| Jun | -0.3073 | -0.2600 | -0.5175 | -0.2517 | -0.1768 | -0.1359 |
| Jul | 0.0679 | -0.2868 | -0.3807 | -0.1967 | -0.0891 | -0.3734 |
| Aug | 0.2190 | 0.0009 | -0.1841 | -0.2063 | -0.1718 | -0.1143 |
| Sep | 0.0507 | -0.0356 | -0.1789 | -0.2121 | -0.1094 | -0.0423 |
| Oct | 0.0953 | -0.4792 | -0.0819 | -0.1361 | -0.2012 | 0.0214 |
| Nov | -0.1925 | 0.0438 | 0.0903 | -0.2480 | -0.1915 | -0.1189 |
| Dec | -0.3348 | -0.2767 | -0.1818 | -0.3994 | -0.5316 | -0.5488 |

| BS-A/T-25 | LT=1 | LT=2 | LT=3 | LT=4 | LT=5 | LT=6 |
|---|---|---|---|---|---|---|
| Jan | 33.51 | 38.76 | 53.32 | 79.62 | 60.07 | 68.62 |
| Feb | 28.96 | 85.89 | 64.82 | 61.92 | 91.03 | 76.87 |
| Mar | 28.18 | 45.46 | 78.09 | 92.13 | 81.37 | 88.61 |
| Apr | 31.84 | 42.62 | 63.13 | 72.21 | 78.88 | 79.68 |
| May | 37.86 | 36.87 | 53.26 | 83.92 | 67.90 | 72.85 |
| Jun | 30.61 | 21.32 | 71.98 | 60.20 | 90.15 | 87.48 |
| Jul | 17.27 | 35.95 | 46.75 | 66.90 | 52.82 | |
| Aug | 3.39 | 29.47 | 42.90 | 57.04 | 64.25 | 83.99 |
| Sep | 0.00 | 10.67 | 31.82 | 29.00 | 43.02 | 56.85 |
| Oct | 12.41 | 14.03 | 26.74 | 28.20 | 28.72 | 31.50 |
| Nov | 24.26 | 51.43 | 48.01 | 43.59 | 56.29 | 64.10 |
| Dec | 6.20 | 20.30 | 55.65 | 51.34 | 44.22 | 48.72 |

| BS-A/T-75 | LT=1 | LT=2 | LT=3 | LT=4 | LT=5 | LT=6 |
|---|---|---|---|---|---|---|
| Jan | 45.00 | 44.83 | 56.73 | 87.76 | 73.06 | 77.90 |
| Feb | 13.61 | 50.06 | 73.20 | 68.08 | 80.14 | 74.24 |
| Mar | 41.39 | 51.61 | 52.81 | 67.65 | 88.41 | 84.83 |
| Apr | 14.65 | 47.40 | 64.36 | 59.59 | 83.98 | 74.12 |
| May | 26.94 | 55.65 | 63.82 | 81.88 | 60.71 | 71.56 |
| Jun | 42.11 | 51.62 | 45.90 | 73.43 | 64.13 | 72.72 |
| Jul | 16.10 | 45.20 | 58.66 | 55.97 | 75.97 | 66.47 |
| Aug | 12.87 | 30.16 | 65.98 | 54.89 | 75.20 | 67.77 |
| Sep | 33.52 | 9.84 | 25.30 | 46.34 | 50.31 | 54.40 |
| Oct | 21.04 | 51.18 | 59.40 | 51.90 | 69.56 | 76.90 |
| Nov | 33.33 | 38.46 | 69.21 | 64.45 | 51.95 | 70.61 |
| Dec | 10.97 | 72.94 | 92.62 | 93.73 | 83.64 | 69.07 |

**Table 2: Skill scores in the Amazon**



| BS-theo-25 | LT=1 | LT=2 | LT=3 | LT=4 | LT=5 | LT=6 |
|---|---|---|---|---|---|---|
| Jan | 0.1180 | 0.1000 | 0.2140 | 0.2200 | 0.2300 | 0.2180 |
| Feb | 0.0294 | 0.1520 | 0.1980 | 0.2590 | 0.1920 | 0.2330 |
| Mar | 0.0763 | 0.1980 | 0.2180 | 0.2060 | 0.2310 | 0.2310 |
| Apr | 0.0744 | 0.0755 | 0.1620 | 0.1700 | 0.2140 | 0.2130 |
| May | 0.0449 | 0.1500 | 0.1010 | 0.1800 | 0.1760 | 0.2040 |
| Jun | 0.1140 | 0.1310 | 0.2480 | 0.2110 | 0.2270 | 0.2190 |
| Jul | 0.1350 | 0.1550 | 0.1070 | 0.2200 | 0.2000 | 0.2290 |
| Aug | 0.0405 | 0.1040 | 0.1260 | 0.0926 | 0.1410 | 0.1560 |
| Sep | 0.0570 | 0.0978 | 0.1200 | 0.1730 | 0.1380 | 0.2230 |
| Oct | 0.0650 | 0.2090 | 0.2290 | 0.1870 | 0.2700 | 0.2220 |
| Nov | 0.0444 | 0.1860 | 0.1550 | 0.2430 | 0.1710 | 0.2120 |
| Dec | 0.0725 | 0.1480 | 0.1760 | 0.1870 | 0.1740 | 0.1550 |

| BSS-theo-25 | LT=1 | LT=2 | LT=3 | LT=4 | LT=5 | LT=6 |
|---|---|---|---|---|---|---|
| Jan | -1.4481 | 0.0000 | -0.4459 | -0.1702 | -0.1330 | -0.0283 |
| Feb | 0.1901 | -0.1692 | -0.1579 | -0.3704 | 0.0052 | -0.1650 |
| Mar | 0.2446 | -0.2941 | -0.1295 | -0.1196 | -0.1436 | -0.1268 |
| Apr | -0.6352 | 0.4683 | 0.0122 | 0.0341 | -0.0594 | -0.0545 |
| May | -0.0022 | -0.7647 | 0.3221 | -0.0714 | -0.0057 | -0.0303 |
| Jun | -0.9487 | 0.2384 | -0.3478 | -0.0821 | -0.1350 | -0.0950 |
| Jul | -0.5625 | -0.2302 | 0.4456 | -0.1055 | 0.0099 | -0.0362 |
| Aug | -0.0305 | -0.2855 | -0.0080 | 0.3241 | 0.1076 | 0.1034 |
| Sep | -0.1377 | 0.1782 | 0.1111 | 0.1636 | 0.2995 | -0.0825 |
| Oct | 0.1144 | -0.1484 | -0.1684 | 0.1053 | -0.2796 | -0.0521 |
| Nov | 0.1913 | -0.3286 | 0.0663 | -0.2462 | 0.1493 | -0.0242 |
| Dec | 0.2464 | -0.2870 | -0.0732 | 0.0209 | 0.1168 | 0.2132 |

| BS-theo-75 | LT=1 | LT=2 | LT=3 | LT=4 | LT=5 | LT=6 |
|---|---|---|---|---|---|---|
| Jan | 0.0871 | 0.1170 | 0.1800 | 0.2040 | 0.1830 | 0.2100 |
| Feb | 0.1130 | 0.1850 | 0.1990 | 0.2140 | 0.2000 | 0.1810 |
| Mar | 0.1180 | 0.1390 | 0.2130 | 0.2110 | 0.2460 | 0.2830 |
| Apr | 0.1940 | 0.1270 | 0.2310 | 0.1490 | 0.2050 | 0.2430 |
| May | 0.1300 | 0.2550 | 0.1590 | 0.1670 | 0.1650 | 0.1780 |
| Jun | 0.1610 | 0.1320 | 0.2160 | 0.1660 | 0.1920 | 0.1680 |
| Jul | 0.0361 | 0.1120 | 0.1470 | 0.2060 | 0.1710 | 0.1570 |
| Aug | 0.0289 | 0.1020 | 0.1320 | 0.1110 | 0.1770 | 0.1380 |
| Sep | 0.0281 | 0.0708 | 0.0912 | 0.1380 | 0.1500 | 0.1620 |
| Oct | 0.0579 | 0.0603 | 0.0697 | 0.1310 | 0.1560 | 0.1270 |
| Nov | 0.0061 | 0.0727 | 0.0901 | 0.1050 | 0.1190 | 0.1570 |
| Dec | 0.0460 | 0.1570 | 0.1490 | 0.1460 | 0.1470 | 0.1440 |

| BSS-theo-75 | LT=1 | LT=2 | LT=3 | LT=4 | LT=5 | LT=6 |
|---|---|---|---|---|---|---|
| Jan | -0.8184 | 0.0085 | -0.0909 | -0.1724 | 0.0054 | -0.1538 |
| Feb | -0.7767 | -0.6667 | -0.4962 | -0.3806 | -0.1765 | -0.0343 |
| Mar | -0.9187 | 0.1090 | -0.3148 | -0.1405 | -0.2118 | -0.3738 |
| Apr | -0.9836 | 0.2160 | -0.2762 | 0.0449 | -0.2275 | -0.2857 |
| May | -1.6263 | -0.9173 | 0.1405 | 0.0670 | 0.0000 | 0.0326 |
| Jun | -2.2329 | -0.0560 | -0.3252 | 0.1309 | 0.0154 | 0.0667 |
| Jul | 0.3111 | 0.0894 | 0.1695 | -0.1196 | 0.1140 | 0.2189 |
| Aug | 0.4939 | -0.0472 | 0.0149 | 0.3019 | 0.0684 | 0.2698 |
| Sep | -0.0808 | 0.3843 | 0.3577 | 0.1636 | 0.1620 | 0.0899 |
| Oct | -1.5619 | -0.1024 | 0.4143 | 0.0296 | 0.0659 | 0.2825 |
| Nov | 0.7220 | 0.0558 | 0.1657 | 0.2500 | 0.2171 | 0.0655 |
| Dec | 0.3285 | -0.5747 | -0.3423 | -0.1145 | -0.0280 | 0.1056 |

| BS-act-25 | LT=1 | LT=2 | LT=3 | LT=4 | LT=5 | LT=6 |
|---|---|---|---|---|---|---|
| Jan | 0.6570 | 0.6490 | 0.6750 | 0.6120 | 0.6100 | 0.5840 |
| Feb | 0.6660 | 0.6910 | 0.6340 | 0.6130 | 0.6160 | 0.5870 |
| Mar | 0.6160 | 0.6370 | 0.5990 | 0.6600 | 0.6410 | 0.6230 |
| Apr | 0.6740 | 0.6390 | 0.5940 | 0.5820 | 0.6750 | 0.5920 |
| May | 0.7120 | 0.6890 | 0.5740 | 0.6060 | 0.5700 | 0.5960 |
| Jun | 0.8050 | 0.6700 | 0.6180 | 0.6050 | 0.5630 | 0.5160 |
| Jul | 0.8560 | 0.7010 | 0.6100 | 0.6080 | 0.5450 | 0.5140 |
| Aug | 0.7740 | 0.7400 | 0.6530 | 0.5830 | 0.6020 | 0.5440 |
| Sep | 0.6990 | 0.7280 | 0.6230 | 0.6150 | 0.6110 | 0.6040 |
| Oct | 0.7200 | 0.5910 | 0.6110 | 0.6110 | 0.6400 | 0.6280 |
| Nov | 0.7350 | 0.6830 | 0.5970 | 0.5520 | 0.6130 | 0.5700 |
| Dec | 0.7090 | 0.6760 | 0.6250 | 0.6170 | 0.5500 | 0.5910 |

| BSS-act-25 | LT=1 | LT=2 | LT=3 | LT=4 | LT=5 | LT=6 |
|---|---|---|---|---|---|---|
| Jan | 0.0492 | -0.0674 | -0.1269 | -0.0268 | -0.0739 | -0.0121 |
| Feb | 0.0826 | -0.0848 | -0.1382 | 0.0510 | -0.1180 | 0.0200 |
| Mar | 0.1238 | 0.0275 | 0.0952 | -0.1765 | -0.1805 | -0.1601 |
| Apr | 0.0426 | -0.0510 | -0.0224 | | -0.2273 | 0.0017 |
| May | 0.0193 | -0.0439 | 0.0651 | -0.0050 | -0.0215 | -0.1352 |
| Jun | -0.0938 | -0.0652 | 0.0112 | 0.0040 | 0.0053 | 0.1179 |
| Jul | -0.2073 | -0.0868 | -0.0099 | 0.0065 | 0.0652 | 0.0838 |
| Aug | -0.1089 | -0.0996 | -0.0415 | 0.0411 | -0.0326 | 0.0198 |
| Sep | 0.0224 | -0.0882 | -0.0081 | -0.0233 | -0.0645 | -0.1269 |
| Oct | -0.1285 | 0.0880 | -0.0589 | -0.0498 | -0.1130 | -0.1115 |
| Nov | -0.1520 | -0.1498 | -0.0136 | 0.0995 | -0.0355 | 0.0206 |
| Dec | -0.0858 | -0.1477 | -0.1161 | -0.0565 | 0.0998 | 0.0248 |

| BS-act-75 | LT=1 | LT=2 | LT=3 | LT=4 | LT=5 | LT=6 |
|---|---|---|---|---|---|---|
| Jan | 0.7660 | 0.6910 | 0.5960 | 0.6020 | 0.5620 | 0.5640 |
| Feb | 0.8370 | 0.6820 | 0.6000 | 0.5400 | 0.5330 | 0.5260 |
| Mar | 0.8390 | 0.6750 | 0.6130 | 0.5340 | 0.5060 | 0.5050 |
| Apr | 0.7700 | 0.6790 | 0.6220 | 0.5800 | 0.5470 | 0.4780 |
| May | 0.7910 | 0.5460 | 0.5960 | 0.5960 | 0.5590 | 0.5640 |
| Jun | 0.8150 | 0.6780 | 0.5190 | 0.5230 | 0.5560 | 0.5560 |
| Jul | 0.7630 | 0.7180 | 0.6800 | 0.5760 | 0.5280 | 0.5520 |
| Aug | 0.7020 | 0.7390 | 0.6890 | 0.6560 | 0.5650 | 0.5320 |
| Sep | 0.6520 | 0.7130 | 0.6610 | 0.6890 | 0.6600 | 0.5870 |
| Oct | 0.6640 | 0.6420 | 0.6940 | 0.6710 | 0.6830 | 0.6300 |
| Nov | 0.7030 | 0.6420 | 0.6540 | 0.6870 | 0.6760 | 0.5930 |
| Dec | 0.7790 | 0.6240 | 0.6400 | 0.6120 | 0.5900 | 0.5930 |

| BSS-act-75 | LT=1 | LT=2 | LT=3 | LT=4 | LT=5 | LT=6 |
|---|---|---|---|---|---|---|
| Jan | -0.0713 | -0.0267 | 0.0132 | -0.0101 | -0.0237 | -0.0311 |
| Feb | -0.1403 | -0.1180 | -0.0508 | 0.0510 | 0.0566 | 0.1174 |
| Mar | -0.1800 | -0.1538 | -0.0569 | 0.0648 | 0.1650 | 0.1293 |
| Apr | -0.1702 | -0.1727 | -0.0596 | 0.0429 | 0.0974 | 0.1898 |
| May | -0.3430 | -0.0361 | -0.0329 | -0.0102 | 0.1320 | 0.1790 |
| Jun | -0.0954 | -0.2486 | 0.0716 | 0.1121 | 0.0974 | 0.0989 |
| Jul | -0.0283 | -0.1115 | -0.2364 | -0.0378 | 0.0704 | 0.0450 |
| Aug | 0.0462 | -0.1130 | -0.1314 | -0.1884 | -0.0386 | 0.0764 |
| Sep | 0.1117 | -0.1354 | -0.1241 | -0.1718 | -0.2336 | -0.0408 |
| Oct | 0.0816 | 0.0258 | -0.1490 | -0.1315 | -0.2509 | -0.1624 |
| Nov | 0.0343 | 0.0346 | -0.0531 | -0.1906 | -0.2246 | -0.0665 |
| Dec | -0.1081 | 0.0560 | -0.0458 | -0.0552 | -0.0369 | -0.0570 |

| BS-A/T-25 | LT=1 | LT=2 | LT=3 | LT=4 | LT=5 | LT=6 |
|---|---|---|---|---|---|---|
| Jan | 17.96 | 15.41 | 31.70 | 35.95 | 37.70 | 37.33 |
| Feb | 4.41 | 22.00 | 31.23 | 42.25 | 39.67 | 39.69 |
| Mar | 12.39 | 31.08 | 36.39 | 31.21 | 36.04 | 37.08 |
| Apr | 11.04 | 11.82 | 27.27 | 29.21 | 31.70 | 35.98 |
| May | 6.31 | 21.77 | 17.60 | 29.70 | 30.88 | 34.23 |
| Jun | 14.16 | 19.55 | 40.13 | 34.88 | 40.32 | 42.44 |
| Jul | 15.77 | 22.11 | 17.54 | 36.18 | 36.70 | 44.55 |
| Aug | 5.23 | 14.05 | 19.30 | 15.88 | 23.42 | 28.68 |
| Sep | 8.15 | 13.43 | 19.26 | 28.13 | 22.59 | 36.92 |
| Oct | 9.03 | 35.36 | 37.48 | 30.61 | 42.19 | 35.35 |
| Nov | 6.04 | 27.23 | 25.96 | 44.02 | 27.90 | 37.19 |
| Dec | 10.23 | 21.89 | 28.16 | 30.31 | 31.64 | 26.23 |

| BS-A/T-75 | LT=1 | LT=2 | LT=3 | LT=4 | LT=5 | LT=6 |
|---|---|---|---|---|---|---|
| Jan | 11.37 | 16.93 | 30.20 | 33.89 | 32.56 | 37.23 |
| Feb | 13.50 | 27.13 | 33.17 | 39.63 | 37.52 | 34.41 |
| Mar | 14.06 | 20.59 | 34.75 | 39.51 | 48.62 | 56.04 |
| Apr | 25.19 | 18.70 | 37.14 | 25.69 | 37.48 | 50.84 |
| May | 16.43 | 46.70 | 26.68 | 28.02 | 29.52 | 31.56 |
| Jun | 19.75 | 19.47 | 41.62 | 31.74 | 33.28 | 30.22 |
| Jul | 4.73 | 15.60 | 21.62 | 35.76 | 32.39 | 28.44 |
| Aug | 4.12 | 13.80 | 19.16 | 16.92 | 31.33 | 25.94 |
| Sep | 4.31 | 9.93 | 13.80 | 20.03 | 22.73 | 27.60 |
| Oct | 8.72 | 9.39 | 10.04 | 19.52 | 22.84 | 20.16 |
| Nov | 0.86 | 11.32 | 13.78 | 15.28 | 17.60 | 26.48 |
| Dec | 5.91 | 25.16 | 23.28 | 23.86 | 24.92 | 24.28 |

**Table 3: Skill scores in the Parana**





| BS-theo-25 | LT=1 | LT=2 | LT=3 | LT=4 | LT=5 | LT=6 |
|---|---|---|---|---|---|---|
| Jan | 0.0567 | 0.0359 | 0.0105 | 0.0584 | 0.1310 | 0.1450 |
| Feb | 0.0530 | 0.0278 | 0.0142 | 0.0313 | 0.0886 | 0.1470 |
| Mar | 0.0284 | 0.0547 | 0.0912 | 0.0909 | 0.0949 | 0.1110 |
| Apr | 0.1250 | 0.1670 | 0.1940 | 0.1630 | 0.1920 | 0.2100 |
| May | 0.1800 | 0.2030 | 0.2150 | 0.2390 | 0.1790 | 0.2180 |
| Jun | 0.2410 | 0.2550 | 0.2390 | 0.2310 | 0.2360 | 0.2290 |
| Jul | 0.1710 | 0.2360 | 0.2540 | 0.2260 | 0.2200 | 0.1950 |
| Aug | 0.1220 | 0.1540 | 0.2090 | 0.2090 | 0.1950 | 0.2540 |
| Sep | 0.1290 | 0.1290 | 0.1790 | 0.1870 | 0.1940 | 0.2050 |
| Oct | 0.0813 | 0.1310 | 0.1600 | 0.1960 | 0.2040 | 0.2110 |
| Nov | 0.0846 | 0.0510 | 0.1350 | 0.1280 | 0.1590 | 0.1880 |
| Dec | 0.0501 | 0.0240 | 0.0394 | 0.1290 | 0.1280 | 0.1480 |

| BSS-theo-25 | LT=1 | LT=2 | LT=3 | LT=4 | LT=5 | LT=6 |
|---|---|---|---|---|---|---|
| Jan | -544.1923 | -10.5064 | -0.4810 | 0.1911 | -0.0155 | -0.0902 |
| Feb | -26.1795 | -3.6880 | -1.0610 | -0.3973 | -0.3758 | -0.1136 |
| Mar | 0.3772 | 0.0666 | -0.2095 | -0.1639 | -0.1846 | -0.0571 |
| Apr | 0.0530 | -0.0915 | -0.4265 | -0.1560 | -0.2308 | -0.4000 |
| May | -0.3333 | -0.1154 | -0.2356 | -0.3427 | 0.0000 | -0.1474 |
| Jun | -0.2888 | -0.3421 | -0.1327 | -0.1214 | -0.1185 | -0.1393 |
| Jul | -0.0556 | -0.2356 | -0.4033 | -0.1832 | -0.1957 | -0.0428 |
| Aug | 0.1921 | 0.1809 | -0.0609 | -0.0609 | 0.0000 | -0.2764 |
| Sep | 0.1103 | 0.1042 | -0.0655 | -0.0275 | 0.0202 | -0.0459 |
| Oct | -0.0012 | 0.0224 | -0.1189 | -0.1879 | -0.1271 | -0.0821 |
| Nov | -15.4591 | 0.4097 | 0.0217 | 0.1111 | 0.0914 | -0.0053 |
| Dec | -0.9494 | 0.0283 | 0.4573 | 0.0227 | 0.0376 | 0.1445 |

| BS-theo-75 | LT=1 | LT=2 | LT=3 | LT=4 | LT=5 | LT=6 |
|---|---|---|---|---|---|---|
| Jan | 0.0678 | 0.0476 | 0.0456 | 0.0547 | 0.1270 | 0.1320 |
| Feb | 0.0522 | 0.0647 | 0.0675 | 0.0727 | 0.0955 | 0.1680 |
| Mar | 0.0829 | 0.1330 | 0.1010 | 0.1000 | 0.1180 | 0.0958 |
| Apr | 0.1870 | 0.2010 | 0.2690 | 0.2460 | 0.2390 | 0.2190 |
| May | 0.1910 | 0.2210 | 0.2040 | 0.2050 | 0.1960 | 0.2200 |
| Jun | 0.2740 | 0.2180 | 0.1950 | 0.1980 | 0.2170 | 0.2310 |
| Jul | 0.1830 | 0.2580 | 0.2260 | 0.2210 | 0.2510 | 0.2220 |
| Aug | 0.1690 | 0.1690 | 0.2070 | 0.2100 | 0.2030 | 0.2280 |
| Sep | 0.0978 | 0.1620 | 0.1840 | 0.1560 | 0.1560 | 0.1220 |
| Oct | 0.1040 | 0.1910 | 0.2000 | 0.2640 | 0.2360 | 0.2360 |
| Nov | 0.1330 | 0.0906 | 0.1130 | 0.1540 | 0.1700 | 0.1450 |
| Dec | 0.1610 | 0.0567 | 0.0493 | 0.1340 | 0.1930 | 0.1440 |

| BSS-theo-75 | LT=1 | LT=2 | LT=3 | LT=4 | LT=5 | LT=6 |
|---|---|---|---|---|---|---|
| Jan | -12.1141 | -2.0127 | -1.5909 | 0.0180 | -0.1981 | -0.1892 |
| Feb | -0.9848 | 0.0167 | 0.0189 | -0.1049 | 0.0545 | -0.0980 |
| Mar | -0.1776 | -0.6259 | -0.2347 | -0.2107 | -0.5226 | -0.0597 |
| Apr | -0.4275 | -0.1618 | -0.5824 | -0.4819 | -0.4663 | -0.4408 |
| May | -0.0214 | -0.0625 | -0.0625 | -0.0963 | 0.0000 | -0.1518 |
| Jun | -0.8639 | -0.1414 | 0.0051 | -0.0051 | -0.1186 | -0.2554 |
| Jul | -0.1159 | -0.5542 | -0.1832 | -0.1510 | -0.2938 | -0.1684 |
| Aug | -0.3520 | 0.0000 | -0.1374 | -0.1932 | -0.0684 | -0.1400 |
| Sep | -0.0252 | -0.0519 | -0.0337 | 0.1333 | 0.1832 | 0.3807 |
| Oct | 0.2000 | 0.0354 | -0.1050 | -0.3822 | -0.2620 | -0.2826 |
| Nov | -13.1791 | 0.0940 | 0.1871 | 0.0217 | -0.0559 | 0.1159 |
| Dec | -25.5677 | -2.3158 | 0.3679 | -0.0152 | -0.4621 | 0.0069 |

| BS-act-25 | LT=1 | LT=2 | LT=3 | LT=4 | LT=5 | LT=6 |
|---|---|---|---|---|---|---|
| Jan | 0.6570 | 0.6490 | 0.6750 | 0.6120 | 0.6100 | 0.5840 |
| Feb | 0.6660 | 0.6910 | 0.6340 | 0.6130 | 0.6160 | 0.5870 |
| Mar | 0.6160 | 0.6370 | 0.5990 | 0.6600 | 0.6410 | 0.6230 |
| Apr | 0.6740 | 0.6390 | 0.5940 | 0.5820 | 0.6750 | 0.5920 |
| May | 0.7120 | 0.6890 | 0.5740 | 0.6060 | 0.5700 | 0.5960 |
| Jun | 0.8050 | 0.6700 | 0.6180 | 0.6050 | 0.5630 | 0.5140 |
| Jul | 0.8560 | 0.7010 | 0.6100 | 0.6080 | 0.5450 | 0.5140 |
| Aug | 0.7740 | 0.7400 | 0.6530 | 0.5830 | 0.6020 | 0.5440 |
| Sep | 0.6990 | 0.7280 | 0.6230 | 0.6150 | 0.6110 | 0.6040 |
| Oct | 0.7200 | 0.5910 | 0.6110 | 0.6110 | 0.6400 | 0.6280 |
| Nov | 0.7350 | 0.6830 | 0.5970 | 0.5520 | 0.6130 | 0.5700 |
| Dec | 0.7090 | 0.6760 | 0.6250 | 0.6170 | 0.5500 | 0.5910 |

| BSS-act-25 | LT=1 | LT=2 | LT=3 | LT=4 | LT=5 | LT=6 |
|---|---|---|---|---|---|---|
| Jan | 0.0492 | -0.0674 | -0.1269 | -0.0268 | -0.0739 | -0.0121 |
| Feb | 0.0826 | -0.0848 | -0.1382 | -0.0698 | -0.1180 | 0.0200 |
| Mar | 0.1238 | 0.0275 | 0.0952 | -0.1765 | -0.1805 | -0.1601 |
| Apr | 0.0426 | -0.0510 | -0.0224 | 0.0085 | -0.2273 | 0.0017 |
| May | 0.0193 | -0.0439 | 0.0651 | -0.0050 | -0.0215 | -0.1352 |
| Jun | -0.0938 | -0.0652 | 0.0112 | 0.0053 | | 0.1179 |
| Jul | -0.2073 | -0.0868 | -0.0099 | 0.0065 | 0.0652 | 0.0838 |
| Aug | -0.1089 | -0.0996 | -0.0415 | 0.0411 | -0.0326 | 0.0198 |
| Sep | 0.0224 | -0.0882 | -0.0081 | -0.0233 | -0.0645 | -0.1269 |
| Oct | -0.1285 | 0.0880 | -0.0589 | -0.0498 | -0.1130 | -0.1115 |
| Nov | -0.1520 | -0.1498 | -0.0136 | 0.0995 | -0.0355 | 0.0206 |
| Dec | -0.0858 | -0.1477 | -0.1161 | -0.0565 | 0.0998 | 0.0248 |

| BS-act-75 | LT=1 | LT=2 | LT=3 | LT=4 | LT=5 | LT=6 |
|---|---|---|---|---|---|---|
| Jan | 0.4230 | 0.4500 | 0.4680 | 0.5100 | 0.4880 | 0.5080 |
| Feb | 0.4170 | 0.4370 | 0.4590 | 0.4760 | 0.5150 | 0.5110 |
| Mar | 0.4370 | 0.4120 | 0.3910 | 0.3860 | 0.4460 | 0.5120 |
| Apr | 0.4710 | 0.4350 | 0.3790 | 0.3750 | 0.3960 | 0.4070 |
| May | 0.5470 | 0.4670 | 0.4240 | 0.4770 | 0.4980 | 0.4430 |
| Jun | 0.3900 | 0.4200 | 0.4900 | 0.4640 | 0.5490 | 0.5480 |
| Jul | 0.3800 | 0.4630 | 0.4330 | 0.5140 | 0.5200 | 0.5560 |
| Aug | 0.3860 | 0.4900 | 0.5090 | 0.4790 | 0.5250 | 0.4870 |
| Sep | 0.4230 | 0.4710 | 0.4540 | 0.5170 | 0.4820 | 0.5470 |
| Oct | 0.3330 | 0.4190 | 0.4720 | 0.5260 | 0.5400 | 0.5230 |
| Nov | 0.3910 | 0.4300 | 0.4600 | 0.4820 | 0.4950 | 0.5310 |
| Dec | 0.4170 | 0.4270 | 0.4790 | 0.4780 | 0.4940 | 0.5020 |

| BSS-act-75 | LT=1 | LT=2 | LT=3 | LT=4 | LT=5 | LT=6 |
|---|---|---|---|---|---|---|
| Jan | 0.1540 | 0.0405 | 0.0168 | -0.0759 | -0.0893 | -0.1239 |
| Feb | 0.1627 | 0.1207 | 0.0255 | -0.1096 | -0.0981 | -0.1483 |
| Mar | 0.0622 | 0.0885 | 0.1803 | 0.0721 | -0.1320 | -0.2580 |
| Apr | -0.2395 | 0.0972 | -0.0954 | -0.0458 | -0.1250 | -0.1530 |
| May | -0.2993 | -0.1588 | -0.0192 | -0.1955 | -0.1829 | -0.0911 |
| Jun | 0.0347 | 0.0192 | -0.1395 | -0.0213 | -0.2013 | -0.2598 |
| Jul | 0.1593 | -0.1077 | -0.0261 | -0.1981 | -0.2264 | -0.3301 |
| Aug | -0.0266 | -0.1086 | -0.1236 | -0.1244 | -0.1364 | -0.0703 |
| Sep | 0.0231 | -0.0903 | -0.0111 | -0.1804 | -0.1080 | -0.2048 |
| Oct | 0.2274 | 0.0346 | -0.0306 | -0.1510 | -0.2981 | -0.3241 |
| Nov | 0.1888 | 0.0205 | -0.0407 | -0.1055 | -0.0903 | -0.2263 |
| Dec | 0.1090 | 0.0677 | -0.0105 | -0.0670 | -0.0786 | -0.0985 |

| BS-A/T-25 | LT=1 | LT=2 | LT=3 | LT=4 | LT=5 | LT=6 |
|---|---|---|---|---|---|---|
| Jan | 13.40 | 7.98 | 2.24 | 11.45 | 26.84 | 28.54 |
| Feb | 12.71 | 6.36 | 3.09 | 6.58 | 17.20 | 28.77 |
| Mar | 6.50 | 13.28 | 23.32 | 23.55 | 21.28 | 21.68 |
| Apr | 26.54 | 51.38 | 51.19 | 43.47 | 48.48 | 51.60 |
| May | 32.91 | 43.47 | 50.71 | 50.10 | 35.94 | 49.21 |
| Jun | 61.79 | 62.35 | 48.78 | 49.78 | 42.99 | 41.79 |
| Jul | 45.00 | 58.66 | 58.66 | 43.97 | 42.31 | 35.07 |
| Aug | 31.61 | 31.43 | 41.06 | 43.63 | 37.14 | 52.16 |
| Sep | 30.50 | 27.39 | 39.43 | 36.17 | 40.25 | 37.48 |
| Oct | 24.41 | 31.26 | 33.90 | 37.26 | 37.78 | 40.34 |
| Nov | 21.64 | 11.86 | 29.35 | 26.56 | 32.12 | 35.40 |
| Dec | 12.01 | 5.62 | 8.23 | 26.99 | 25.91 | 29.48 |

| BS-A/T-75 | LT=1 | LT=2 | LT=3 | LT=4 | LT=5 | LT=6 |
|---|---|---|---|---|---|---|
| Jan | 8.46 | 6.05 | 6.60 | 8.66 | 21.27 | 22.37 |
| Feb | 6.78 | 8.36 | 8.99 | 10.98 | 15.84 | 29.22 |
| Mar | 10.97 | 19.70 | 16.24 | 15.80 | 20.07 | 17.51 |
| Apr | 96.89 | 109.84 | 124.54 | 124.24 | 120.10 | 119.02 |
| May | 28.01 | 35.25 | 31.88 | 42.89 | 36.16 | 41.04 |
| Jun | 36.10 | 33.96 | 33.45 | 37.71 | 44.83 | 40.96 |
| Jul | 25.10 | 44.56 | 33.78 | 34.31 | 49.60 | 45.49 |
| Aug | 22.50 | 32.07 | 34.79 | 34.77 | 33.67 | 41.01 |
| Sep | 13.89 | 26.82 | 28.26 | 25.79 | 27.76 | 23.64 |
| Oct | 14.77 | 30.46 | 30.58 | 45.60 | 43.78 | 43.30 |
| Nov | 15.45 | 13.98 | 18.31 | 26.60 | 28.91 | 28.16 |
| Dec | 18.01 | 7.58 | 7.74 | 22.08 | 32.88 | 27.38 |

**Table 4: Skill scores in the Brahmaputra**





| BS-theo-25 | LT=1 | LT=2 | LT=3 | LT=4 | LT=5 | LT=6 |
|---|---|---|---|---|---|---|
| Jan | 0.0526 | 0.0932 | 0.0701 | 0.0852 | 0.0855 | 0.1720 |
| Feb | 0.0393 | 0.0876 | 0.0912 | 0.1150 | 0.1030 | 0.0647 |
| Mar | 0.0854 | 0.1060 | 0.1130 | 0.1260 | 0.1290 | 0.1700 |
| Apr | 0.0678 | 0.0997 | 0.0909 | 0.1120 | 0.1760 | 0.1650 |
| May | 0.1590 | 0.1900 | 0.1660 | 0.2010 | 0.1870 | 0.2030 |
| Jun | 0.2260 | 0.1770 | 0.1730 | 0.1940 | 0.1850 | 0.2000 |
| Jul | 0.1930 | 0.2040 | 0.1740 | 0.2310 | 0.2040 | 0.1810 |
| Aug | 0.0862 | 0.2370 | 0.2310 | 0.2260 | 0.2370 | 0.2980 |
| Sep | 0.0171 | 0.1430 | 0.2170 | 0.1820 | 0.1870 | 0.1980 |
| Oct | 0.0705 | 0.1150 | 0.1930 | 0.2170 | 0.1820 | 0.1620 |
| Nov | 0.0303 | 0.0639 | 0.1030 | 0.1630 | 0.2520 | 0.1950 |
| Dec | 0.0223 | 0.0477 | 0.0612 | 0.1360 | 0.1590 | 0.1720 |

| BSS-theo-25 | LT=1 | LT=2 | LT=3 | LT=4 | LT=5 | LT=6 |
|---|---|---|---|---|---|---|
| Jan | 0.3617 | 0.0431 | -0.0204 | 0.1235 | 0.3214 | -0.2836 |
| Feb | 0.2640 | -0.2807 | -0.1245 | -0.2459 | 0.0190 | 0.5243 |
| Mar | 0.1627 | 0.2090 | 0.1374 | 0.1656 | 0.1569 | 0.0395 |
| Apr | 0.2125 | -0.0767 | 0.2361 | 0.0427 | -0.2662 | -0.0313 |
| May | -0.2326 | -0.3669 | -0.1528 | -0.2331 | -0.1761 | -0.1154 |
| Jun | -1.1121 | -0.4274 | -0.0949 | -0.2763 | -0.1709 | -0.2579 |
| Jul | 0.0000 | -0.1148 | 0.2055 | -0.0794 | -0.0000 | 0.1298 |
| Aug | 0.4579 | -0.0822 | -0.0845 | -0.0610 | -0.1449 | -0.5051 |
| Sep | 0.7578 | 0.1386 | -0.1302 | 0.0900 | 0.0900 | 0.0050 |
| Oct | -0.0714 | 0.0417 | -0.1420 | -0.5390 | -0.1447 | 0.0182 |
| Nov | 0.3791 | 0.2338 | 0.0636 | 0.0181 | -0.4651 | -0.1337 |
| Dec | 0.2664 | 0.1343 | 0.2697 | -0.1333 | -0.0392 | -0.0238 |

| BS-theo-75 | LT=1 | LT=2 | LT=3 | LT=4 | LT=5 | LT=6 |
|---|---|---|---|---|---|---|
| Jan | 0.0909 | 0.0855 | 0.1070 | 0.1010 | 0.1460 | 0.1590 |
| Feb | 0.0376 | 0.0763 | 0.1180 | 0.1530 | 0.1240 | 0.1480 |
| Mar | 0.0424 | 0.1470 | 0.1180 | 0.1830 | 0.1990 | 0.1970 |
| Apr | 0.0234 | 0.1200 | 0.1040 | 0.1200 | 0.1410 | 0.1510 |
| May | 0.1170 | 0.1210 | 0.1750 | 0.1610 | 0.1630 | 0.1530 |
| Jun | 0.1650 | 0.1620 | 0.2100 | 0.1870 | 0.1630 | 0.1820 |
| Jul | 0.1420 | 0.1760 | 0.1980 | 0.2040 | 0.2210 | 0.1970 |
| Aug | 0.1860 | 0.1740 | 0.1640 | 0.2040 | 0.1930 | 0.1650 |
| Sep | 0.1260 | 0.1690 | 0.2190 | 0.1830 | 0.2210 | 0.1680 |
| Oct | 0.0802 | 0.1170 | 0.1890 | 0.1610 | 0.2030 | 0.2070 |
| Nov | 0.0623 | 0.1530 | 0.1650 | 0.1800 | 0.1830 | 0.2120 |
| Dec | 0.1000 | 0.0534 | 0.1240 | 0.1340 | 0.1230 | 0.1690 |

| BSS-theo-75 | LT=1 | LT=2 | LT=3 | LT=4 | LT=5 | LT=6 |
|---|---|---|---|---|---|---|
| Jan | 0.0055 | 0.3571 | -0.1823 | 0.3267 | 0.1043 | 0.0422 |
| Feb | 0.4069 | -0.0612 | 0.1449 | -0.0338 | 0.2530 | 0.2332 |
| Mar | 0.5081 | 0.0000 | 0.2761 | 0.0214 | 0.0829 | 0.0000 |
| Apr | 0.4834 | 0.0244 | 0.1938 | 0.1892 | 0.0784 | 0.0904 |
| May | -0.2829 | 0.1418 | -0.0938 | 0.0000 | 0.0632 | 0.0671 |
| Jun | -0.3095 | 0.0795 | -0.1538 | -0.0219 | 0.1093 | 0.0471 |
| Jul | -0.3028 | -0.1139 | 0.0246 | -0.0462 | -0.0676 | 0.0390 |
| Aug | -1.2545 | 0.0057 | 0.1368 | -0.0462 | 0.0203 | 0.1750 |
| Sep | -0.1455 | -0.0305 | -0.1587 | 0.0317 | -0.1755 | 0.1429 |
| Oct | -0.5453 | 0.0859 | -0.1962 | 0.1154 | -0.0856 | -0.0781 |
| Nov | 0.3250 | -0.0552 | 0.1176 | 0.0000 | -0.0227 | -0.1277 |
| Dec | -0.2674 | -0.4432 | -0.2731 | 0.0429 | 0.1689 | -0.0696 |

| BS-act-25 | LT=1 | LT=2 | LT=3 | LT=4 | LT=5 | LT=6 |
|---|---|---|---|---|---|---|
| Jan | 0.7250 | 0.7610 | 0.7570 | 0.7280 | 0.6470 | 0.6260 |
| Feb | 0.6600 | 0.6790 | 0.7240 | 0.6850 | 0.6980 | 0.6790 |
| Mar | 0.6340 | 0.6260 | 0.6440 | 0.6590 | 0.6680 | 0.7090 |
| Apr | 0.6280 | 0.6170 | 0.6240 | 0.6550 | 0.6650 | 0.6730 |
| May | 0.6220 | 0.6170 | 0.6420 | 0.5830 | 0.6690 | 0.6790 |
| Jun | 0.6350 | 0.5680 | 0.6790 | 0.7120 | 0.5420 | 0.6150 |
| Jul | 0.6680 | 0.5280 | 0.5110 | 0.6280 | 0.6800 | 0.6480 |
| Aug | 0.6230 | 0.5500 | 0.5850 | 0.5860 | 0.6590 | 0.6830 |
| Sep | 0.7290 | 0.6310 | 0.5770 | 0.5790 | 0.5720 | 0.6800 |
| Oct | 0.8310 | 0.6750 | 0.6200 | 0.5950 | 0.5730 | 0.5040 |
| Nov | 0.7180 | 0.7460 | 0.6630 | 0.6080 | 0.5820 | 0.5980 |
| Dec | 0.7340 | 0.7290 | 0.7490 | 0.6660 | 0.6410 | 0.5570 |

| BSS-act-25 | LT=1 | LT=2 | LT=3 | LT=4 | LT=5 | LT=6 |
|---|---|---|---|---|---|---|
| Jan | 0.0435 | -0.0242 | -0.0783 | -0.1629 | -0.0572 | -0.0868 |
| Feb | 0.0337 | 0.0382 | -0.1020 | -0.1211 | -0.1653 | -0.1431 |
| Mar | 0.0495 | 0.0501 | 0.0153 | -0.0965 | -0.1497 | -0.2593 |
| Apr | 0.0218 | 0.0045 | 0.0370 | -0.0301 | -0.1310 | -0.1869 |
| May | 0.0576 | -0.0115 | -0.0142 | 0.0395 | -0.1301 | -0.2715 |
| Jun | 0.0305 | 0.1139 | -0.2039 | -0.2297 | 0.0356 | -0.1431 |
| Jul | -0.0844 | 0.1316 | 0.1051 | -0.1254 | -0.2616 | -0.1613 |
| Aug | -0.0130 | 0.0401 | -0.0503 | -0.0335 | -0.1789 | -0.2004 |
| Sep | -0.1373 | -0.1664 | -0.0685 | -0.0284 | 0.0087 | -0.1467 |
| Oct | -0.2440 | -0.1421 | -0.2062 | 0.0100 | 0.0402 | 0.1897 |
| Nov | -0.0213 | -0.1477 | -0.1050 | -0.1450 | 0.0119 | -0.0017 |
| Dec | 0.0134 | -0.0596 | -0.1559 | -0.0990 | -0.0809 | 0.0732 |

| BS-act-75 | LT=1 | LT=2 | LT=3 | LT=4 | LT=5 | LT=6 |
|---|---|---|---|---|---|---|
| Jan | 0.7700 | 0.7310 | 0.6710 | 0.5970 | 0.5910 | 0.5600 |
| Feb | 0.6990 | 0.7130 | 0.6410 | 0.6200 | 0.6100 | 0.6280 |
| Mar | 0.7270 | 0.6710 | 0.6760 | 0.5810 | 0.5780 | 0.6260 |
| Apr | 0.7510 | 0.6470 | 0.6280 | 0.5560 | 0.5800 | 0.6220 |
| May | 0.7410 | 0.6420 | 0.6290 | 0.5970 | 0.5510 | 0.5480 |
| Jun | 0.5900 | 0.5620 | 0.6040 | 0.5380 | 0.6900 | 0.6420 |
| Jul | 0.6930 | 0.7090 | 0.6400 | 0.5610 | 0.5060 | 0.5640 |
| Aug | 0.8220 | 0.6220 | 0.6070 | 0.5070 | 0.5690 | 0.4920 |
| Sep | 0.8290 | 0.6410 | 0.5470 | 0.6200 | 0.5180 | 0.5440 |
| Oct | 0.8010 | 0.6500 | 0.6010 | 0.5490 | 0.5910 | 0.5160 |
| Nov | 0.7900 | 0.7100 | 0.6440 | 0.5310 | 0.5530 | 0.5940 |
| Dec | 0.8280 | 0.7380 | 0.6330 | 0.6130 | 0.5290 | 0.5390 |

| BSS-act-75 | LT=1 | LT=2 | LT=3 | LT=4 | LT=5 | LT=6 |
|---|---|---|---|---|---|---|
| Jan | -0.1920 | -0.0830 | 0.0469 | 0.0913 | 0.0678 | 0.0378 |
| Feb | -0.0770 | -0.1228 | -0.0372 | 0.0343 | 0.0333 | -0.0537 |
| Mar | -0.1431 | -0.1373 | -0.2093 | 0.0491 | 0.1094 | 0.0369 |
| Apr | -0.1680 | -0.0641 | -0.0698 | 0.0496 | 0.0445 | 0.0189 |
| May | -0.2412 | -0.1031 | -0.1113 | -0.0119 | 0.0937 | 0.0821 |
| Jun | -0.0034 | 0.0377 | -0.0962 | 0.0578 | -0.2321 | -0.0223 |
| Jul | -0.1177 | -0.1817 | -0.1208 | 0.0344 | 0.1044 | -0.0089 |
| Aug | -0.2342 | -0.1167 | -0.0782 | 0.1105 | 0.0018 | 0.1151 |
| Sep | -0.1113 | -0.0339 | -0.0148 | -0.1923 | 0.0650 | 0.0000 |
| Oct | -0.1476 | 0.0385 | -0.0309 | 0.0300 | -0.1067 | 0.0301 |
| Nov | -0.1174 | -0.0534 | -0.0016 | 0.0038 | -0.0184 | -0.0919 |
| Dec | -0.1421 | -0.0380 | 0.0452 | 0.0362 | 0.0537 | -0.0132 |

| BS-A/T-25 | LT=1 | LT=2 | LT=3 | LT=4 | LT=5 | LT=6 |
|---|---|---|---|---|---|---|
| Jan | 7.26 | 12.25 | 9.26 | 11.70 | 13.21 | 27.48 |
| Feb | 5.95 | 12.90 | 12.60 | 16.79 | 14.76 | 9.53 |
| Mar | 13.47 | 16.93 | 17.55 | 19.12 | 19.31 | 23.98 |
| Apr | 10.80 | 15.11 | 14.57 | 17.10 | 26.47 | 24.52 |
| May | 25.56 | 30.79 | 25.86 | 34.48 | 27.95 | 29.90 |
| Jun | 35.59 | 31.16 | 25.48 | 27.25 | 34.13 | 32.52 |
| Jul | 28.89 | 38.64 | 34.05 | 36.78 | 30.00 | 27.93 |
| Aug | 13.84 | 43.09 | 39.49 | 38.57 | 35.96 | 43.63 |
| Sep | 2.35 | 22.66 | 37.61 | 31.43 | 32.69 | 29.12 |
| Oct | 8.48 | 17.04 | 31.13 | 36.47 | 31.76 | 32.14 |
| Nov | 4.22 | 8.57 | 15.54 | 26.81 | 43.30 | 32.61 |
| Dec | 3.04 | 6.54 | 8.17 | 20.42 | 24.80 | 30.88 |

| BS-A/T-75 | LT=1 | LT=2 | LT=3 | LT=4 | LT=5 | LT=6 |
|---|---|---|---|---|---|---|
| Jan | 11.81 | 11.70 | 15.95 | 16.92 | 24.70 | 28.39 |
| Feb | 5.38 | 10.70 | 18.41 | 24.68 | 20.33 | 23.57 |
| Mar | 5.83 | 21.91 | 17.46 | 31.50 | 34.43 | 31.47 |
| Apr | 3.12 | 18.55 | 16.56 | 21.58 | 24.31 | 24.28 |
| May | 15.79 | 18.85 | 27.82 | 26.97 | 29.58 | 27.92 |
| Jun | 27.97 | 28.83 | 34.77 | 34.76 | 23.62 | 28.35 |
| Jul | 20.49 | 24.82 | 30.94 | 36.36 | 43.68 | 34.93 |
| Aug | 22.63 | 27.97 | 27.02 | 40.24 | 33.92 | 33.54 |
| Sep | 15.20 | 26.37 | 40.04 | 29.52 | 42.66 | 30.88 |
| Oct | 10.01 | 18.00 | 31.45 | 29.33 | 34.35 | 40.12 |
| Nov | 7.89 | 21.55 | 25.62 | 33.90 | 33.09 | 35.69 |
| Dec | 12.08 | 7.24 | 19.59 | 21.86 | 23.25 | 31.35 |

**Table 5: Skill scores in the Yangtze**



| BS-theo-25 | LT=1 | LT=2 | LT=3 | LT=4 | LT=5 | LT=6 |
|---|---|---|---|---|---|---|
| Jan | 0.0283 | 0.0214 | 0.0375 | 0.0576 | 0.1500 | 0.1920 |
| Feb | 0.0618 | 0.1100 | 0.0857 | 0.0840 | 0.1110 | 0.1470 |
| Mar | 0.0660 | 0.0778 | 0.0715 | 0.0749 | 0.0855 | 0.0746 |
| Apr | 0.0897 | 0.0760 | 0.0836 | 0.0731 | 0.0864 | 0.0958 |
| May | 0.1340 | 0.1470 | 0.1620 | 0.1580 | 0.1560 | 0.1280 |
| Jun | 0.1520 | 0.1610 | 0.1780 | 0.1730 | 0.1740 | 0.1830 |
| Jul | 0.1000 | 0.1560 | 0.2030 | 0.1980 | 0.1900 | 0.1830 |
| Aug | 0.1040 | 0.1930 | 0.2010 | 0.2040 | 0.1990 | 0.2050 |
| Sep | 0.0650 | 0.1620 | 0.1920 | 0.1920 | 0.2030 | 0.1950 |
| Oct | 0.0461 | 0.1360 | 0.1800 | 0.1980 | 0.1970 | 0.1930 |
| Nov | 0.0484 | 0.0965 | 0.1620 | 0.1890 | 0.2000 | 0.2010 |
| Dec | 0.0059 | 0.0344 | 0.0902 | 0.1680 | 0.1970 | 0.2030 |

| BSS-theo-25 | LT=1 | LT=2 | LT=3 | LT=4 | LT=5 | LT=6 |
|---|---|---|---|---|---|---|
| Jan | 0.7470 | 0.1215 | -0.0347 | -0.2569 | 0.1800 | 0.1042 |
| Feb | 0.4644 | -0.1000 | -0.5519 | -0.2857 | -0.0541 | -0.1020 |
| Mar | 0.0197 | -0.1941 | -0.2979 | -0.6956 | -0.2632 | -0.4343 |
| Apr | -0.5162 | -0.2355 | -0.0873 | -0.3817 | -0.5625 | -0.2630 |
| May | 0.0000 | -0.0544 | -0.0062 | 0.0759 | 0.1090 | 0.1042 |
| Jun | 0.0526 | -0.1925 | -0.0674 | -0.3468 | -0.2759 | -0.1421 |
| Jul | -0.2500 | -0.1154 | -0.0099 | 0.1717 | 0.0947 | -0.0109 |
| Aug | -0.4904 | 0.1088 | -0.0249 | 0.0392 | 0.0503 | -0.0341 |
| Sep | -0.0600 | 0.0617 | -0.0154 | 0.0260 | -0.0197 | -0.1641 |
| Oct | -0.8937 | 0.1912 | 0.0667 | -0.2727 | -0.1878 | -0.2124 |
| Nov | 0.2769 | 0.2580 | 0.1173 | 0.0899 | -0.2550 | -0.0647 |
| Dec | 0.9532 | 0.1599 | 0.0466 | 0.1429 | 0.0914 | -0.1232 |

| BS-theo-75 | LT=1 | LT=2 | LT=3 | LT=4 | LT=5 | LT=6 |
|---|---|---|---|---|---|---|
| Jan | 0.0135 | 0.0295 | 0.0543 | 0.0580 | 0.1070 | 0.1620 |
| Feb | 0.0709 | 0.1300 | 0.1470 | 0.1370 | 0.1430 | 0.1730 |
| Mar | 0.1260 | 0.1760 | 0.2000 | 0.1970 | 0.2080 | 0.1850 |
| Apr | 0.1450 | 0.1530 | 0.1800 | 0.1920 | 0.1910 | 0.2050 |
| May | 0.1670 | 0.1810 | 0.1780 | 0.1890 | 0.1910 | 0.1940 |
| Jun | 0.1150 | 0.1910 | 0.2080 | 0.2140 | 0.2000 | 0.2050 |
| Jul | 0.0939 | 0.1180 | 0.1520 | 0.1480 | 0.1550 | 0.1630 |
| Aug | 0.1050 | 0.1750 | 0.1810 | 0.1830 | 0.1810 | 0.1800 |
| Sep | 0.1060 | 0.1670 | 0.1860 | 0.1900 | 0.1910 | 0.1950 |
| Oct | 0.0683 | 0.1440 | 0.1710 | 0.1920 | 0.2000 | 0.1990 |
| Nov | 0.0753 | 0.1000 | 0.1360 | 0.1770 | 0.1930 | 0.2000 |
| Dec | 0.0350 | 0.0665 | 0.0698 | 0.1140 | 0.1450 | 0.1680 |

| BSS-theo-75 | LT=1 | LT=2 | LT=3 | LT=4 | LT=5 | LT=6 |
|---|---|---|---|---|---|---|
| Jan | -3.5111 | -0.7966 | -0.6059 | -0.4500 | 0.0561 | 0.0741 |
| Feb | -0.1100 | -0.0615 | -0.1361 | -0.2701 | -0.0420 | -0.1503 |
| Mar | 0.2960 | 0.0398 | -0.0100 | -0.0406 | 0.0962 | 0.1027 |
| Apr | -0.2069 | -0.2484 | 0.0444 | -0.0417 | 0.0209 | -0.0732 |
| May | 0.0958 | -0.0884 | 0.1124 | -0.0159 | -0.0419 | -0.2216 |
| Jun | -0.1913 | -0.0209 | 0.1202 | 0.0047 | -0.1450 | -0.1122 |
| Jul | -0.7891 | -0.0085 | 0.0724 | -0.7027 | -0.2581 | -0.3742 |
| Aug | -0.8286 | 0.2114 | -0.0939 | 0.0492 | -0.2376 | -0.1889 |
| Sep | 0.0000 | -0.1617 | 0.0376 | -0.0526 | -0.1361 | -0.3231 |
| Oct | -0.3075 | -0.3125 | -0.1345 | -0.0521 | 0.0250 | -0.0201 |
| Nov | -0.0505 | -0.1000 | -0.1324 | 0.0000 | -0.0052 | 0.0200 |
| Dec | -1.7000 | -0.3504 | -0.3338 | 0.0175 | -0.0828 | -0.1786 |

| BS-act-25 | LT=1 | LT=2 | LT=3 | LT=4 | LT=5 | LT=6 |
|---|---|---|---|---|---|---|
| Jan | 0.7400 | 0.7430 | 0.7500 | 0.6710 | 0.5590 | 0.6830 |
| Feb | 0.7290 | 0.7150 | 0.7190 | 0.7130 | 0.6280 | 0.5670 |
| Mar | 0.6770 | 0.6600 | 0.6630 | 0.6320 | 0.6690 | 0.6370 |
| Apr | 0.6450 | 0.6150 | 0.6390 | 0.6220 | 0.6580 | 0.6580 |
| May | 0.6970 | 0.5430 | 0.5990 | 0.6010 | 0.6290 | 0.6290 |
| Jun | 0.6320 | 0.6740 | 0.5810 | 0.5820 | 0.5290 | 0.5580 |
| Jul | 0.6130 | 0.5710 | 0.5960 | 0.6100 | 0.6660 | 0.5610 |
| Aug | 0.5390 | 0.5080 | 0.5480 | 0.6170 | 0.6950 | 0.6750 |
| Sep | 0.5420 | 0.5820 | 0.5920 | 0.5990 | 0.6680 | 0.7240 |
| Oct | 0.6210 | 0.5340 | 0.6350 | 0.6390 | 0.6390 | 0.6640 |
| Nov | 0.7500 | 0.6350 | 0.5190 | 0.6150 | 0.6390 | 0.6630 |
| Dec | 0.7340 | 0.7500 | 0.6560 | 0.5560 | 0.6510 | 0.6640 |

| BSS-act-25 | LT=1 | LT=2 | LT=3 | LT=4 | LT=5 | LT=6 |
|---|---|---|---|---|---|---|
| Jan | 0.0289 | -0.0234 | -0.0431 | -0.0788 | 0.0428 | -0.1518 |
| Feb | -0.0429 | -0.0317 | -0.1062 | -0.0363 | 0.0324 |
| Mar | 0.0203 | 0.0379 | -0.0541 | 0.0186 | -0.0437 | -0.0307 |
| Apr | 0.0168 | 0.0821 | 0.0274 | 0.0580 | -0.0682 | -0.0265 |
| May | -0.0823 | 0.0843 | 0.0370 | 0.0050 | -0.0431 | -0.1374 |
| Jun | 0.0841 | -0.1845 | -0.0544 | -0.0430 | 0.0131 | 0.0526 |
| Jul | 0.0569 | 0.0371 | -0.0241 | -0.0646 | -0.1914 | 0.0018 |
| Aug | 0.1135 | -0.0099 | 0.0552 | -0.1511 | -0.2478 | -0.1989 |
| Sep | 0.2312 | 0.0718 | -0.0225 | -0.0971 | -0.2748 | -0.3482 |
| Oct | 0.0577 | 0.1387 | -0.0708 | -0.1036 | -0.1275 | -0.1773 |
| Nov | 0.0040 | -0.0427 | 0.0019 | -0.0904 | -0.0047 | -0.0851 |
| Dec | 0.0068 | -0.0331 | -0.1007 | -0.0072 | -0.0868 | 0.0178 |

| BS-act-75 | LT=1 | LT=2 | LT=3 | LT=4 | LT=5 | LT=6 |
|---|---|---|---|---|---|---|
| Jan | 0.7940 | 0.7770 | 0.7990 | 0.6200 | 0.5620 | 0.4670 |
| Feb | 0.7810 | 0.7590 | 0.6560 | 0.7070 | 0.5690 | 0.5630 |
| Mar | 0.6670 | 0.5890 | 0.5990 | 0.6180 | 0.5830 | 0.5800 |
| Apr | 0.7290 | 0.5760 | 0.5890 | 0.5720 | 0.5510 | 0.5400 |
| May | 0.6570 | 0.6180 | 0.5740 | 0.5690 | 0.5830 | 0.5320 |
| Jun | 0.7250 | 0.6440 | 0.5950 | 0.5580 | 0.5270 | 0.5610 |
| Jul | 0.7200 | 0.6830 | 0.6260 | 0.5970 | 0.5280 | 0.4880 |
| Aug | 0.8220 | 0.6410 | 0.6410 | 0.5500 | 0.5150 | 0.4440 |
| Sep | 0.7580 | 0.6520 | 0.5450 | 0.5120 | 0.5540 | 0.4610 |
| Oct | 0.7530 | 0.6650 | 0.5550 | 0.5140 | 0.5250 | 0.4970 |
| Nov | 0.7790 | 0.6880 | 0.6360 | 0.5370 | 0.5070 | 0.4470 |
| Dec | 0.7940 | 0.7840 | 0.6360 | 0.5870 | 0.4810 | 0.4490 |

| BSS-act-75 | LT=1 | LT=2 | LT=3 | LT=4 | LT=5 | LT=6 |
|---|---|---|---|---|---|---|
| Jan | -0.1012 | -0.0444 | -0.0625 | 0.0549 | -0.0072 | 0.0985 |
| Feb | -0.1401 | -0.2974 | -0.0031 | -0.0910 | 0.0733 | 0.0679 |
| Mar | -0.1098 | -0.0034 | 0.0212 | -0.0198 | 0.0218 | 0.0317 |
| Apr | -0.1164 | 0.0319 | -0.0406 | 0.0222 | 0.0072 | 0.0146 |
| May | -0.0987 | -0.0748 | -0.0380 | -0.0089 | -0.0581 | 0.0617 |
| Jun | -0.0902 | -0.0751 | -0.0136 | 0.0036 | 0.1304 | 0.0311 |
| Jul | -0.1557 | -0.1616 | -0.0925 | 0.0083 | 0.0897 | 0.1812 |
| Aug | -0.0887 | -0.1870 | -0.1783 | -0.0110 | 0.0771 | 0.1808 |
| Sep | -0.2089 | -0.0252 | -0.0322 | 0.0192 | -0.0433 | 0.1447 |
| Oct | -0.2344 | -0.1896 | -0.0018 | 0.1199 | 0.0523 | 0.1401 |
| Nov | -0.1193 | -0.1187 | -0.1297 | 0.0693 | 0.0765 | 0.2102 |
| Dec | -0.1012 | -0.1058 | -0.0325 | -0.0501 | 0.1109 | 0.1851 |

| BS-A/T-25 | LT=1 | LT=2 | LT=3 | LT=4 | LT=5 | LT=6 |
|---|---|---|---|---|---|---|
| Jan | 3.82 | 2.88 | 5.00 | 8.58 | 26.83 | 28.11 |
| Feb | 8.48 | 15.38 | 11.92 | 11.78 | 17.68 | 25.93 |
| Mar | 9.75 | 11.79 | 10.78 | 11.85 | 12.78 | 11.71 |
| Apr | 13.91 | 12.36 | 13.08 | 11.75 | 13.13 | 15.45 |
| May | 19.23 | 27.07 | 27.05 | 26.29 | 24.80 | 20.35 |
| Jun | 24.05 | 23.89 | 30.64 | 29.73 | 32.89 | 32.80 |
| Jul | 16.31 | 27.32 | 34.06 | 32.46 | 28.53 | 32.62 |
| Aug | 19.29 | 37.99 | 36.68 | 33.06 | 28.63 | 30.37 |
| Sep | 11.99 | 27.84 | 32.94 | 32.05 | 30.39 | 26.93 |
| Oct | 7.42 | 25.47 | 28.35 | 30.99 | 31.37 | 29.07 |
| Nov | 6.45 | 15.20 | 31.21 | 30.73 | 31.30 | 30.32 |
| Dec | 0.80 | 4.59 | 13.75 | 30.22 | 30.26 | 30.57 |

| BS-A/T-75 | LT=1 | LT=2 | LT=3 | LT=4 | LT=5 | LT=6 |
|---|---|---|---|---|---|---|
| Jan | 1.70 | 3.80 | 6.80 | 9.35 | 19.04 | 34.69 |
| Feb | 9.08 | 17.13 | 22.41 | 19.38 | 25.13 | 30.73 |
| Mar | 18.89 | 29.88 | 33.39 | 31.88 | 35.68 | 31.90 |
| Apr | 19.89 | 26.56 | 30.56 | 33.57 | 34.66 | 37.96 |
| May | 25.42 | 29.29 | 31.01 | 33.22 | 32.76 | 36.47 |
| Jun | 15.86 | 29.66 | 34.96 | 38.35 | 37.95 | 36.54 |
| Jul | 13.04 | 17.28 | 24.28 | 24.79 | 29.36 | 33.40 |
| Aug | 12.77 | 27.30 | 28.24 | 33.27 | 35.15 | 40.54 |
| Sep | 13.98 | 25.61 | 34.13 | 37.11 | 34.48 | 42.30 |
| Oct | 9.07 | 21.65 | 30.81 | 37.35 | 38.10 | 40.04 |
| Nov | 9.67 | 14.53 | 21.38 | 32.96 | 38.07 | 44.74 |
| Dec | 4.41 | 8.48 | 10.97 | 19.42 | 30.15 | 37.42 |

**Table 6: Skill scores in the Yellow River**



| BS-theo-25 | LT=1 | LT=2 | LT=3 | LT=4 | LT=5 | LT=6 |
|---|---|---|---|---|---|---|
| Jan | 0.0678 | 0.0493 | 0.0977 | 0.0875 | 0.1050 | 0.1930 |
| Feb | 0.0875 | 0.0829 | 0.1040 | 0.1030 | 0.1580 | 0.1740 |
| Mar | 0.0887 | 0.0732 | 0.0722 | 0.0727 | 0.1100 | 0.1690 |
| Apr | 0.1280 | 0.0785 | 0.0975 | 0.1400 | 0.0958 | 0.1130 |
| May | 0.1800 | 0.1360 | 0.1600 | 0.1740 | 0.1540 | 0.1860 |
| Jun | 0.1870 | 0.1980 | 0.1690 | 0.2070 | 0.1050 | 0.1840 |
| Jul | 0.0609 | 0.2010 | 0.2580 | 0.1690 | 0.1980 | 0.1860 |
| Aug | 0.1660 | 0.1310 | 0.1820 | 0.2110 | 0.1540 | 0.2180 |
| Sep | 0.0959 | 0.1860 | 0.1660 | 0.2090 | 0.2190 | 0.1910 |
| Oct | 0.1650 | 0.1870 | 0.2290 | 0.2130 | 0.2060 | 0.2220 |
| Nov | 0.2020 | 0.1770 | 0.1710 | 0.2290 | 0.2190 | 0.2140 |
| Dec | 0.1040 | 0.1020 | 0.1550 | 0.1520 | 0.2030 | 0.2040 |

| BSS-theo-25 | LT=1 | LT=2 | LT=3 | LT=4 | LT=5 | LT=6 |
|---|---|---|---|---|---|---|
| Jan | -1.6077 | -0.0883 | -0.4326 | 0.1215 | 0.1102 | -0.2452 |
| Feb | -0.2982 | 0.1427 | -0.0833 | -0.0489 | 0.2020 | 0.0595 |
| Mar | -0.5214 | -0.2241 | 0.0177 | -0.1082 | -0.0185 | 0.1675 |
| Apr | -0.5459 | -0.0565 | -0.5452 | -0.6766 | -0.2127 | 0.0738 |
| May | -0.5000 | -0.0462 | -0.3913 | -0.4032 | -0.0845 | -0.3577 |
| Jun | -0.4841 | -0.1061 | 0.1378 | -0.0098 | 0.4670 | 0.0316 |
| Jul | 0.1577 | -0.4255 | -0.3651 | 0.1550 | 0.0481 | 0.1014 |
| Aug | -0.4821 | 0.2012 | -0.0520 | -0.0990 | 0.2261 | -0.0955 |
| Sep | -1.6713 | -0.6460 | -0.2117 | -0.2440 | -0.1838 | 0.0052 |
| Oct | 0.1033 | -0.0219 | -0.1804 | -0.0979 | 0.0000 | -0.0829 |
| Nov | -3.2797 | -0.1132 | 0.0552 | -0.1927 | -0.1406 | -0.0918 |
| Dec | -2.1231 | -0.6346 | -0.0473 | 0.1508 | -0.0973 | -0.1027 |

| BS-theo-75 | LT=1 | LT=2 | LT=3 | LT=4 | LT=5 | LT=6 |
|---|---|---|---|---|---|---|
| Jan | 0.0333 | 0.0348 | 0.1600 | 0.1420 | 0.1540 | 0.1540 |
| Feb | 0.0208 | 0.0124 | 0.0345 | 0.0932 | 0.1570 | 0.1400 |
| Mar | 0.0457 | 0.0681 | 0.0579 | 0.0735 | 0.0778 | 0.0838 |
| Apr | 0.1240 | 0.0928 | 0.0895 | 0.0683 | 0.0892 | 0.0664 |
| May | 0.0826 | 0.1360 | 0.1600 | 0.1940 | 0.1820 | 0.2280 |
| Jun | 0.1440 | 0.1720 | 0.1710 | 0.1550 | 0.1850 | 0.1740 |
| Jul | 0.1090 | 0.1240 | 0.2050 | 0.1810 | 0.1900 | 0.2360 |
| Aug | 0.1530 | 0.2010 | 0.1600 | 0.2080 | 0.1800 | 0.1980 |
| Sep | 0.1900 | 0.1840 | 0.2400 | 0.1890 | 0.2200 | 0.2070 |
| Oct | 0.1420 | 0.2080 | 0.1980 | 0.1820 | 0.1930 | 0.1900 |
| Nov | 0.0402 | 0.1950 | 0.1640 | 0.2070 | 0.1940 | 0.1780 |
| Dec | 0.0713 | 0.1390 | 0.2000 | 0.1590 | 0.1920 | 0.2070 |

| BSS-theo-75 | LT=1 | LT=2 | LT=3 | LT=4 | LT=5 | LT=6 |
|---|---|---|---|---|---|---|
| Jan | 0.2165 | 0.2912 | -0.1940 | 0.0000 | -0.1493 | 0.0191 |
| Feb | -7.6307 | -1.3529 | -1.2403 | 0.0624 | -0.3193 | -0.0606 |
| Mar | 0.1092 | -0.8760 | -0.5317 | -0.6406 | -0.1083 | -0.3538 |
| Apr | -1.4701 | -0.6601 | -0.7379 | -0.2532 | -0.8898 | 0.1669 |
| May | 0.4671 | 0.2093 | 0.0361 | 0.0321 | -0.1447 | -0.3029 |
| Jun | -0.9073 | 0.0115 | 0.0447 | 0.1667 | 0.0160 | 0.1429 |
| Jul | -1.4661 | -0.1071 | -0.0963 | 0.0321 | 0.0206 | -0.2041 |
| Aug | -0.4571 | -0.2968 | 0.1061 | -0.0505 | 0.0323 | -0.0154 |
| Sep | -0.6814 | -0.1018 | -0.3559 | 0.0053 | -0.1340 | -0.0561 |
| Oct | 0.1744 | 0.0189 | 0.2048 | 0.1765 | 0.1187 | 0.0777 |
| Nov | 0.4470 | -0.0209 | 0.1881 | -0.0561 | 0.0000 | 0.0825 |
| Dec | -2.6564 | -0.2087 | -0.1111 | 0.1762 | 0.0448 | -0.0615 |

| BS-act-25 | LT=1 | LT=2 | LT=3 | LT=4 | LT=5 | LT=6 |
|---|---|---|---|---|---|---|
| Jan | 0.4680 | 0.4940 | 0.5180 | 0.5080 | 0.4130 | 0.4440 |
| Feb | 0.5010 | 0.5100 | 0.5210 | 0.5080 | 0.5560 | 0.4780 |
| Mar | 0.5160 | 0.4430 | 0.4210 | 0.4710 | 0.4830 | 0.5230 |
| Apr | 0.5670 | 0.3970 | 0.4360 | 0.4090 | 0.4120 | 0.4540 |
| May | 0.5770 | 0.4540 | 0.4360 | 0.4690 | 0.4660 | 0.4090 |
| Jun | 0.3960 | 0.4010 | 0.4270 | 0.4170 | 0.4370 | 0.4480 |
| Jul | 0.5250 | 0.4100 | 0.4010 | 0.3420 | 0.3640 | 0.3430 |
| Aug | 0.5200 | 0.4550 | 0.4210 | 0.3900 | 0.3200 | 0.3970 |
| Sep | 0.5230 | 0.4430 | 0.3930 | 0.3630 | 0.4040 | 0.2910 |
| Oct | 0.5560 | 0.4560 | 0.4080 | 0.3860 | 0.3600 | 0.3520 |
| Nov | 0.5020 | 0.5260 | 0.3920 | 0.3660 | 0.3860 | 0.3630 |
| Dec | 0.5130 | 0.5230 | 0.5100 | 0.4150 | 0.4180 | 0.3760 |

| BSS-act-25 | LT=1 | LT=2 | LT=3 | LT=4 | LT=5 | LT=6 |
|---|---|---|---|---|---|---|
| Jan | 0.0106 | -0.0444 | -0.0860 | -0.0431 | 0.1554 | 0.1120 |
| Feb | 0.0618 | -0.0079 | -0.0097 | 0.0662 | -0.0091 | 0.1449 |
| Mar | -0.1467 | 0.0574 | 0.1137 | -0.0043 | 0.1089 | 0.0560 |
| Apr | -0.3279 | 0.0317 | -0.0660 | 0.0788 | 0.0615 | 0.0810 |
| May | -0.2794 | -0.0295 | -0.0307 | -0.1356 | -0.1394 | 0.0619 |
| Jun | 0.1852 | -0.0693 | -0.1509 | -0.1880 | -0.2275 | -0.2800 |
| Jul | -0.1218 | 0.0049 | -0.1457 | -0.0029 | -0.0866 | -0.0208 |
| Aug | -0.0078 | -0.1490 | -0.0712 | -0.0627 | 0.0831 | -0.1711 |
| Sep | 0.0132 | 0.0023 | -0.0565 | 0.0242 | -0.1348 | 0.1686 |
| Oct | -0.0923 | 0.0420 | 0.0355 | -0.0185 | 0.0955 | 0.1394 |
| Nov | -0.0819 | -0.2584 | 0.0485 | 0.0850 | -0.0185 | -0.0111 |
| Dec | 0.0375 | -0.2335 | -0.2143 | 0.0000 | 0.0369 | 0.0334 |

| BS-act-75 | LT=1 | LT=2 | LT=3 | LT=4 | LT=5 | LT=6 |
|---|---|---|---|---|---|---|
| Jan | 0.3000 | 0.2640 | 0.3140 | 0.2390 | 0.2510 | 0.2450 |
| Feb | 0.3150 | 0.3090 | 0.2950 | 0.3160 | 0.3290 | 0.2870 |
| Mar | 0.3180 | 0.3310 | 0.3610 | 0.3150 | 0.3250 | 0.3060 |
| Apr | 0.3790 | 0.3400 | 0.3400 | 0.3530 | 0.2620 | 0.3200 |
| May | 0.3190 | 0.2630 | 0.2570 | 0.2880 | 0.2610 | 0.2500 |
| Jun | 0.2110 | 0.2210 | 0.2010 | 0.2160 | 0.1970 | 0.2410 |
| Jul | 0.1940 | 0.2210 | 0.2170 | 0.1810 | 0.2020 | 0.2260 |
| Aug | 0.2930 | 0.2070 | 0.2630 | 0.2870 | 0.1800 | 0.1920 |
| Sep | 0.3350 | 0.2210 | 0.2340 | 0.2500 | 0.2200 | 0.1950 |
| Oct | 0.1850 | 0.2260 | 0.2590 | 0.2240 | 0.2540 | 0.2390 |
| Nov | 0.2340 | 0.2430 | 0.2130 | 0.2260 | 0.2310 | 0.2210 |
| Dec | 0.2710 | 0.2660 | 0.2240 | 0.2320 | 0.2530 | 0.2610 |

| BSS-act-75 | LT=1 | LT=2 | LT=3 | LT=4 | LT=5 | LT=6 |
|---|---|---|---|---|---|---|
| Jan | 0.0291 | 0.0186 | -0.1018 | 0.0440 | 0.0456 | -0.0560 |
| Feb | 0.0426 | 0.0693 | 0.0605 | -0.0194 | -0.0281 | 0.0712 |
| Mar | 0.0507 | 0.0265 | -0.0939 | -0.0161 | -0.0350 | 0.0377 |
| Apr | -0.0587 | 0.0365 | -0.0029 | -0.0660 | -0.0633 | -0.0356 |
| May | -0.2510 | 0.0113 | -0.0890 | -0.2203 | -0.1549 | -0.1161 |
| Jun | 0.2570 | -0.2067 | 0.0737 | -0.0964 | -0.0479 | -0.1531 |
| Jul | 0.2451 | -0.0780 | -0.1186 | 0.0905 | -0.0688 | -0.1531 |
| Aug | 0.0669 | -0.0299 | -0.2524 | -0.4000 | 0.1133 | 0.0400 |
| Sep | -0.1632 | 0.0349 | -0.2000 | -0.2376 | -0.0891 | 0.0152 |
| Oct | 0.2259 | 0.1439 | -0.1261 | -0.0874 | -0.1598 | -0.1602 |
| Nov | -0.1143 | -0.0848 | 0.1198 | -0.0415 | -0.1214 | -0.0524 |
| Dec | -0.1483 | -0.0598 | -0.0090 | 0.0000 | -0.1659 | -0.2794 |

| BS-A/T-25 | LT=1 | LT=2 | LT=3 | LT=4 | LT=5 | LT=6 |
|---|---|---|---|---|---|---|
| Jan | 14.49 | 9.98 | 18.86 | 17.22 | 25.42 | 43.47 |
| Feb | 17.47 | 16.25 | 19.96 | 20.28 | 28.42 | 36.40 |
| Mar | 17.19 | 16.52 | 17.15 | 15.44 | 22.77 | 32.31 |
| Apr | 22.57 | 19.77 | 22.36 | 34.23 | 23.25 | 24.89 |
| May | 31.20 | 29.96 | 36.70 | 37.10 | 33.05 | 45.48 |
| Jun | 47.22 | 49.38 | 39.58 | 49.64 | 24.03 | 41.07 |
| Jul | 11.60 | 40.12 | 64.34 | 49.42 | 54.40 | 54.23 |
| Aug | 31.92 | 28.79 | 43.23 | 54.10 | 48.13 | 54.91 |
| Sep | 18.34 | 41.99 | 42.24 | 57.58 | 54.21 | 65.64 |
| Oct | 29.68 | 41.01 | 56.13 | 55.18 | 57.22 | 63.07 |
| Nov | 40.24 | 33.65 | 43.62 | 62.57 | 56.74 | 58.95 |
| Dec | 20.27 | 19.50 | 30.39 | 36.63 | 48.56 | 54.26 |

| BS-A/T-75 | LT=1 | LT=2 | LT=3 | LT=4 | LT=5 | LT=6 |
|---|---|---|---|---|---|---|
| Jan | 11.10 | 13.18 | 50.96 | 59.41 | 61.35 | 62.86 |
| Feb | 6.60 | 4.01 | 11.69 | 29.49 | 47.72 | 48.78 |
| Mar | 14.37 | 20.57 | 16.04 | 23.33 | 23.94 | 27.39 |
| Apr | 32.72 | 29.27 | 26.32 | 19.35 | 34.05 | 20.75 |
| May | 25.89 | 51.71 | 62.26 | 67.36 | 69.73 | 91.20 |
| Jun | 68.25 | 68.53 | 85.07 | 71.76 | 93.91 | 72.20 |
| Jul | 56.19 | 56.11 | 94.47 | 100.00 | 94.06 | 104.42 |
| Aug | 52.22 | 97.10 | 60.84 | 72.47 | 100.00 | 103.13 |
| Sep | 56.72 | 83.26 | 102.56 | 75.60 | 100.00 | 106.15 |
| Oct | 76.76 | 92.04 | 76.45 | 81.25 | 75.98 | 79.50 |
| Nov | 17.18 | 80.25 | 77.00 | 91.59 | 83.98 | 80.54 |
| Dec | 26.31 | 52.26 | 89.29 | 68.53 | 75.89 | 79.31 |

**Table 7: Skill scores in the Mekong**





| BS-theo-25 | LT=1 | LT=2 | LT=3 | LT=4 | LT=5 | LT=6 |
|---|---|---|---|---|---|---|
| Jan | 0.0275 | 0.0356 | 0.0356 | 0.0416 | 0.0333 | 0.0977 |
| Feb | 0.0026 | 0.0000 | 0.0046 | 0.0011 | 0.0083 | 0.0305 |
| Mar | 0.0333 | 0.0180 | 0.0017 | 0.0154 | 0.0029 | 0.0174 |
| Apr | 0.0854 | 0.1010 | 0.1250 | 0.1280 | 0.1600 | 0.0983 |
| May | 0.1090 | 0.1420 | 0.1550 | 0.1770 | 0.1580 | 0.1210 |
| Jun | 0.0912 | 0.1160 | 0.1380 | 0.1100 | 0.0960 | 0.1440 |
| Jul | 0.0419 | 0.1290 | 0.1280 | 0.1310 | 0.1220 | 0.1100 |
| Aug | 0.0862 | 0.0937 | 0.1210 | 0.1070 | 0.1350 | 0.1690 |
| Sep | 0.0355 | 0.0851 | 0.1010 | 0.1800 | 0.1420 | 0.1840 |
| Oct | 0.0237 | 0.0259 | 0.0893 | 0.1200 | 0.1890 | 0.1880 |
| Nov | 0.0176 | 0.0028 | 0.0256 | 0.0835 | 0.1230 | 0.1870 |
| Dec | 0.0000 | 0.0000 | 0.0025 | 0.0256 | 0.0769 | 0.1310 |

| BSS-theo-25 | LT=1 | LT=2 | LT=3 | LT=4 | LT=5 | LT=6 |
|---|---|---|---|---|---|---|
| Jan | -1.6077 | -0.0883 | -0.4326 | 0.1215 | 0.1102 | -0.2452 |
| Feb | -0.2982 | 0.1427 | -0.0833 | -0.0489 | 0.2020 | 0.0595 |
| Mar | -0.5214 | -0.2241 | 0.0177 | -0.1082 | -0.0185 | 0.1675 |
| Apr | -0.5459 | -0.0565 | -0.5452 | -0.6766 | -0.2127 | 0.0738 |
| May | -0.5000 | -0.0462 | -0.3913 | -0.4032 | -0.0845 | -0.3577 |
| Jun | -0.4841 | -0.1061 | 0.1378 | -0.0098 | 0.4670 | 0.0316 |
| Jul | 0.1577 | -0.4255 | -0.3651 | 0.1550 | 0.0481 | 0.1014 |
| Aug | -0.4821 | 0.2012 | -0.0520 | -0.0990 | 0.2261 | -0.0955 |
| Sep | -1.6713 | -0.6460 | -0.2117 | -0.2440 | -0.1838 | 0.0052 |
| Oct | 0.1033 | -0.0219 | -0.1804 | -0.0979 | 0.0000 | -0.0829 |
| Nov | -3.2797 | -0.1132 | 0.0552 | -0.1927 | -0.1406 | -0.0918 |
| Dec | -2.1231 | -0.6346 | -0.0473 | 0.1508 | -0.0973 | -0.1027 |

| BS-theo-75 | LT=1 | LT=2 | LT=3 | LT=4 | LT=5 | LT=6 |
|---|---|---|---|---|---|---|
| Jan | 0.0333 | 0.0348 | 0.1600 | 0.1420 | 0.1540 | 0.1540 |
| Feb | 0.0208 | 0.0124 | 0.0345 | 0.0932 | 0.1570 | 0.1400 |
| Mar | 0.0457 | 0.0681 | 0.0579 | 0.0735 | 0.0778 | 0.0838 |
| Apr | 0.1240 | 0.0928 | 0.0895 | 0.0683 | 0.0892 | 0.0664 |
| May | 0.0826 | 0.1360 | 0.1600 | 0.1940 | 0.1820 | 0.2280 |
| Jun | 0.1440 | 0.1720 | 0.1710 | 0.1550 | 0.1850 | 0.1740 |
| Jul | 0.1090 | 0.1240 | 0.2050 | 0.1810 | 0.1900 | 0.2360 |
| Aug | 0.1530 | 0.2010 | 0.1600 | 0.2080 | 0.1800 | 0.1980 |
| Sep | 0.1900 | 0.1840 | 0.2400 | 0.1890 | 0.2200 | 0.2070 |
| Oct | 0.1420 | 0.2080 | 0.1980 | 0.1820 | 0.1930 | 0.1900 |
| Nov | 0.0402 | 0.1950 | 0.1640 | 0.2070 | 0.1940 | 0.1780 |
| Dec | 0.0713 | 0.1390 | 0.2000 | 0.1590 | 0.1920 | 0.2070 |

| BSS-theo-75 | LT=1 | LT=2 | LT=3 | LT=4 | LT=5 | LT=6 |
|---|---|---|---|---|---|---|
| Jan | 0.2165 | 0.2912 | -0.1940 | 0.0000 | -0.1493 | 0.0191 |
| Feb | -7.6307 | -1.3529 | -1.2403 | 0.0624 | -0.3193 | -0.0606 |
| Mar | 0.1092 | -0.8760 | -0.5317 | -0.6406 | -0.1083 | -0.3538 |
| Apr | -1.4701 | -0.6601 | -0.7379 | -0.2532 | -0.8898 | 0.1669 |
| May | 0.4671 | 0.2093 | 0.0361 | 0.0361 | -0.1447 | -0.3029 |
| Jun | -0.9073 | 0.0115 | 0.0447 | 0.1667 | 0.0160 | 0.1429 |
| Jul | -1.4661 | -0.1071 | -0.0963 | 0.0321 | 0.0206 | -0.2041 |
| Aug | -0.4571 | -0.2968 | 0.1061 | -0.0505 | 0.0323 | -0.0154 |
| Sep | -0.6814 | -0.1018 | -0.3559 | 0.0053 | -0.1340 | -0.0561 |
| Oct | 0.1744 | 0.0189 | 0.2048 | 0.1765 | 0.1187 | 0.0777 |
| Nov | 0.4470 | -0.0209 | 0.1881 | -0.0561 | 0.0000 | 0.0825 |
| Dec | -2.6564 | -0.2087 | -0.1111 | 0.1762 | 0.0448 | -0.0615 |

| BS-act-25 | LT=1 | LT=2 | LT=3 | LT=4 | LT=5 | LT=6 |
|---|---|---|---|---|---|---|
| Jan | 0.4680 | 0.4940 | 0.5180 | 0.5080 | 0.4130 | 0.4440 |
| Feb | 0.5010 | 0.5100 | 0.5210 | 0.5080 | 0.5560 | 0.4780 |
| Mar | 0.5160 | 0.4430 | 0.4210 | 0.4710 | 0.4830 | 0.5230 |
| Apr | 0.5670 | 0.3970 | 0.4360 | 0.4090 | 0.4120 | 0.4540 |
| May | 0.5770 | 0.4540 | 0.4360 | 0.4690 | 0.4660 | 0.4090 |
| Jun | 0.3960 | 0.4010 | 0.4270 | 0.4170 | 0.4370 | 0.4480 |
| Jul | 0.5250 | 0.4100 | 0.4010 | 0.3420 | 0.3640 | 0.3430 |
| Aug | 0.5200 | 0.4550 | 0.4210 | 0.3900 | 0.3200 | 0.3970 |
| Sep | 0.5230 | 0.4430 | 0.3930 | 0.3630 | 0.4040 | 0.2910 |
| Oct | 0.5560 | 0.4560 | 0.4080 | 0.3860 | 0.3600 | 0.3520 |
| Nov | 0.5020 | 0.5260 | 0.3920 | 0.3660 | 0.3860 | 0.3630 |
| Dec | 0.5130 | 0.5230 | 0.5100 | 0.4150 | 0.4180 | 0.3760 |

| BSS-act-25 | LT=1 | LT=2 | LT=3 | LT=4 | LT=5 | LT=6 |
|---|---|---|---|---|---|---|
| Jan | 0.0106 | -0.0444 | -0.0860 | -0.0431 | 0.1554 | 0.1120 |
| Feb | 0.0618 | -0.0079 | -0.0097 | 0.0560 | -0.0091 | 0.1449 |
| Mar | -0.1467 | 0.0574 | 0.1137 | -0.0043 | 0.1089 | 0.0560 |
| Apr | -0.3279 | 0.0317 | -0.0660 | 0.0788 | 0.0615 | 0.0810 |
| May | -0.2794 | -0.0295 | -0.0307 | -0.1356 | -0.1394 | 0.0619 |
| Jun | 0.1852 | -0.0693 | -0.1509 | -0.1880 | -0.2275 | -0.2800 |
| Jul | -0.1218 | 0.0049 | -0.1457 | -0.0029 | -0.0866 | -0.0208 |
| Aug | -0.0078 | -0.1490 | -0.0712 | -0.0627 | 0.0831 | -0.1711 |
| Sep | 0.0132 | 0.0023 | -0.0565 | 0.0242 | -0.1348 | 0.1686 |
| Oct | -0.0923 | 0.0420 | 0.0355 | -0.0185 | 0.0955 | 0.1394 |
| Nov | -0.0819 | -0.2584 | 0.0485 | 0.0850 | -0.0185 | -0.0111 |
| Dec | 0.0375 | -0.2335 | -0.2143 | 0.0000 | 0.0369 | 0.0334 |

| BS-act-75 | LT=1 | LT=2 | LT=3 | LT=4 | LT=5 | LT=6 |
|---|---|---|---|---|---|---|
| Jan | 0.3000 | 0.2640 | 0.3140 | 0.2390 | 0.2510 | 0.2450 |
| Feb | 0.3150 | 0.3090 | 0.2950 | 0.3160 | 0.3290 | 0.2870 |
| Mar | 0.3180 | 0.3310 | 0.3610 | 0.3150 | 0.3250 | 0.3060 |
| Apr | 0.3790 | 0.3400 | 0.3400 | 0.3530 | 0.2620 | 0.3200 |
| May | 0.3190 | 0.2630 | 0.2570 | 0.2880 | 0.2610 | 0.2500 |
| Jun | 0.2110 | 0.2570 | 0.2010 | 0.2160 | 0.1970 | 0.2410 |
| Jul | 0.1940 | 0.2210 | 0.2170 | 0.1810 | 0.2020 | 0.2260 |
| Aug | 0.2930 | 0.2070 | 0.2630 | 0.2870 | 0.1800 | 0.1920 |
| Sep | 0.3350 | 0.2210 | 0.2340 | 0.2500 | 0.2200 | 0.1950 |
| Oct | 0.1850 | 0.2260 | 0.2590 | 0.2240 | 0.2540 | 0.2390 |
| Nov | 0.2340 | 0.2430 | 0.2130 | 0.2260 | 0.2310 | 0.2210 |
| Dec | 0.2710 | 0.2660 | 0.2240 | 0.2320 | 0.2530 | 0.2610 |

| BSS-act-75 | LT=1 | LT=2 | LT=3 | LT=4 | LT=5 | LT=6 |
|---|---|---|---|---|---|---|
| Jan | 0.0291 | 0.0186 | -0.1018 | 0.0440 | 0.0456 | -0.0560 |
| Feb | 0.0426 | 0.0693 | 0.0605 | -0.0194 | -0.0281 | 0.0712 |
| Mar | 0.0507 | 0.0265 | -0.0939 | -0.0161 | -0.0350 | 0.0377 |
| Apr | -0.0587 | 0.0365 | -0.0029 | -0.0633 | 0.1988 | -0.0356 |
| May | -0.2510 | 0.0113 | -0.0890 | -0.2203 | -0.1549 | -0.1161 |
| Jun | 0.2570 | -0.2067 | 0.0737 | 0.1970 | -0.0964 | -0.1531 |
| Jul | 0.2451 | -0.0780 | -0.1186 | 0.0905 | -0.0688 | -0.1531 |
| Aug | 0.0669 | -0.0299 | -0.2524 | -0.4000 | 0.1133 | 0.0400 |
| Sep | -0.1632 | 0.0349 | -0.2000 | -0.2376 | -0.0891 | 0.0152 |
| Oct | 0.2259 | 0.1439 | -0.1261 | -0.0185 | -0.1598 | -0.1602 |
| Nov | -0.1143 | -0.0848 | 0.1198 | -0.0415 | -0.1214 | -0.0524 |
| Dec | -0.1483 | -0.0598 | -0.0090 | 0.0000 | -0.1659 | -0.2794 |

| BS-A/T-25 | LT=1 | LT=2 | LT=3 | LT=4 | LT=5 | LT=6 |
|---|---|---|---|---|---|---|
| Jan | 12.06 | 15.48 | 15.48 | 19.17 | 20.18 | 63.44 |
| Feb | 1.83 | 0.00 | 3.21 | 0.82 | 6.16 | 23.28 |
| Mar | 19.94 | 9.94 | 0.84 | 7.33 | 1.40 | 9.02 |
| Apr | 70.00 | 54.30 | 64.43 | 65.98 | 73.73 | 54.61 |
| May | 48.66 | 55.25 | 64.58 | 67.05 | 57.88 | 51.93 |
| Jun | 55.61 | 70.73 | 71.50 | 56.41 | 48.24 | 68.25 |
| Jul | 23.94 | 90.85 | 78.05 | 68.23 | 62.56 | 59.46 |
| Aug | 41.64 | 50.65 | 58.74 | 54.04 | 61.36 | 75.78 |
| Sep | 29.58 | 48.35 | 54.30 | 83.33 | 69.95 | 78.97 |
| Oct | 15.10 | 16.93 | 48.01 | 56.87 | 81.47 | 81.39 |
| Nov | 4.77 | 0.71 | 7.78 | 28.21 | 42.86 | 79.24 |
| Dec | 0.00 | 0.00 | 0.77 | 8.95 | 33.73 | 60.65 |

| BS-A/T-75 | LT=1 | LT=2 | LT=3 | LT=4 | LT=5 | LT=6 |
|---|---|---|---|---|---|---|
| Jan | 11.43 | 11.13 | 11.13 | 10.42 | 25.32 | 26.90 |
| Feb | 14.32 | 14.87 | 14.32 | 11.78 | 14.36 | 36.38 |
| Mar | 5.83 | 11.73 | 12.01 | 11.73 | 11.51 | 10.71 |
| Apr | 49.76 | 49.78 | 40.85 | 47.63 | 62.09 | 44.87 |
| May | 11.71 | 16.98 | 28.83 | 41.45 | 38.99 | 54.00 |
| Jun | 25.21 | 52.96 | 50.61 | 55.51 | 58.37 | 75.74 |
| Jul | 100.00 | 95.83 | 122.22 | 79.19 | 65.05 | 84.95 |
| Aug | 82.73 | 67.70 | 63.45 | 58.24 | 68.51 | 65.00 |
| Sep | 53.37 | 60.75 | 74.62 | 77.90 | 76.53 | 59.92 |
| Oct | 37.44 | 47.00 | 48.18 | 62.95 | 91.74 | 100.00 |
| Nov | 22.92 | 21.23 | 28.86 | 24.83 | 65.49 | 88.99 |
| Dec | 19.94 | 20.48 | 23.10 | 53.14 | 25.95 | 70.80 |

**Table 8: Skill scores in the Mckenzie**




| BS-theo-25 | LT=1 | LT=2 | LT=3 | LT=4 | LT=5 | LT=6 |
|---|---|---|---|---|---|---|
| Jan | 0.0402 | 0.0690 | 0.0559 | 0.0929 | 0.1010 | 0.0667 |
| Feb | 0.0693 | 0.0358 | 0.0633 | 0.0453 | 0.0792 | 0.0995 |
| Mar | 0.0047 | 0.0162 | 0.0300 | 0.0145 | 0.0103 | 0.0630 |
| Apr | 0.0603 | 0.0736 | 0.0693 | 0.1010 | 0.0561 | 0.0673 |
| May | 0.0391 | 0.0769 | 0.0961 | 0.0946 | 0.1090 | 0.0875 |
| Jun | 0.0113 | 0.1370 | 0.1030 | 0.1050 | 0.1310 | 0.0928 |
| Jul | 0.0545 | 0.0788 | 0.1090 | 0.1140 | 0.1170 | 0.1300 |
| Aug | 0.0388 | 0.1020 | 0.1610 | 0.1250 | 0.1150 | 0.1190 |
| Sep | 0.0231 | 0.0653 | 0.0873 | 0.0978 | 0.1140 | 0.1100 |
| Oct | 0.0567 | 0.0614 | 0.0694 | 0.0904 | 0.0893 | 0.1140 |
| Nov | 0.0336 | 0.0760 | 0.0931 | 0.1130 | 0.0771 | 0.0939 |
| Dec | 0.0636 | 0.0587 | 0.0945 | 0.0983 | 0.0711 | 0.0835 |

| BSS-theo-25 | LT=1 | LT=2 | LT=3 | LT=4 | LT=5 | LT=6 |
|---|---|---|---|---|---|---|
| Jan | 0.3973 | -0.0192 | -0.4045 | 0.0313 | -0.3377 | 0.2472 |
| Feb | -0.0390 | 0.2200 | -0.1943 | -0.5784 | -0.0272 | -0.1476 |
| Mar | 0.8058 | -0.3279 | -0.2195 | -0.6219 | -3.8357 | 0.0125 |
| Apr | -0.2996 | -0.0455 | -0.1017 | -0.6212 | -0.1022 | -0.1786 |
| May | 0.2053 | -0.0052 | 0.0351 | 0.0217 | 0.0763 | 0.0427 |
| Jun | 0.0504 | -0.4406 | -0.0521 | 0.0708 | -0.1391 | 0.2455 |
| Jul | 0.0778 | -0.1193 | 0.0684 | -0.1068 | -0.0086 | -0.1111 |
| Aug | 0.4648 | 0.0000 | -0.5333 | 0.0741 | 0.0086 | 0.0916 |
| Sep | 0.3265 | 0.2546 | 0.0623 | -0.0438 | 0.0880 | 0.0909 |
| Oct | -0.0366 | 0.0767 | -0.1433 | -0.1864 | 0.0335 | 0.1024 |
| Nov | 0.0693 | -0.0541 | 0.0643 | -0.3614 | -0.0874 | 0.1690 |
| Dec | -0.0478 | 0.1252 | -0.1331 | -0.5170 | 0.1828 | 0.1383 |

| BS-theo-75 | LT=1 | LT=2 | LT=3 | LT=4 | LT=5 | LT=6 |
|---|---|---|---|---|---|---|
| Jan | 0.0278 | 0.0165 | 0.0114 | 0.0653 | 0.1330 | 0.1710 |
| Feb | 0.0151 | 0.0080 | 0.0185 | 0.0254 | 0.0504 | 0.1130 |
| Mar | 0.0333 | 0.0182 | 0.0234 | 0.0368 | 0.0433 | 0.0493 |
| Apr | 0.0041 | 0.0185 | 0.0268 | 0.0399 | 0.0265 | 0.0345 |
| May | 0.0242 | 0.0474 | 0.0460 | 0.0567 | 0.0603 | 0.0863 |
| Jun | 0.0854 | 0.0678 | 0.1460 | 0.0931 | 0.1050 | 0.1430 |
| Jul | 0.0620 | 0.0882 | 0.0758 | 0.0879 | 0.0661 | 0.0698 |
| Aug | 0.0275 | 0.0961 | 0.1130 | 0.1120 | 0.1130 | 0.1490 |
| Sep | 0.0449 | 0.1300 | 0.1370 | 0.1720 | 0.1220 | 0.2010 |
| Oct | 0.0174 | 0.0807 | 0.1370 | 0.1360 | 0.1360 | 0.1270 |
| Nov | 0.0080 | 0.0826 | 0.1180 | 0.1390 | 0.1530 | 0.1310 |
| Dec | 0.0204 | 0.0259 | 0.0837 | 0.1420 | 0.1520 | 0.1230 |

| BSS-theo-75 | LT=1 | LT=2 | LT=3 | LT=4 | LT=5 | LT=6 |
|---|---|---|---|---|---|---|
| Jan | -800.4740 | -22.6717 | -0.7823 | 0.0394 | -0.0580 | -0.2040 |
| Feb | NA | 0.2138 | -4.4361 | -0.1222 | 0.2765 | 0.0454 |
| Mar | -0.7699 | -0.9017 | 0.1712 | -0.1191 | -0.5086 | 0.2822 |
| Apr | 0.8092 | 0.1257 | -0.1232 | -0.2237 | 0.2128 | 0.0913 |
| May | 0.3800 | -0.1073 | 0.3025 | 0.1203 | 0.2137 | 0.0025 |
| Jun | -2.6186 | -0.0311 | -0.3860 | 0.0501 | -0.0378 | -0.4114 |
| Jul | -0.1151 | -0.5154 | -0.3706 | -0.5316 | -0.0072 | -0.0657 |
| Aug | -0.2405 | -0.4933 | -0.5715 | -0.2548 | -0.1909 | -0.3639 |
| Sep | 0.2771 | -0.3357 | -0.2894 | -0.6193 | -0.0067 | -0.4556 |
| Oct | 0.4265 | 0.0447 | -0.4772 | -0.2457 | -0.2808 | -0.1428 |
| Nov | 0.7424 | -0.1650 | -0.2979 | -0.3228 | -0.0748 | -0.0407 |
| Dec | 0.2147 | 0.0017 | -0.0171 | -0.0996 | -0.1303 | 0.0509 |

| BS-act-25 | LT=1 | LT=2 | LT=3 | LT=4 | LT=5 | LT=6 |
|---|---|---|---|---|---|---|
| Jan | 0.1430 | 0.1380 | 0.1500 | 0.1990 | 0.2130 | 0.1670 |
| Feb | 0.1380 | 0.1510 | 0.1390 | 0.1520 | 0.2170 | 0.2120 |
| Mar | 0.1740 | 0.1600 | 0.1570 | 0.1430 | 0.1510 | 0.2040 |
| Apr | 0.2240 | 0.2010 | 0.1950 | 0.2280 | 0.2070 | 0.1860 |
| May | 0.1660 | 0.1070 | 0.1330 | 0.1320 | 0.1510 | 0.1380 |
| Jun | 0.2420 | 0.2160 | 0.1700 | 0.1780 | 0.1940 | 0.2140 |
| Jul | 0.2120 | 0.1940 | 0.1640 | 0.1450 | 0.1830 | 0.1370 |
| Aug | 0.2150 | 0.2590 | 0.2640 | 0.2150 | 0.1990 | 0.2160 |
| Sep | 0.1930 | 0.1800 | 0.1600 | 0.2010 | 0.1680 | 0.1710 |
| Oct | 0.3540 | 0.3040 | 0.2820 | 0.2660 | 0.2830 | 0.2840 |
| Nov | 0.2280 | 0.2700 | 0.2510 | 0.2290 | 0.2170 | 0.2390 |
| Dec | 0.2030 | 0.2280 | 0.2640 | 0.2740 | 0.2290 | 0.2110 |

| BSS-act-25 | LT=1 | LT=2 | LT=3 | LT=4 | LT=5 | LT=6 |
|---|---|---|---|---|---|---|
| Jan | -0.0752 | 0.0213 | -0.0067 | 0.0613 | -0.1390 | 0.0670 |
| Feb | -0.0376 | -0.0942 | 0.0915 | 0.0410 | 0.0356 | -0.0816 |
| Mar | -0.1373 | -0.0256 | -0.0329 | 0.1333 | 0.0736 | 0.0811 |
| Apr | -0.3254 | -0.0524 | -0.0156 | -0.2459 | -0.1829 | 0.0000 |
| May | -0.0375 | 0.1705 | 0.0148 | 0.0435 | 0.0258 | -0.1040 |
| Jun | -0.1101 | -0.2706 | 0.0173 | 0.0729 | 0.0251 | -0.0239 |
| Jul | -0.0242 | -0.0960 | -0.1549 | 0.0136 | -0.2282 | 0.1329 |
| Aug | 0.0928 | -0.1719 | -0.1379 | 0.0402 | 0.0701 | -0.0189 |
| Sep | 0.0302 | -0.0405 | 0.1534 | -0.0469 | 0.0919 | 0.0552 |
| Oct | -0.0085 | -0.0667 | 0.0000 | 0.0148 | 0.0407 | -0.0637 |
| Nov | -0.0704 | -0.0075 | -0.0502 | -0.0088 | 0.0441 | 0.0205 |
| Dec | -0.0684 | -0.3333 | -0.0732 | -0.2287 | -0.1450 | 0.0140 |

| BS-act-75 | LT=1 | LT=2 | LT=3 | LT=4 | LT=5 | LT=6 |
|---|---|---|---|---|---|---|
| Jan | 0.2280 | 0.2230 | 0.2180 | 0.2600 | 0.2140 | 0.3150 |
| Feb | 0.1440 | 0.1960 | 0.2190 | 0.1760 | 0.2070 | 0.2010 |
| Mar | 0.1670 | 0.1560 | 0.1570 | 0.1750 | 0.1810 | 0.1870 |
| Apr | 0.1370 | 0.1520 | 0.1650 | 0.1550 | 0.1580 | 0.1540 |
| May | 0.0970 | 0.1080 | 0.1190 | 0.1190 | 0.0967 | 0.1490 |
| Jun | 0.1460 | 0.1710 | 0.1700 | 0.1050 | 0.1680 | 0.1560 |
| Jul | 0.1230 | 0.1060 | 0.1240 | 0.1300 | 0.1330 | 0.1200 |
| Aug | 0.1670 | 0.1580 | 0.1740 | 0.1970 | 0.1430 | 0.2100 |
| Sep | 0.2090 | 0.1720 | 0.2280 | 0.2200 | 0.2310 | 0.2370 |
| Oct | 0.2900 | 0.2440 | 0.2210 | 0.2020 | 0.1910 | 0.2600 |
| Nov | 0.2440 | 0.2710 | 0.2760 | 0.2720 | 0.2130 | 0.2340 |
| Dec | 0.2870 | 0.2800 | 0.2660 | 0.2630 | 0.2670 | 0.2070 |

| BSS-act-75 | LT=1 | LT=2 | LT=3 | LT=4 | LT=5 | LT=6 |
|---|---|---|---|---|---|---|
| Jan | -0.1400 | -0.0721 | -0.0955 | -0.1556 | 0.1083 | -0.0900 |
| Feb | 0.2800 | 0.0485 | -0.0429 | -0.0476 | -0.0197 | 0.1184 |
| Mar | -0.0987 | -0.0909 | 0.0309 | -0.0234 | -0.0838 | 0.0966 |
| Apr | 0.0927 | -0.1014 | -0.2313 | -0.0915 | -0.0327 | -0.1000 |
| May | 0.1019 | 0.0357 | -0.1782 | -0.3951 | 0.1128 | -0.2957 |
| Jun | 0.1705 | -0.1958 | -0.4167 | 0.1176 | -0.4000 | -0.1232 |
| Jul | 0.2013 | 0.0364 | -0.1376 | -0.1017 | 0.0221 | -0.0169 |
| Aug | 0.1608 | 0.1604 | -0.0807 | -0.1867 | 0.1173 | -0.2883 |
| Sep | 0.2400 | 0.1810 | -0.1515 | -0.0476 | -0.0267 | -0.1023 |
| Oct | -0.0507 | 0.0758 | -0.0138 | 0.0982 | 0.1659 | -0.1304 |
| Nov | -0.0383 | -0.0226 | 0.0891 | -0.0584 | 0.1377 | 0.0965 |
| Dec | -0.0709 | -0.0980 | -0.0769 | 0.0075 | -0.0553 | 0.1653 |

| BS-A/T-25 | LT=1 | LT=2 | LT=3 | LT=4 | LT=5 | LT=6 |
|---|---|---|---|---|---|---|
| Jan | 28.11 | 50.00 | 37.27 | 46.68 | 47.42 | 39.94 |
| Feb | 50.22 | 23.71 | 45.54 | 29.80 | 36.50 | 46.93 |
| Mar | 2.69 | 10.13 | 19.11 | 10.14 | 6.82 | 30.88 |
| Apr | 26.92 | 36.62 | 35.54 | 44.30 | 27.10 | 36.18 |
| May | 23.55 | 71.87 | 72.26 | 71.67 | 72.19 | 63.41 |
| Jun | 4.67 | 63.43 | 60.59 | 58.99 | 67.53 | 43.36 |
| Jul | 25.71 | 40.62 | 66.46 | 78.62 | 63.93 | 94.89 |
| Aug | 18.05 | 39.38 | 60.98 | 58.14 | 57.79 | 55.09 |
| Sep | 11.97 | 36.28 | 54.56 | 48.66 | 67.86 | 64.33 |
| Oct | 16.02 | 20.20 | 24.61 | 33.98 | 31.55 | 40.14 |
| Nov | 14.74 | 28.15 | 37.09 | 49.34 | 35.53 | 39.29 |
| Dec | 31.33 | 25.75 | 35.80 | 35.88 | 31.05 | 39.57 |

| BS-A/T-75 | LT=1 | LT=2 | LT=3 | LT=4 | LT=5 | LT=6 |
|---|---|---|---|---|---|---|
| Jan | 12.19 | 7.40 | 5.23 | 25.12 | 62.15 | 54.29 |
| Feb | 10.49 | 4.08 | 8.45 | 14.43 | 24.35 | 56.22 |
| Mar | 19.94 | 11.67 | 14.90 | 21.03 | 23.92 | 26.36 |
| Apr | 3.01 | 12.17 | 16.24 | 25.74 | 16.77 | 22.40 |
| May | 24.95 | 43.89 | 38.66 | 47.65 | 62.36 | 57.92 |
| Jun | 58.49 | 39.65 | 85.88 | 88.67 | 62.50 | 92.26 |
| Jul | 50.41 | 83.21 | 61.13 | 67.62 | 49.70 | 58.17 |
| Aug | 16.47 | 61.21 | 64.94 | 56.85 | 79.02 | 70.95 |
| Sep | 21.48 | 75.58 | 60.09 | 78.18 | 52.81 | 84.81 |
| Oct | 6.00 | 33.07 | 61.99 | 67.33 | 71.20 | 48.85 |
| Nov | 3.27 | 30.48 | 42.75 | 51.10 | 71.83 | 55.98 |
| Dec | 7.11 | 9.25 | 31.47 | 53.99 | 56.93 | 59.42 |

**Table 9: Skill scores in the Nelson**





| BS-theo-25 | LT=1 | LT=2 | LT=3 | LT=4 | LT=5 | LT=6 |
|---|---|---|---|---|---|---|
| Jan | 0.0044 | 0.0291 | 0.0160 | 0.0684 | 0.0977 | 0.1130 |
| Feb | 0.0373 | 0.0094 | 0.0202 | 0.0134 | 0.0806 | 0.0875 |
| Mar | 0.0612 | 0.0359 | 0.0240 | 0.0342 | 0.0276 | 0.0918 |
| Apr | 0.0986 | 0.1630 | 0.1630 | 0.1740 | 0.2190 | 0.2020 |
| May | 0.0609 | 0.1080 | 0.1250 | 0.1750 | 0.1710 | 0.2120 |
| Jun | 0.0680 | 0.0829 | 0.1680 | 0.1790 | 0.1640 | 0.2420 |
| Jul | 0.1020 | 0.1080 | 0.1170 | 0.1790 | 0.1890 | 0.1780 |
| Aug | 0.1220 | 0.1320 | 0.1320 | 0.1250 | 0.2180 | 0.2130 |
| Sep | 0.0466 | 0.1310 | 0.1490 | 0.2130 | 0.1710 | 0.1810 |
| Oct | 0.0441 | 0.0901 | 0.1610 | 0.1640 | 0.2160 | 0.1520 |
| Nov | 0.0061 | 0.0948 | 0.1010 | 0.1560 | 0.1610 | 0.2050 |
| Dec | 0.0344 | 0.0146 | 0.1020 | 0.1160 | 0.1460 | 0.1680 |

| BSS-theo-25 | LT=1 | LT=2 | LT=3 | LT=4 | LT=5 | LT=6 |
|---|---|---|---|---|---|---|
| Jan | 0.8676 | -6.2388 | 0.2195 | 0.0976 | -0.0427 | 0.1630 |
| Feb | -0.6878 | 0.2192 | -0.2866 | 0.5231 | -0.1273 | 0.1026 |
| Mar | -0.4134 | -0.2128 | 0.0283 | -1.1242 | 0.0450 | -0.5377 |
| Apr | 0.3105 | 0.1376 | 0.1466 | 0.0595 | -0.3519 | -0.1676 |
| May | 0.0272 | 0.1220 | 0.1554 | -0.1824 | -0.0118 | -0.1648 |
| Jun | -6.4074 | 0.2598 | -0.1915 | -0.1118 | -0.0123 | -0.4938 |
| Jul | -1.2030 | -1.0690 | 0.0859 | -0.1258 | -0.0559 | -0.0114 |
| Aug | -1.5957 | -0.2816 | -0.1579 | 0.2188 | -0.2044 | -0.2312 |
| Sep | -0.0787 | -0.0397 | -0.0205 | -0.2529 | 0.0284 | 0.0321 |
| Oct | -0.2115 | 0.0415 | -0.0952 | -0.0123 | -0.2000 | 0.1602 |
| Nov | 0.6189 | -0.3640 | -0.1464 | -0.0612 | -0.0662 | -0.2059 |
| Dec | -0.0269 | 0.5829 | -0.3546 | -0.0545 | 0.0267 | 0.0000 |

| BS-theo-75 | LT=1 | LT=2 | LT=3 | LT=4 | LT=5 | LT=6 |
|---|---|---|---|---|---|---|
| Jan | 0.0000 | 0.0234 | 0.0755 | 0.1290 | 0.2030 | 0.1740 |
| Feb | 0.0345 | 0.0543 | 0.0732 | 0.1030 | 0.1780 | 0.2200 |
| Mar | 0.0289 | 0.0353 | 0.0438 | 0.0684 | 0.0923 | 0.1500 |
| Apr | 0.1080 | 0.2350 | 0.1980 | 0.1680 | 0.2220 | 0.2060 |
| May | 0.1290 | 0.1120 | 0.1410 | 0.1540 | 0.1290 | 0.1350 |
| Jun | 0.1210 | 0.1370 | 0.1140 | 0.1820 | 0.1400 | 0.1990 |
| Jul | 0.0959 | 0.1090 | 0.1050 | 0.1200 | 0.1680 | 0.1740 |
| Aug | 0.1290 | 0.1080 | 0.1640 | 0.1500 | 0.1440 | 0.2000 |
| Sep | 0.1450 | 0.0887 | 0.1560 | 0.1590 | 0.1990 | 0.1470 |
| Oct | 0.0953 | 0.1220 | 0.1450 | 0.1350 | 0.1680 | 0.1750 |
| Nov | 0.1280 | 0.1070 | 0.1850 | 0.1640 | 0.1850 | 0.2220 |
| Dec | 0.0099 | 0.0730 | 0.1300 | 0.2100 | 0.1440 | 0.1850 |

| BSS-theo-75 | LT=1 | LT=2 | LT=3 | LT=4 | LT=5 | LT=6 |
|---|---|---|---|---|---|---|
| Jan | NA | 0.2909 | 0.2525 | 0.0786 | -0.2012 | -0.0116 |
| Feb | 0.0227 | -0.0442 | 0.1766 | 0.2536 | -0.1410 | -0.2429 |
| Mar | -58.2213 | -0.0633 | -0.0868 | 0.0087 | 0.2308 | -0.1029 |
| Apr | 0.1496 | -0.2368 | -0.0102 | 0.2038 | -0.1156 | -0.0300 |
| May | -1.1287 | 0.1040 | -0.1371 | -0.1667 | 0.2412 | 0.2241 |
| Jun | -3.5660 | -0.1913 | 0.0579 | -0.2215 | 0.1617 | -0.1845 |
| Jul | -2.4873 | -0.6006 | 0.1250 | 0.1111 | -0.1507 | -0.1373 |
| Aug | -2.4309 | 0.0526 | -0.1549 | 0.0506 | 0.1000 | -0.1696 |
| Sep | -1.1137 | -0.2184 | -0.4312 | -0.3140 | -0.2675 | -0.0809 |
| Oct | -3.9895 | -0.0083 | -0.1328 | -0.1538 | -0.2353 | -0.0736 |
| Nov | -3.5552 | 0.0531 | -0.2759 | -0.0933 | -0.2759 | -0.3293 |
| Dec | -82.3613 | 0.1816 | 0.0226 | -0.2963 | 0.1628 | -0.1491 |

| BS-act-25 | LT=1 | LT=2 | LT=3 | LT=4 | LT=5 | LT=6 |
|---|---|---|---|---|---|---|
| Jan | 0.3800 | 0.4020 | 0.3510 | 0.3220 | 0.3700 | 0.3480 |
| Feb | 0.3760 | 0.3730 | 0.3930 | 0.3550 | 0.3470 | 0.3410 |
| Mar | 0.3940 | 0.3560 | 0.3510 | 0.3290 | 0.3540 | 0.3050 |
| Apr | 0.2870 | 0.2290 | 0.2570 | 0.2350 | 0.2540 | 0.2550 |
| May | 0.2970 | 0.2230 | 0.2100 | 0.2000 | 0.2310 | 0.2490 |
| Jun | 0.4200 | 0.3680 | 0.2410 | 0.2030 | 0.1760 | 0.1820 |
| Jul | 0.4170 | 0.3200 | 0.3170 | 0.2520 | 0.1890 | 0.1840 |
| Aug | 0.3760 | 0.3500 | 0.2410 | 0.2640 | 0.2420 | 0.2010 |
| Sep | 0.3800 | 0.2880 | 0.3010 | 0.2070 | 0.2680 | 0.2240 |
| Oct | 0.4080 | 0.3260 | 0.2640 | 0.2430 | 0.1980 | 0.2430 |
| Nov | 0.4550 | 0.3250 | 0.3070 | 0.2650 | 0.2330 | 0.2110 |
| Dec | 0.4340 | 0.4150 | 0.3020 | 0.3100 | 0.2910 | 0.2410 |

| BSS-act-25 | LT=1 | LT=2 | LT=3 | LT=4 | LT=5 | LT=6 |
|---|---|---|---|---|---|---|
| Jan | -0.0354 | -0.0836 | -0.0541 | -0.1459 | -0.4859 | -0.4561 |
| Feb | -0.0273 | -0.0219 | -0.0593 | 0.0706 | -0.2305 | -0.3586 |
| Mar | -0.0591 | -0.0471 | -0.0415 | 0.0706 | -0.0926 | -0.0410 |
| Apr | -0.3602 | -0.2793 | -0.4278 | -0.2304 | -0.2893 | -0.2562 |
| May | -0.1041 | -0.0985 | -0.2651 | -0.1299 | -0.2031 | -0.2388 |
| Jun | -0.1111 | -0.3237 | 0.0549 | -0.2229 | 0.0000 | 0.1034 |
| Jul | -0.0246 | -0.1034 | -0.2834 | -0.1351 | -0.1739 | -0.0952 |
| Aug | -0.3239 | -0.4170 | 0.0041 | -0.2110 | -0.1980 | -0.1618 |
| Sep | -0.0734 | -0.2522 | -0.3378 | 0.0591 | -0.2823 | -0.0566 |
| Oct | -0.1148 | -0.2587 | -0.1681 | 0.1111 | 0.0435 | -0.1912 |
| Nov | -0.1098 | -0.0726 | -0.1762 | -0.2500 | -0.0888 | -0.0193 |
| Dec | -0.0023 | -0.1067 | -0.0272 | -0.1742 | -0.3288 | -0.1262 |

| BS-act-75 | LT=1 | LT=2 | LT=3 | LT=4 | LT=5 | LT=6 |
|---|---|---|---|---|---|---|
| Jan | 0.2000 | 0.1610 | 0.1570 | 0.1470 | 0.2150 | 0.2170 |
| Feb | 0.3100 | 0.2360 | 0.2050 | 0.1530 | 0.1660 | 0.2270 |
| Mar | 0.3440 | 0.2800 | 0.2260 | 0.1840 | 0.2270 | 0.1720 |
| Apr | 0.1740 | 0.1990 | 0.2600 | 0.2100 | 0.2030 | 0.2620 |
| May | 0.2570 | 0.1120 | 0.1650 | 0.2040 | 0.2020 | 0.1910 |
| Jun | 0.2790 | 0.2710 | 0.2660 | 0.2730 | 0.1870 | 0.2410 |
| Jul | 0.2050 | 0.1640 | 0.1470 | 0.1990 | 0.2040 | 0.2250 |
| Aug | 0.1470 | 0.1500 | 0.1580 | 0.1260 | 0.1680 | 0.2060 |
| Sep | 0.1690 | 0.1370 | 0.1860 | 0.1900 | 0.2110 | 0.1710 |
| Oct | 0.1680 | 0.2070 | 0.1510 | 0.2010 | 0.1800 | 0.2230 |
| Nov | 0.1340 | 0.1070 | 0.1730 | 0.1760 | 0.2160 | 0.2280 |
| Dec | 0.2770 | 0.2370 | 0.1550 | 0.2220 | 0.2230 | 0.1790 |

| BSS-act-75 | LT=1 | LT=2 | LT=3 | LT=4 | LT=5 | LT=6 |
|---|---|---|---|---|---|---|
| Jan | 0.0000 | -0.0592 | -0.6286 | -0.2458 | -0.3961 | -0.4000 |
| Feb | -0.1111 | 0.0484 | -0.1141 | -0.2645 | -0.1528 | -0.4367 |
| Mar | -0.0488 | -0.0687 | 0.0583 | 0.0265 | -0.3275 | 0.0227 |
| Apr | -0.0175 | -0.1243 | -0.4525 | -0.1475 | -0.1404 | -0.3936 |
| May | -0.2297 | 0.3293 | 0.0833 | -0.0049 | 0.0648 | 0.0773 |
| Jun | -0.2455 | -0.2905 | 0.0112 | 0.0361 | 0.0361 | -0.3464 |
| Jul | -0.2733 | -0.4138 | -0.0138 | 0.0433 | -0.1397 | -0.2931 |
| Aug | 0.0577 | -0.0490 | -0.0128 | 0.1000 | -0.0120 | -0.2561 |
| Sep | 0.1378 | -0.0787 | -0.2157 | -0.0857 | -0.3439 | 0.0393 |
| Oct | -0.0307 | -0.1436 | 0.1703 | -0.1044 | -0.0112 | -0.1737 |
| Nov | -0.5896 | -0.0190 | -0.2815 | -0.1210 | -0.2706 | -0.2193 |
| Dec | -0.0492 | -0.4192 | 0.1969 | -0.0936 | -0.2458 | -0.0113 |

| BS-A/T-25 | LT=1 | LT=2 | LT=3 | LT=4 | LT=5 | LT=6 |
|---|---|---|---|---|---|---|
| Jan | 1.16 | 7.24 | 4.56 | 21.24 | 26.41 | 32.47 |
| Feb | 9.92 | 2.51 | 5.14 | 3.77 | 23.23 | 25.66 |
| Mar | 15.53 | 10.08 | 6.84 | 10.40 | 7.80 | 30.10 |
| Apr | 34.36 | 71.18 | 63.42 | 74.04 | 86.22 | 79.22 |
| May | 20.51 | 48.43 | 59.52 | 87.50 | 74.03 | 85.14 |
| Jun | 16.19 | 22.53 | 69.71 | 88.18 | 93.18 | 132.97 |
| Jul | 24.46 | 33.75 | 36.91 | 71.03 | 100.00 | 96.74 |
| Aug | 32.45 | 37.71 | 54.77 | 47.35 | 90.08 | 105.97 |
| Sep | 12.26 | 45.49 | 49.50 | 102.90 | 83.68 | 80.80 |
| Oct | 10.81 | 27.64 | 60.98 | 67.49 | 109.09 | 62.55 |
| Nov | 1.33 | 29.17 | 32.90 | 58.87 | 69.10 | 97.16 |
| Dec | 7.93 | 3.52 | 33.77 | 37.42 | 50.17 | 69.71 |

| BS-A/T-75 | LT=1 | LT=2 | LT=3 | LT=4 | LT=5 | LT=6 |
|---|---|---|---|---|---|---|
| Jan | 0.00 | 14.53 | 48.09 | 87.76 | 94.42 | 80.18 |
| Feb | 11.13 | 23.01 | 35.71 | 67.32 | 107.23 | 96.92 |
| Mar | 8.40 | 12.61 | 19.38 | 37.17 | 40.66 | 87.21 |
| Apr | 62.07 | 118.09 | 76.15 | 80.00 | 109.36 | 78.63 |
| May | 50.19 | 100.00 | 85.45 | 75.49 | 63.86 | 70.68 |
| Jun | 43.37 | 50.55 | 42.86 | 66.67 | 74.87 | 82.57 |
| Jul | 46.78 | 66.46 | 71.43 | 60.30 | 82.35 | 77.33 |
| Aug | 87.76 | 72.00 | 103.80 | 119.05 | 85.71 | 97.09 |
| Sep | 85.80 | 64.74 | 83.87 | 83.68 | 94.31 | 85.96 |
| Oct | 56.73 | 58.94 | 96.03 | 67.16 | 93.33 | 78.48 |
| Nov | 95.52 | 100.00 | 106.94 | 93.18 | 85.65 | 97.37 |
| Dec | 3.58 | 30.80 | 83.87 | 94.59 | 64.57 | 103.35 |

**Table 10: Skill scores in the Ob**





| BS-theo-25 | LT=1 | LT=2 | LT=3 | LT=4 | LT=5 | LT=6 |
|---|---|---|---|---|---|---|
| Jan | 0.0333 | 0.0231 | 0.0285 | 0.0345 | 0.0388 | 0.1250 |
| Feb | 0.0000 | 0.0003 | 0.0006 | 0.0006 | 0.0348 | 0.0524 |
| Mar | 0.0011 | 0.0037 | 0.0080 | 0.0037 | 0.0017 | 0.0296 |
| Apr | 0.0284 | 0.0537 | 0.0590 | 0.0565 | 0.0524 | 0.0533 |
| May | 0.1690 | 0.1970 | 0.2100 | 0.1720 | 0.2080 | 0.1700 |
| Jun | 0.1300 | 0.1180 | 0.1470 | 0.1490 | 0.1780 | 0.1760 |
| Jul | 0.1020 | 0.2220 | 0.1770 | 0.1850 | 0.1740 | 0.1930 |
| Aug | 0.1250 | 0.1470 | 0.2120 | 0.1690 | 0.1700 | 0.1620 |
| Sep | 0.1030 | 0.1350 | 0.1600 | 0.1800 | 0.1890 | 0.1660 |
| Oct | 0.0620 | 0.0606 | 0.1110 | 0.1460 | 0.1740 | 0.1990 |
| Nov | 0.0667 | 0.0579 | 0.0865 | 0.1320 | 0.1680 | 0.1750 |
| Dec | 0.0003 | 0.0344 | 0.0609 | 0.0311 | 0.1540 | 0.1800 |

| BSS-theo-25 | LT=1 | LT=2 | LT=3 | LT=4 | LT=5 | LT=6 |
|---|---|---|---|---|---|---|
| Jan | 0.0000 | 0.3304 | 0.1739 | -0.2321 | 0.4663 | -0.0331 |
| Feb | NA | 0.9449 | 0.9034 | 0.9714 | -1.0964 | 0.3886 |
| Mar | 0.4712 | -2.0833 | -1.8134 | -2.4579 | 0.7759 | -0.9603 |
| Apr | 0.2427 | -0.1258 | -0.1895 | -0.1460 | -0.0459 | -0.0451 |
| May | -0.1986 | -0.3046 | -0.3548 | -0.0886 | -0.4345 | -0.2230 |
| Jun | -1.3636 | -0.1569 | -0.0809 | -0.1732 | -0.3588 | -0.1812 |
| Jul | 0.0097 | -0.3373 | -0.1203 | -0.1783 | -0.1299 | -0.2781 |
| Aug | -0.1161 | 0.1788 | -0.2471 | 0.0452 | 0.0286 | 0.0847 |
| Sep | 0.0000 | 0.1000 | 0.0805 | 0.0110 | -0.0442 | 0.0929 |
| Oct | -0.3995 | 0.3341 | 0.1328 | 0.0581 | -0.0419 | -0.1775 |
| Nov | -2.5479 | -0.4733 | 0.0335 | 0.0222 | -0.0701 | -0.0606 |
| Dec | -2.4678 | 0.2182 | -0.5575 | 0.6013 | -0.1407 | -0.1180 |

| BS-theo-75 | LT=1 | LT=2 | LT=3 | LT=4 | LT=5 | LT=6 |
|---|---|---|---|---|---|---|
| Jan | 0.0300 | 0.0180 | 0.0382 | 0.0499 | 0.0855 | 0.1430 |
| Feb | 0.0140 | 0.0140 | 0.0390 | 0.0425 | 0.0427 | 0.1040 |
| Mar | 0.0069 | 0.0011 | 0.0072 | 0.0142 | 0.0074 | 0.0755 |
| Apr | 0.0433 | 0.0868 | 0.1060 | 0.1030 | 0.1410 | 0.1060 |
| May | 0.0562 | 0.0986 | 0.1360 | 0.0923 | 0.0661 | 0.0769 |
| Jun | 0.1770 | 0.1640 | 0.1590 | 0.2060 | 0.1780 | 0.1750 |
| Jul | 0.1360 | 0.2430 | 0.2400 | 0.2090 | 0.1980 | 0.1890 |
| Aug | 0.1400 | 0.2010 | 0.2140 | 0.1490 | 0.1970 | 0.2140 |
| Sep | 0.1090 | 0.1730 | 0.1910 | 0.2040 | 0.1830 | 0.2090 |
| Oct | 0.0391 | 0.1400 | 0.1740 | 0.1750 | 0.1810 | 0.1500 |
| Nov | 0.0160 | 0.0581 | 0.0727 | 0.1400 | 0.1700 | 0.2200 |
| Dec | 0.0410 | 0.0540 | 0.0504 | 0.0959 | 0.1620 | 0.1840 |

| BSS-theo-75 | LT=1 | LT=2 | LT=3 | LT=4 | LT=5 | LT=6 |
|---|---|---|---|---|---|---|
| Jan | -0.2821 | 0.5477 | -0.0106 | -0.0991 | 0.2934 | 0.0714 |
| Feb | 0.2265 | 0.5989 | -0.0456 | -0.2040 | 0.0469 | 0.2180 |
| Mar | 0.1729 | 0.9208 | 0.4272 | 0.0405 | 0.4072 | 0.1076 |
| Apr | 0.3489 | 0.2180 | 0.0275 | 0.0283 | -0.2589 | 0.0619 |
| May | 0.2922 | 0.1351 | -0.2252 | 0.1125 | 0.4626 | 0.2386 |
| Jun | -2.3396 | 0.0520 | -0.0461 | -0.1444 | -0.0349 | -0.0479 |
| Jul | -0.1333 | -0.3352 | -0.2973 | 0.0203 | -0.0703 | -0.0500 |
| Aug | -0.0606 | -0.1486 | -0.1694 | 0.1722 | -0.1322 | -0.1758 |
| Sep | -0.0481 | 0.0335 | -0.0324 | -0.1087 | 0.0000 | -0.1421 |
| Oct | -3.6000 | 0.0411 | 0.0000 | 0.0223 | -0.0056 | 0.1667 |
| Nov | -244.7757 | 0.0836 | 0.3206 | 0.0968 | 0.0710 | -0.2500 |
| Dec | -9.5670 | -0.5836 | -0.0020 | 0.2735 | -0.0125 | -0.1572 |

| BS-act-25 | LT=1 | LT=2 | LT=3 | LT=4 | LT=5 | LT=6 |
|---|---|---|---|---|---|---|
| Jan | 0.5000 | 0.4710 | 0.4770 | 0.4830 | 0.4430 | 0.3410 |
| Feb | 0.4830 | 0.4610 | 0.4360 | 0.4430 | 0.4770 | 0.4130 |
| Mar | 0.4560 | 0.4550 | 0.4320 | 0.4210 | 0.4370 | 0.4650 |
| Apr | 0.4650 | 0.4110 | 0.4290 | 0.3900 | 0.3940 | 0.3890 |
| May | 0.2660 | 0.2820 | 0.2890 | 0.3260 | 0.2630 | 0.2770 |
| Jun | 0.3900 | 0.3120 | 0.2560 | 0.2340 | 0.2530 | 0.2430 |
| Jul | 0.3690 | 0.2770 | 0.2380 | 0.2640 | 0.2830 | 0.2370 |
| Aug | 0.3970 | 0.2500 | 0.2660 | 0.2420 | 0.2000 | 0.2530 |
| Sep | 0.4060 | 0.2990 | 0.2270 | 0.2160 | 0.2320 | 0.2200 |
| Oct | 0.4620 | 0.3760 | 0.2930 | 0.2730 | 0.2410 | 0.2530 |
| Nov | 0.4670 | 0.4580 | 0.4500 | 0.3320 | 0.3080 | 0.2840 |
| Dec | 0.4670 | 0.4890 | 0.4670 | 0.4490 | 0.3480 | 0.3010 |

| BSS-act-25 | LT=1 | LT=2 | LT=3 | LT=4 | LT=5 | LT=6 |
|---|---|---|---|---|---|---|
| Jan | 0.0000 | 0.0248 | 0.0124 | -0.0233 | -0.1845 | -0.2222 |
| Feb | -0.0343 | 0.0233 | 0.0396 | 0.0432 | -0.0767 | -0.1223 |
| Mar | -0.0088 | 0.0000 | 0.0400 | 0.0366 | -0.0069 | -0.0941 |
| Apr | -0.0839 | 0.0024 | -0.0567 | 0.0299 | -0.0181 | -0.0026 |
| May | 0.0221 | -0.1235 | -0.1333 | -0.2443 | -0.1050 | -0.1399 |
| Jun | -0.4234 | -0.0722 | 0.1172 | 0.1333 | 0.0996 | 0.1413 |
| Jul | -0.4819 | -0.1352 | -0.0485 | -0.1379 | -0.2748 | -0.0128 |
| Aug | -0.5269 | -0.1737 | -0.2430 | -0.1635 | 0.0431 | -0.1991 |
| Sep | -0.2378 | -0.3172 | -0.0758 | -0.0237 | -0.1154 | -0.1055 |
| Oct | -0.1186 | -0.3381 | -0.2851 | -0.1421 | -0.1767 | -0.1878 |
| Nov | -0.1146 | -0.0827 | -0.3514 | -0.2969 | -0.3391 | -0.2241 |
| Dec | -0.0065 | -0.0516 | -0.0424 | -0.2612 | -0.3182 | -0.2438 |

| BS-act-75 | LT=1 | LT=2 | LT=3 | LT=4 | LT=5 | LT=6 |
|---|---|---|---|---|---|---|
| Jan | 0.4180 | 0.4440 | 0.4780 | 0.4630 | 0.4190 | 0.3110 |
| Feb | 0.3770 | 0.3680 | 0.3770 | 0.3840 | 0.3960 | 0.2950 |
| Mar | 0.4550 | 0.3780 | 0.3600 | 0.4060 | 0.3980 | 0.4070 |
| Apr | 0.2620 | 0.2380 | 0.2450 | 0.2490 | 0.2300 | 0.2460 |
| May | 0.2310 | 0.2180 | 0.2320 | 0.2600 | 0.2430 | 0.2540 |
| Jun | 0.1540 | 0.2250 | 0.1920 | 0.2330 | 0.1860 | 0.1670 |
| Jul | 0.1890 | 0.2370 | 0.2200 | 0.2340 | 0.2750 | 0.2260 |
| Aug | 0.2180 | 0.2170 | 0.2400 | 0.2530 | 0.2010 | 0.2510 |
| Sep | 0.3540 | 0.2370 | 0.2350 | 0.2420 | 0.2480 | 0.2600 |
| Oct | 0.4160 | 0.3910 | 0.3210 | 0.2730 | 0.3150 | 0.3410 |
| Nov | 0.3800 | 0.3330 | 0.3230 | 0.2840 | 0.2650 | 0.2870 |
| Dec | -0.0160 | -0.0215 | -0.0660 | -0.0717 | -0.1778 | -0.2588 |

| BSS-act-75 | LT=1 | LT=2 | LT=3 | LT=4 | LT=5 | LT=6 |
|---|---|---|---|---|---|---|
| Jan | -0.1196 | -0.0475 | -0.0758 | 0.0113 | -0.0811 | 0.0000 |
| Feb | 0.0302 | -0.0374 | -0.0717 | -0.0358 | -0.0244 | 0.0064 |
| Mar | -0.0053 | 0.0081 | -0.0134 | -0.0026 | -0.0259 | 0.0781 |
| Apr | -0.0911 | 0.0207 | 0.0674 | 0.0149 | -0.0049 | |
| May | -0.1855 | -0.0128 | -0.0746 | -0.1018 | 0.0000 | -0.0123 |
| Jun | 0.1923 | 0.0046 | -0.1208 | -0.0429 | -0.1092 | |
| Jul | -0.0769 | -0.2097 | -0.0492 | -0.2873 | -0.0054 | 0.0722 |
| Aug | -0.1053 | -0.2216 | -0.0377 | -0.1304 | -0.3285 | -0.0813 |
| Sep | 0.0840 | -0.0155 | -0.1881 | -0.1878 | 0.0243 | -0.2067 |
| Oct | -0.0960 | -0.1286 | -0.1576 | -0.1421 | -0.1754 | -0.2808 |
| Nov | -0.0400 | -0.0130 | -0.0808 | -0.1328 | -0.3291 | -0.4328 |
| Dec | -0.0160 | -0.0215 | -0.0660 | -0.0717 | -0.1778 | -0.2588 |

| BS-A/T-25 | LT=1 | LT=2 | LT=3 | LT=4 | LT=5 | LT=6 |
|---|---|---|---|---|---|---|
| Jan | 6.66 | 4.90 | 5.97 | 7.14 | 8.76 | 36.66 |
| Feb | 0.00 | 0.06 | 0.13 | 0.13 | 7.30 | 12.69 |
| Mar | 0.24 | 0.81 | 1.85 | 0.88 | 0.39 | 6.37 |
| Apr | 6.11 | 13.07 | 13.75 | 14.49 | 13.30 | 13.70 |
| May | 63.53 | 69.86 | 72.66 | 52.76 | 79.09 | 61.37 |
| Jun | 33.33 | 37.82 | 57.42 | 63.68 | 70.36 | 72.43 |
| Jul | 27.64 | 80.14 | 74.37 | 70.08 | 61.48 | 81.43 |
| Aug | 31.49 | 58.80 | 79.70 | 69.83 | 85.00 | 64.03 |
| Sep | 25.37 | 45.15 | 70.48 | 83.33 | 81.47 | 75.45 |
| Oct | 13.42 | 16.12 | 37.88 | 53.48 | 72.20 | 78.66 |
| Nov | 14.28 | 12.64 | 19.22 | 39.76 | 54.55 | 61.62 |
| Dec | 0.06 | 7.03 | 13.04 | 6.93 | 44.25 | 59.80 |

| BS-A/T-75 | LT=1 | LT=2 | LT=3 | LT=4 | LT=5 | LT=6 |
|---|---|---|---|---|---|---|
| Jan | 7.18 | 4.05 | 7.99 | 10.78 | 20.41 | 45.98 |
| Feb | 3.71 | 3.80 | 10.34 | 11.07 | 10.78 | 35.25 |
| Mar | 1.51 | 0.30 | 1.99 | 3.50 | 1.86 | 18.55 |
| Apr | 16.53 | 36.47 | 43.27 | 41.37 | 61.30 | 43.09 |
| May | 24.33 | 45.23 | 58.62 | 35.50 | 27.20 | 30.28 |
| Jun | 114.94 | 72.89 | 82.81 | 88.41 | 95.70 | 104.79 |
| Jul | 71.96 | 102.53 | 109.09 | 89.32 | 72.00 | 83.63 |
| Aug | 64.22 | 102.03 | 89.17 | 58.89 | 98.01 | 85.26 |
| Sep | 30.79 | 73.00 | 81.28 | 84.30 | 73.79 | 80.38 |
| Oct | 9.40 | 35.81 | 54.21 | 64.10 | 57.46 | 43.99 |
| Nov | 4.21 | 17.45 | 22.51 | 49.30 | 64.15 | 76.66 |
| Dec | -255.57 | -251.49 | -76.36 | -133.76 | -91.13 | -71.11 |

Table 11: Skill scores in the Lena





| BS-theo-25 | LT=1 | LT=2 | LT=3 | LT=4 | LT=5 | LT=6 |
|---|---|---|---|---|---|---|
| Jan | 0.1180 | 0.1520 | 0.1990 | 0.2370 | 0.2070 | 0.2010 |
| Feb | 0.1580 | 0.2120 | 0.1470 | 0.2480 | 0.2080 | 0.1920 |
| Mar | 0.1110 | 0.1780 | 0.2180 | 0.2100 | 0.2290 | 0.2470 |
| Apr | 0.0471 | 0.1890 | 0.1670 | 0.2340 | 0.2320 | 0.1910 |
| May | 0.0835 | 0.0672 | 0.1400 | 0.2040 | 0.2210 | 0.2020 |
| Jun | 0.1110 | 0.1400 | 0.1690 | 0.1850 | 0.2020 | 0.2420 |
| Jul | 0.1090 | 0.1530 | 0.1510 | 0.2180 | 0.2110 | 0.2400 |
| Aug | 0.1290 | 0.1840 | 0.1210 | 0.2010 | 0.1640 | 0.2010 |
| Sep | 0.1040 | 0.1500 | 0.1920 | 0.1540 | 0.2130 | 0.2090 |
| Oct | 0.0581 | 0.1160 | 0.1720 | 0.2090 | 0.1690 | 0.2040 |
| Nov | 0.0997 | 0.1580 | 0.1730 | 0.2330 | 0.2150 | 0.1760 |
| Dec | 0.0992 | 0.1190 | 0.2080 | 0.2040 | 0.1960 | 0.2230 |

| BSS-theo-25 | LT=1 | LT=2 | LT=3 | LT=4 | LT=5 | LT=6 |
|---|---|---|---|---|---|---|
| Jan | 0.2716 | 0.2245 | -0.0102 | -0.1618 | -0.0670 | -0.0308 |
| Feb | -0.2540 | 0.0047 | 0.2222 | -0.2525 | -0.0146 | 0.0400 |
| Mar | -0.0571 | -0.0230 | -0.1913 | -0.1413 | -0.1927 | -0.3722 |
| Apr | 0.5337 | -0.1183 | 0.0457 | -0.1878 | -0.1485 | 0.0354 |
| May | -0.2575 | 0.3720 | 0.1411 | -0.2671 | -0.1392 | 0.0000 |
| Jun | 0.1395 | 0.0476 | -0.3307 | -0.0511 | -0.0860 | -0.2347 |
| Jul | 0.1138 | 0.1774 | 0.1564 | -0.4342 | -0.2197 | -0.2632 |
| Aug | 0.1835 | -0.1288 | 0.0163 | -0.2036 | -0.1233 | -0.1754 |
| Sep | 0.1811 | 0.0506 | -0.0909 | 0.0191 | -0.2384 | -0.2081 |
| Oct | 0.3108 | -0.0357 | -0.0058 | -0.0503 | 0.0663 | -0.0625 |
| Nov | 0.1692 | -0.0897 | 0.0798 | -0.3239 | -0.1559 | 0.1020 |
| Dec | 0.0040 | 0.1678 | -0.2308 | -0.1724 | -0.0595 | -0.1378 |

| BS-theo-75 | LT=1 | LT=2 | LT=3 | LT=4 | LT=5 | LT=6 |
|---|---|---|---|---|---|---|
| Jan | 0.1230 | 0.1160 | 0.1160 | 0.2240 | 0.1840 | 0.2230 |
| Feb | 0.1230 | 0.2600 | 0.2330 | 0.2540 | 0.1850 | 0.1890 |
| Mar | 0.1090 | 0.1310 | 0.2320 | 0.2160 | 0.1580 | 0.1820 |
| Apr | 0.0388 | 0.1390 | 0.1360 | 0.2170 | 0.1970 | 0.1770 |
| May | 0.0950 | 0.1500 | 0.2050 | 0.1940 | 0.2070 | 0.2150 |
| Jun | 0.0893 | 0.1710 | 0.1710 | 0.2100 | 0.2690 | 0.2310 |
| Jul | 0.0937 | 0.1710 | 0.1620 | 0.1800 | 0.1780 | 0.1860 |
| Aug | 0.1700 | 0.2090 | 0.2390 | 0.2180 | 0.1720 | 0.1910 |
| Sep | 0.1730 | 0.1640 | 0.2090 | 0.2340 | 0.2580 | 0.2180 |
| Oct | 0.1440 | 0.2060 | 0.1650 | 0.1660 | 0.2350 | 0.1970 |
| Nov | 0.0961 | 0.1630 | 0.1890 | 0.1910 | 0.2080 | 0.2670 |
| Dec | 0.1510 | 0.1650 | 0.1990 | 0.1910 | 0.2310 | 0.2070 |

| BSS-theo-75 | LT=1 | LT=2 | LT=3 | LT=4 | LT=5 | LT=6 |
|---|---|---|---|---|---|---|
| Jan | -0.0982 | 0.3012 | 0.2418 | -0.1546 | 0.0213 | -0.1495 |
| Feb | 0.1277 | -0.3131 | -0.1888 | -0.3804 | -0.0054 | 0.0308 |
| Mar | -0.0283 | 0.1914 | -0.3410 | -0.2273 | 0.0760 | -0.0225 |
| Apr | 0.3701 | -0.0373 | 0.1282 | -0.1361 | -0.1067 | -0.0412 |
| May | -0.8340 | -0.1719 | -0.2893 | -0.0778 | -0.0615 | -0.1140 |
| Jun | -0.6945 | -0.1958 | 0.0284 | -0.2805 | -0.5028 | -0.1786 |
| Jul | 0.1160 | -0.1477 | -0.0658 | -0.2329 | -0.0988 | -0.0751 |
| Aug | -0.4655 | -0.1237 | -0.0482 | 0.0354 | 0.1313 | -0.0611 |
| Sep | -0.1234 | 0.0296 | -0.0609 | -0.1471 | -0.2344 | -0.1010 |
| Oct | 0.0000 | -0.1319 | -0.0248 | 0.1399 | -0.2051 | -0.0103 |
| Nov | -0.4649 | -0.0724 | -0.0272 | -0.0852 | -0.0348 | -0.4053 |
| Dec | -0.1705 | -0.3306 | -0.0529 | 0.0255 | -0.2094 | -0.0455 |

| BS-act-25 | LT=1 | LT=2 | LT=3 | LT=4 | LT=5 | LT=6 |
|---|---|---|---|---|---|---|
| Jan | 0.1790 | 0.1580 | 0.1990 | 0.2300 | 0.1880 | 0.2270 |
| Feb | 0.1890 | 0.2490 | 0.1910 | 0.2420 | 0.2200 | 0.2040 |
| Mar | 0.2020 | 0.2280 | 0.2060 | 0.1790 | 0.2230 | 0.2410 |
| Apr | 0.0592 | 0.1830 | 0.1790 | 0.2340 | 0.2260 | 0.2030 |
| May | 0.0835 | 0.1100 | 0.1700 | 0.2540 | 0.2270 | 0.2270 |
| Jun | 0.1720 | 0.1460 | 0.2180 | 0.2220 | 0.2210 | 0.2790 |
| Jul | 0.1450 | 0.2020 | 0.1870 | 0.2300 | 0.2170 | 0.2020 |
| Aug | 0.0989 | 0.1780 | 0.1520 | 0.1650 | 0.1890 | 0.2130 |
| Sep | 0.1400 | 0.1020 | 0.1560 | 0.1480 | 0.1640 | 0.2150 |
| Oct | 0.1310 | 0.1880 | 0.1840 | 0.1970 | 0.1870 | 0.1860 |
| Nov | 0.1480 | 0.1640 | 0.1790 | 0.2330 | 0.2030 | 0.1700 |
| Dec | 0.1900 | 0.1130 | 0.2260 | 0.2220 | 0.1780 | 0.2100 |

| BSS-act-25 | LT=1 | LT=2 | LT=3 | LT=4 | LT=5 | LT=6 |
|---|---|---|---|---|---|---|
| Jan | 0.1476 | 0.1939 | -0.0642 | -0.1558 | 0.0309 | -0.1350 |
| Feb | -0.1524 | 0.0040 | 0.0255 | -0.2737 | -0.1702 | -0.0303 |
| Mar | 0.0049 | 0.0215 | -0.0842 | 0.0867 | -0.2186 | -0.3242 |
| Apr | 0.4038 | -0.0223 | 0.0479 | -0.1642 | -0.1078 | -0.0253 |
| May | -0.0583 | -0.0280 | -0.0059 | -0.3804 | -0.1701 | -0.1350 |
| Jun | -0.1026 | 0.1609 | -0.8319 | -0.2333 | -0.1632 | -0.3881 |
| Jul | -0.2185 | -0.0745 | 0.0459 | -0.3939 | -0.2690 | -0.1222 |
| Aug | 0.1758 | -0.0920 | 0.0256 | 0.0237 | -0.1317 | -0.1094 |
| Sep | 0.1566 | 0.2444 | 0.0000 | 0.0133 | 0.0120 | -0.2573 |
| Oct | -0.0077 | -0.1394 | -0.0337 | -0.0479 | -0.1265 | 0.0106 |
| Nov | -0.1385 | -0.0250 | 0.1050 | -0.2073 | -0.0684 | 0.1626 |
| Dec | -0.2418 | 0.3110 | -0.1244 | -0.0725 | -0.0114 | -0.0825 |

| BS-act-75 | LT=1 | LT=2 | LT=3 | LT=4 | LT=5 | LT=6 |
|---|---|---|---|---|---|---|
| Jan | 0.1110 | 0.1850 | 0.1660 | 0.2050 | 0.1970 | 0.2350 |
| Feb | 0.1540 | 0.2360 | 0.2080 | 0.2410 | 0.1790 | 0.2070 |
| Mar | 0.1570 | 0.1590 | 0.2380 | 0.2290 | 0.1840 | 0.1880 |
| Apr | 0.1840 | 0.2790 | 0.1920 | 0.2290 | 0.2070 | 0.1680 |
| May | 0.1250 | 0.1860 | 0.1990 | 0.2220 | 0.2070 | 0.2270 |
| Jun | 0.0893 | 0.1710 | 0.1710 | 0.2100 | 0.2690 | 0.2310 |
| Jul | 0.1480 | 0.2190 | 0.2040 | 0.1980 | 0.2080 | 0.1980 |
| Aug | 0.1340 | 0.2010 | 0.2270 | 0.2000 | 0.1900 | 0.2030 |
| Sep | 0.2210 | 0.2010 | 0.1790 | 0.2150 | 0.2210 | 0.2420 |
| Oct | 0.1680 | 0.2010 | 0.1830 | 0.1780 | 0.2170 | 0.1850 |
| Nov | 0.1260 | 0.2540 | 0.2010 | 0.2210 | 0.2440 | 0.2790 |
| Dec | 0.1450 | 0.2080 | 0.1870 | 0.2030 | 0.2310 | 0.2010 |

| BSS-act-75 | LT=1 | LT=2 | LT=3 | LT=4 | LT=5 | LT=6 |
|---|---|---|---|---|---|---|
| Jan | 0.1395 | 0.1629 | 0.1123 | 0.0238 | 0.0248 | -0.1809 |
| Feb | -0.0199 | -0.3409 | -0.1183 | -0.3931 | 0.0000 | -0.0615 |
| Mar | -0.2171 | 0.1675 | -0.3077 | -0.3011 | -0.0760 | -0.0562 |
| Apr | -0.0455 | -0.2798 | 0.0495 | -0.0457 | 0.0000 | 0.1600 |
| May | -0.0593 | -0.0877 | -0.1706 | -0.2198 | -0.0615 | -0.1582 |
| Jun | -0.6945 | -0.1958 | 0.0284 | -0.2805 | -0.5028 | -0.1786 |
| Jul | 0.1084 | -0.0186 | -0.1657 | -0.3200 | -0.2840 | -0.0645 |
| Aug | -0.3814 | -0.2156 | -0.0509 | 0.0654 | -0.0215 | -0.2012 |
| Sep | -0.2346 | -0.0469 | -0.0170 | -0.1749 | -0.1882 | -0.2941 |
| Oct | 0.0667 | -0.1444 | -0.0055 | -0.1333 | -0.1128 | 0.0513 |
| Nov | 0.1544 | -0.2212 | -0.0691 | -0.1333 | -0.1564 | -0.3349 |
| Dec | 0.2120 | -0.2093 | 0.0000 | -0.0856 | -0.2554 | -0.0152 |

| BS-A/T-25 | LT=1 | LT=2 | LT=3 | LT=4 | LT=5 | LT=6 |
|---|---|---|---|---|---|---|
| Jan | 65.92 | 96.20 | 100.00 | 103.04 | 110.11 | 88.55 |
| Feb | 83.60 | 85.14 | 76.96 | 102.48 | 94.55 | 94.12 |
| Mar | 54.95 | 78.07 | 105.83 | 117.32 | 102.69 | 102.49 |
| Apr | 79.56 | 103.28 | 93.30 | 100.00 | 102.65 | 94.09 |
| May | 100.00 | 61.09 | 82.35 | 80.31 | 97.36 | 88.99 |
| Jun | 64.53 | 95.89 | 77.52 | 83.33 | 91.40 | 86.74 |
| Jul | 75.17 | 75.74 | 80.75 | 94.78 | 97.24 | 118.81 |
| Aug | 130.43 | 103.37 | 79.61 | 121.82 | 86.77 | 94.37 |
| Sep | 74.29 | 147.06 | 123.08 | 104.05 | 129.88 | 97.21 |
| Oct | 44.35 | 61.70 | 93.48 | 106.09 | 90.37 | 109.68 |
| Nov | 67.36 | 96.34 | 96.65 | 100.00 | 105.91 | 103.53 |
| Dec | 52.21 | 105.31 | 92.04 | 91.89 | 110.11 | 106.19 |

| BS-A/T-75 | LT=1 | LT=2 | LT=3 | LT=4 | LT=5 | LT=6 |
|---|---|---|---|---|---|---|
| Jan | 110.81 | 62.70 | 69.88 | 109.27 | 93.40 | 94.89 |
| Feb | 79.87 | 110.17 | 112.02 | 105.39 | 103.35 | 91.30 |
| Mar | 69.43 | 82.39 | 97.48 | 94.32 | 85.87 | 96.81 |
| Apr | 21.09 | 49.82 | 70.83 | 94.76 | 95.17 | 105.36 |
| May | 76.00 | 80.65 | 103.02 | 87.39 | 100.00 | 94.71 |
| Jun | 100.00 | 100.00 | 100.00 | 100.00 | 100.00 | 100.00 |
| Jul | 63.31 | 78.08 | 79.41 | 90.91 | 88.58 | 93.94 |
| Aug | 126.87 | 102.96 | 105.29 | 109.00 | 90.53 | 94.09 |
| Sep | 78.28 | 81.59 | 116.76 | 108.84 | 116.74 | 90.08 |
| Oct | 85.71 | 100.00 | 90.16 | 93.26 | 108.29 | 106.49 |
| Nov | 76.27 | 64.17 | 94.03 | 86.43 | 85.25 | 95.70 |
| Dec | 104.14 | 79.33 | 106.42 | 94.09 | 100.00 | 102.99 |

**Table 12: Skill scores in the Rhine**



| BS-theo-25 | LT=1 | LT=2 | LT=3 | LT=4 | LT=5 | LT=6 |
|---|---|---|---|---|---|---|
| Jan | 0.0303 | 0.0892 | 0.1480 | 0.2030 | 0.2260 | 0.1970 |
| Feb | 0.0484 | 0.1700 | 0.1550 | 0.1900 | 0.1900 | 0.1860 |
| Mar | 0.0171 | 0.1170 | 0.1630 | 0.1610 | 0.1970 | 0.2100 |
| Apr | 0.0251 | 0.1070 | 0.1590 | 0.2000 | 0.2430 | 0.2120 |
| May | 0.0727 | 0.1640 | 0.1800 | 0.1950 | 0.2120 | 0.2050 |
| Jun | 0.1340 | 0.1380 | 0.1770 | 0.2100 | 0.2190 | 0.1970 |
| Jul | 0.0749 | 0.1440 | 0.1740 | 0.1940 | 0.2050 | 0.1890 |
| Aug | 0.1010 | 0.2010 | 0.1890 | 0.2090 | 0.2200 | 0.2290 |
| Sep | 0.0331 | 0.1910 | 0.2250 | 0.1730 | 0.2280 | 0.2120 |
| Oct | 0.1460 | 0.1740 | 0.1970 | 0.2110 | 0.2470 | 0.2170 |
| Nov | 0.1080 | 0.1830 | 0.2090 | 0.2020 | 0.2070 | 0.2050 |
| Dec | 0.1040 | 0.1550 | 0.1710 | 0.2100 | 0.2030 | 0.2260 |

| BSS-theo-25 | LT=1 | LT=2 | LT=3 | LT=4 | LT=5 | LT=6 |
|---|---|---|---|---|---|---|
| Jan | 0.2555 | 0.3242 | 0.2449 | -0.0150 | -0.2283 | -0.0103 |
| Feb | -0.0276 | 0.0395 | 0.2365 | 0.0640 | -0.0857 | 0.0262 |
| Mar | 0.8274 | 0.0714 | 0.2756 | 0.1990 | 0.0296 | -0.0606 |
| Apr | 0.6890 | 0.1575 | -0.1955 | -0.0204 | -0.2090 | -0.0242 |
| May | 0.0890 | -0.0186 | -0.1321 | -0.1747 | -0.0443 | -0.0049 |
| Jun | -0.2072 | -0.1129 | -0.0172 | -0.1602 | -0.1711 | 0.0199 |
| Jul | -1.2094 | -0.2743 | -0.1299 | -0.2050 | -0.1326 | 0.0156 |
| Aug | -0.0620 | -0.2182 | -0.0678 | -0.1359 | -0.1579 | -0.2793 |
| Sep | 0.6390 | -0.1576 | -0.2570 | 0.1218 | -0.1343 | -0.0443 |
| Oct | -0.0282 | 0.0169 | -0.1193 | -0.0293 | -0.2167 | -0.0637 |
| Nov | -0.1146 | -0.1024 | -0.0503 | -0.0151 | 0.0048 | 0.0097 |
| Dec | -1.0635 | -0.0265 | 0.0707 | -0.0606 | -0.1033 | -0.1133 |

| BS-theo-75 | LT=1 | LT=2 | LT=3 | LT=4 | LT=5 | LT=6 |
|---|---|---|---|---|---|---|
| Jan | 0.0479 | 0.1440 | 0.1790 | 0.1710 | 0.1860 | 0.1540 |
| Feb | 0.1660 | 0.2590 | 0.2480 | 0.2400 | 0.2840 | 0.2190 |
| Mar | 0.1200 | 0.1650 | 0.2570 | 0.2300 | 0.2100 | 0.2230 |
| Apr | 0.0904 | 0.1720 | 0.2030 | 0.2270 | 0.2030 | 0.1790 |
| May | 0.1550 | 0.1680 | 0.1800 | 0.2340 | 0.1940 | 0.1910 |
| Jun | 0.1360 | 0.1520 | 0.1670 | 0.1570 | 0.2180 | 0.1840 |
| Jul | 0.1380 | 0.1920 | 0.2020 | 0.2210 | 0.2290 | 0.2040 |
| Aug | 0.1300 | 0.1870 | 0.1750 | 0.2220 | 0.2060 | 0.2380 |
| Sep | 0.1120 | 0.2320 | 0.2180 | 0.2660 | 0.2210 | 0.2320 |
| Oct | 0.0884 | 0.1620 | 0.2170 | 0.2060 | 0.2140 | 0.2170 |
| Nov | 0.1320 | 0.1920 | 0.2380 | 0.2330 | 0.2420 | 0.2520 |
| Dec | 0.0810 | 0.1060 | 0.2070 | 0.2420 | 0.1900 | 0.2300 |

| BSS-theo-75 | LT=1 | LT=2 | LT=3 | LT=4 | LT=5 | LT=6 |
|---|---|---|---|---|---|---|
| Jan | 0.3356 | 0.1273 | -0.2606 | 0.0656 | 0.0159 | 0.1630 |
| Feb | -0.4435 | -0.3560 | -0.1324 | -0.0435 | -0.1592 | 0.0045 |
| Mar | -0.1111 | 0.1316 | -0.3526 | -0.1330 | -0.0769 | -0.2186 |
| Apr | -0.6527 | -0.3871 | -0.3268 | -0.1823 | -0.0798 | 0.1182 |
| May | -0.6454 | -0.2263 | -0.3043 | -0.4268 | 0.0396 | 0.1033 |
| Jun | -0.2143 | -0.1259 | -0.0438 | 0.0188 | -0.2179 | 0.1280 |
| Jul | -0.2432 | -0.1779 | -0.1222 | -0.1693 | -0.2246 | 0.0145 |
| Aug | 0.1216 | -0.1472 | -0.0671 | -0.2265 | -0.0619 | -0.2526 |
| Sep | 0.3000 | -0.2021 | -0.1534 | -0.3711 | -0.1162 | -0.1837 |
| Oct | -0.0449 | -0.0062 | -0.2191 | -0.0674 | -0.1568 | -0.0905 |
| Nov | -1.4859 | -0.1566 | -0.1442 | -0.2263 | -0.1579 | -0.3057 |
| Dec | -0.4286 | 0.0093 | -0.1897 | -0.2222 | -0.0497 | -0.1220 |

| BS-act-25 | LT=1 | LT=2 | LT=3 | LT=4 | LT=5 | LT=6 |
|---|---|---|---|---|---|---|
| Jan | 0.2300 | 0.2520 | 0.1800 | 0.2030 | 0.2330 | 0.2030 |
| Feb | 0.1930 | 0.2550 | 0.2240 | 0.2080 | 0.2210 | 0.2370 |
| Mar | 0.1080 | 0.1480 | 0.1750 | 0.2300 | 0.2410 | 0.2100 |
| Apr | 0.1770 | 0.1790 | 0.2340 | 0.2120 | 0.2490 | 0.2310 |
| May | 0.2240 | 0.2360 | 0.2520 | 0.2450 | 0.2370 | 0.2170 |
| Jun | 0.2120 | 0.1990 | 0.2080 | 0.2280 | 0.2320 | 0.2090 |
| Jul | 0.1960 | 0.1990 | 0.1490 | 0.2240 | 0.2230 | 0.2170 |
| Aug | 0.1550 | 0.2320 | 0.2130 | 0.2090 | 0.2500 | 0.1990 |
| Sep | 0.1850 | 0.1300 | 0.2500 | 0.2340 | 0.2040 | 0.2240 |
| Oct | 0.1400 | 0.2100 | 0.1970 | 0.2110 | 0.2410 | 0.2230 |
| Nov | 0.2230 | 0.2020 | 0.2450 | 0.2510 | 0.2010 | 0.2290 |
| Dec | 0.2740 | 0.2150 | 0.1710 | 0.2290 | 0.2390 | 0.2630 |

| BSS-act-25 | LT=1 | LT=2 | LT=3 | LT=4 | LT=5 | LT=6 |
|---|---|---|---|---|---|---|
| Jan | -0.2568 | -0.5181 | 0.0217 | -0.0628 | -0.2073 | -0.0684 |
| Feb | 0.0302 | 0.1748 | 0.0508 | 0.0169 | -0.0728 | -0.1505 |
| Mar | 0.2230 | 0.0327 | 0.3566 | -0.0314 | -0.1587 | -0.1230 |
| Apr | 0.0829 | 0.1225 | -0.3526 | -0.0600 | -0.3387 | -0.1324 |
| May | -0.2308 | -0.0306 | -0.0769 | -0.1239 | -0.0872 | -0.1071 |
| Jun | -0.1042 | -0.3007 | -0.0246 | -0.1287 | -0.2147 | -0.0609 |
| Jul | -0.0103 | -0.0585 | 0.1287 | -0.0516 | -0.0229 | -0.0637 |
| Aug | -0.0131 | -0.2211 | -0.0867 | -0.0885 | -0.2438 | -0.0051 |
| Sep | -0.2500 | 0.1667 | -0.3369 | -0.2251 | -0.0149 | -0.1371 |
| Oct | 0.1304 | -0.0396 | -0.0314 | 0.0094 | -0.1756 | -0.1206 |
| Nov | -0.2528 | 0.0734 | -0.0652 | -0.0819 | 0.1299 | -0.0362 |
| Dec | -0.5480 | 0.0000 | 0.1407 | -0.1450 | -0.2132 | -0.2767 |

| BS-act-75 | LT=1 | LT=2 | LT=3 | LT=4 | LT=5 | LT=6 |
|---|---|---|---|---|---|---|
| Jan | 0.1150 | 0.1690 | 0.1980 | 0.1520 | 0.1550 | 0.1660 |
| Feb | 0.2410 | 0.2590 | 0.2320 | 0.2500 | 0.2120 | 0.1970 |
| Mar | 0.2170 | 0.2470 | 0.2690 | 0.2300 | 0.2230 | 0.1850 |
| Apr | 0.1690 | 0.2020 | 0.2720 | 0.2270 | 0.1720 | 0.1980 |
| May | 0.1790 | 0.1620 | 0.2040 | 0.2340 | 0.2480 | 0.1940 |
| Jun | 0.1910 | 0.2000 | 0.1740 | 0.1510 | 0.2560 | 0.2390 |
| Jul | 0.2050 | 0.1980 | 0.2080 | 0.2280 | 0.2600 | 0.2100 |
| Aug | 0.1910 | 0.1990 | 0.2110 | 0.2160 | 0.2370 | 0.2500 |
| Sep | 0.1720 | 0.2140 | 0.1820 | 0.1990 | 0.2090 | 0.1950 |
| Oct | 0.1550 | 0.1990 | 0.2170 | 0.2120 | 0.2260 | 0.2410 |
| Nov | 0.1800 | 0.2640 | 0.2740 | 0.2450 | 0.2050 | 0.2280 |
| Dec | 0.2080 | 0.1360 | 0.1880 | 0.2300 | 0.2090 | 0.2300 |

| BSS-act-75 | LT=1 | LT=2 | LT=3 | LT=4 | LT=5 | LT=6 |
|---|---|---|---|---|---|---|
| Jan | -1.0175 | -0.0696 | -0.5000 | 0.0617 | 0.0774 | 0.0119 |
| Feb | -0.1814 | -0.5697 | -0.2961 | -0.3298 | -0.0928 | -0.0103 |
| Mar | -0.3152 | -0.2602 | -0.4541 | -0.4744 | -0.3851 | -0.2013 |
| Apr | -0.5794 | -0.4532 | -0.5281 | -0.2337 | -0.0617 | -0.0645 |
| May | -0.5299 | -0.3279 | -0.2830 | -0.4444 | -0.2984 | 0.0051 |
| Jun | -0.5280 | -0.3245 | 0.0169 | 0.2296 | -0.1907 | -0.0437 |
| Jul | -0.0904 | -0.0102 | -0.1429 | -0.1813 | -0.3265 | -0.0145 |
| Aug | -0.1938 | 0.0744 | -0.0933 | -0.1803 | -0.2092 | -0.3158 |
| Sep | -0.2932 | -0.0754 | 0.0267 | -0.0258 | -0.0773 | 0.0051 |
| Oct | -0.0993 | -0.0206 | -0.1128 | -0.0242 | -0.0971 | -0.1814 |
| Nov | -0.1765 | -0.1046 | -0.1760 | -0.2250 | 0.0639 | -0.0962 |
| Dec | -0.3595 | -0.0625 | -0.0053 | -0.1275 | -0.0773 | -0.1005 |

| BS-A/T-25 | LT=1 | LT=2 | LT=3 | LT=4 | LT=5 | LT=6 |
|---|---|---|---|---|---|---|
| Jan | 13.17 | 35.40 | 82.22 | 100.00 | 97.00 | 97.04 |
| Feb | 25.08 | 66.67 | 69.20 | 91.35 | 85.97 | 78.48 |
| Mar | 15.83 | 79.05 | 93.14 | 70.00 | 81.74 | 100.00 |
| Apr | 14.18 | 59.78 | 67.95 | 94.34 | 97.59 | 91.77 |
| May | 32.46 | 86.59 | 71.43 | 79.59 | 89.45 | 94.47 |
| Jun | 63.21 | 69.35 | 85.10 | 92.11 | 94.40 | 94.26 |
| Jul | 38.21 | 72.36 | 116.78 | 86.61 | 91.93 | 87.10 |
| Aug | 65.16 | 86.64 | 88.73 | 100.00 | 88.00 | 115.08 |
| Sep | 17.89 | 146.92 | 90.00 | 73.93 | 111.76 | 94.64 |
| Oct | 104.29 | 82.86 | 100.00 | 100.00 | 102.49 | 97.31 |
| Nov | 48.43 | 90.59 | 85.31 | 80.48 | 102.99 | 89.52 |
| Dec | 37.96 | 72.09 | 100.00 | 91.70 | 84.94 | 85.93 |

| BS-A/T-75 | LT=1 | LT=2 | LT=3 | LT=4 | LT=5 | LT=6 |
|---|---|---|---|---|---|---|
| Jan | 41.65 | 85.21 | 90.40 | 112.50 | 120.00 | 92.77 |
| Feb | 68.88 | 100.00 | 106.90 | 96.00 | 133.96 | 111.17 |
| Mar | 55.30 | 66.80 | 95.54 | 100.00 | 94.17 | 120.54 |
| Apr | 53.49 | 85.15 | 74.63 | 100.00 | 118.02 | 90.40 |
| May | 86.59 | 103.70 | 88.24 | 100.00 | 78.23 | 98.45 |
| Jun | 71.20 | 76.00 | 95.98 | 103.97 | 85.16 | 76.99 |
| Jul | 67.32 | 96.97 | 97.12 | 96.93 | 88.08 | 97.14 |
| Aug | 68.06 | 93.97 | 82.94 | 102.78 | 86.92 | 95.20 |
| Sep | 65.12 | 108.41 | 119.78 | 133.67 | 105.74 | 118.97 |
| Oct | 57.03 | 81.82 | 100.00 | 97.17 | 94.69 | 90.04 |
| Nov | 73.33 | 72.73 | 86.86 | 95.10 | 118.05 | 110.53 |
| Dec | 38.94 | 77.94 | 110.11 | 105.22 | 90.91 | 100.00 |

**Table 13: Skill scores in the Danube**



| BS-theo-25 | LT=1 | LT=2 | LT=3 | LT=4 | LT=5 | LT=6 |
|---|---|---|---|---|---|---|
| Jan | 0.0667 | 0.0217 | 0.0633 | 0.0983 | 0.1750 | 0.1990 |
| Feb | 0.0100 | 0.0609 | 0.0413 | 0.0932 | 0.1540 | 0.2290 |
| Mar | 0.0488 | 0.0496 | 0.1260 | 0.1250 | 0.1260 | 0.1290 |
| Apr | 0.0873 | 0.0964 | 0.1120 | 0.1460 | 0.1420 | 0.1600 |
| May | 0.0664 | 0.0598 | 0.1190 | 0.1230 | 0.2170 | 0.2130 |
| Jun | 0.0320 | 0.1050 | 0.1040 | 0.1250 | 0.1510 | 0.2520 |
| Jul | 0.0361 | 0.1100 | 0.1320 | 0.1520 | 0.1600 | 0.1650 |
| Aug | 0.0639 | 0.1550 | 0.2210 | 0.2130 | 0.1880 | 0.1970 |
| Sep | 0.0755 | 0.1150 | 0.1910 | 0.1990 | 0.1870 | 0.1710 |
| Oct | 0.0857 | 0.1700 | 0.1420 | 0.1600 | 0.1970 | 0.2050 |
| Nov | 0.0333 | 0.0978 | 0.1690 | 0.1590 | 0.1780 | 0.1910 |
| Dec | 0.0333 | 0.0317 | 0.0854 | 0.1680 | 0.1750 | 0.1740 |

| BSS-theo-25 | LT=1 | LT=2 | LT=3 | LT=4 | LT=5 | LT=6 |
|---|---|---|---|---|---|---|
| Jan | -0.9277 | -0.1302 | -0.2387 | 0.3818 | -0.1076 | -0.2061 |
| Feb | 0.8516 | -0.6964 | 0.0418 | -0.0321 | 0.2524 | -0.1684 |
| Mar | 0.4442 | 0.5137 | 0.0935 | 0.0157 | -0.1776 | 0.0652 |
| Apr | 0.0224 | 0.0360 | 0.0088 | -0.0815 | 0.0274 | -0.0884 |
| May | -0.9762 | 0.3311 | -0.1333 | 0.0956 | -0.2330 | -0.0340 |
| Jun | 0.3509 | -0.3690 | 0.0189 | 0.1007 | 0.1371 | -0.1831 |
| Jul | 0.4224 | 0.0090 | 0.0704 | -0.0556 | 0.0184 | 0.0934 |
| Aug | 0.1787 | -0.0544 | -0.2486 | -0.2604 | -0.1750 | -0.1657 |
| Sep | -0.2276 | 0.1786 | -0.1235 | -0.0699 | -0.0506 | 0.0172 |
| Oct | 0.0664 | -0.2879 | -0.0365 | -0.0390 | -0.1453 | -0.1022 |
| Nov | 0.1440 | 0.1028 | -0.3852 | -0.3140 | -0.1946 | -0.2819 |
| Dec | -0.0707 | 0.2892 | 0.1867 | -0.4000 | -0.3158 | -0.1373 |

| BS-theo-75 | LT=1 | LT=2 | LT=3 | LT=4 | LT=5 | LT=6 |
|---|---|---|---|---|---|---|
| Jan | 0.1190 | 0.0812 | 0.1390 | 0.1630 | 0.2010 | 0.2050 |
| Feb | 0.1530 | 0.1090 | 0.1100 | 0.1330 | 0.1850 | 0.1840 |
| Mar | 0.1120 | 0.1350 | 0.1440 | 0.1440 | 0.1760 | 0.2150 |
| Apr | 0.0658 | 0.1070 | 0.1620 | 0.1950 | 0.1780 | 0.2170 |
| May | 0.0884 | 0.1160 | 0.1360 | 0.1610 | 0.1590 | 0.1820 |
| Jun | 0.1070 | 0.1720 | 0.1870 | 0.2460 | 0.2060 | 0.2540 |
| Jul | 0.0683 | 0.1750 | 0.1660 | 0.2450 | 0.2290 | 0.2110 |
| Aug | 0.0873 | 0.1290 | 0.1660 | 0.2170 | 0.2210 | 0.2550 |
| Sep | 0.1060 | 0.0733 | 0.1760 | 0.1550 | 0.1760 | 0.2200 |
| Oct | 0.0160 | 0.1130 | 0.1400 | 0.1880 | 0.2330 | 0.2440 |
| Nov | 0.0515 | 0.1330 | 0.1630 | 0.1620 | 0.1850 | 0.2350 |
| Dec | 0.1180 | 0.0766 | 0.0837 | 0.1540 | 0.1400 | 0.2010 |

| BSS-theo-75 | LT=1 | LT=2 | LT=3 | LT=4 | LT=5 | LT=6 |
|---|---|---|---|---|---|---|
| Jan | -0.8919 | 0.2618 | 0.1472 | 0.1510 | -0.1486 | -0.2202 |
| Feb | -3.2033 | -0.3196 | -0.2486 | 0.0634 | -0.0571 | -0.1646 |
| Mar | -0.9344 | -0.0465 | 0.0526 | 0.0828 | 0.0538 | 0.0092 |
| Apr | 0.2157 | 0.0446 | -0.0318 | -0.1207 | 0.0056 | -0.1481 |
| May | -0.0625 | 0.1471 | 0.1905 | 0.0640 | 0.1167 | 0.1574 |
| Jun | -0.0841 | -0.1026 | -0.1065 | -0.1942 | -0.0510 | -0.2700 |
| Jul | 0.2826 | 0.0642 | 0.1123 | -0.3243 | -0.1684 | -0.1105 |
| Aug | -0.2159 | 0.1569 | 0.1782 | -0.0960 | -0.0833 | -0.2319 |
| Sep | -0.1547 | 0.1998 | -0.1503 | 0.1092 | 0.0383 | -0.1892 |
| Oct | 0.6049 | -0.0367 | 0.0541 | -0.0503 | -0.1949 | -0.2842 |
| Nov | 0.1227 | -0.1982 | -0.3252 | -0.0800 | 0.0212 | -0.2368 |
| Dec | -1.3366 | -0.0804 | 0.0772 | -0.4528 | 0.0000 | -0.1754 |

| BS-act-25 | LT=1 | LT=2 | LT=3 | LT=4 | LT=5 | LT=6 |
|---|---|---|---|---|---|---|
| Jan | 0.4000 | 0.3160 | 0.2950 | 0.2610 | 0.2880 | 0.2430 |
| Feb | 0.4020 | 0.4150 | 0.3830 | 0.3410 | 0.3020 | 0.3140 |
| Mar | 0.3550 | 0.2970 | 0.3530 | 0.3790 | 0.3360 | 0.2890 |
| Apr | 0.2480 | 0.2690 | 0.2780 | 0.2460 | 0.2900 | 0.2570 |
| May | 0.2390 | 0.2330 | 0.2730 | 0.2520 | 0.2620 | 0.2600 |
| Jun | 0.3500 | 0.2110 | 0.2160 | 0.2490 | 0.2420 | 0.2310 |
| Jul | 0.3180 | 0.2770 | 0.2130 | 0.2210 | 0.2720 | 0.2240 |
| Aug | 0.3030 | 0.3150 | 0.2060 | 0.2590 | 0.2090 | 0.2670 |
| Sep | 0.3150 | 0.2390 | 0.2790 | 0.2080 | 0.2140 | 0.2100 |
| Oct | 0.2460 | 0.3310 | 0.2300 | 0.2540 | 0.2240 | 0.2020 |
| Nov | 0.3330 | 0.2640 | 0.3000 | 0.2590 | 0.2480 | 0.2120 |
| Dec | 0.3330 | 0.3070 | 0.2700 | 0.3100 | 0.2630 | 0.2320 |

| BSS-act-25 | LT=1 | LT=2 | LT=3 | LT=4 | LT=5 | LT=6 |
|---|---|---|---|---|---|---|
| Jan | -0.3289 | 0.0511 | 0.0264 | -0.0653 | -0.2576 | -0.0946 |
| Feb | -0.1198 | -0.0641 | -0.0159 | -0.0656 | -0.1439 | -0.2124 |
| Mar | 0.0166 | 0.1027 | 0.0302 | -0.0798 | -0.1626 | -0.1031 |
| Apr | 0.0462 | -0.1303 | -0.1880 | -0.0982 | -0.1508 | -0.0363 |
| May | -0.0529 | 0.0085 | -0.3448 | -0.2857 | -0.3367 | -0.1111 |
| Jun | -0.3258 | 0.0321 | 0.0270 | -0.3757 | -0.3081 | -0.2031 |
| Jul | -0.2823 | -0.2422 | 0.0274 | -0.0091 | -0.4316 | -0.1667 |
| Aug | -0.1264 | -0.5000 | 0.0190 | -0.2634 | 0.0142 | -0.4355 |
| Sep | -0.0465 | -0.0302 | -0.2455 | 0.0142 | 0.0093 | 0.0094 |
| Oct | 0.2141 | -0.3187 | -0.0748 | -0.1925 | -0.0821 | 0.0427 |
| Nov | -0.0812 | -0.0645 | -0.2397 | -0.1614 | -0.2338 | -0.0341 |
| Dec | -0.0505 | -0.0964 | -0.1489 | -0.2971 | -0.2009 | -0.0741 |

| BS-act-75 | LT=1 | LT=2 | LT=3 | LT=4 | LT=5 | LT=6 |
|---|---|---|---|---|---|---|
| Jan | 0.2710 | 0.2000 | 0.1830 | 0.1760 | 0.1880 | 0.1680 |
| Feb | 0.3570 | 0.3030 | 0.2670 | 0.2450 | 0.2350 | 0.1960 |
| Mar | 0.3360 | 0.2630 | 0.2590 | 0.2450 | 0.2760 | 0.2210 |
| Apr | 0.2230 | 0.1920 | 0.1530 | 0.2680 | 0.2530 | 0.2350 |
| May | 0.3550 | 0.2130 | 0.2030 | 0.1900 | 0.2320 | 0.3040 |
| Jun | 0.1740 | 0.1660 | 0.2110 | 0.2040 | 0.1620 | 0.2050 |
| Jul | 0.1590 | 0.1810 | 0.1600 | 0.1790 | 0.2290 | 0.1920 |
| Aug | 0.2390 | 0.2260 | 0.2020 | 0.1870 | 0.1610 | 0.1880 |
| Sep | 0.1720 | 0.2490 | 0.2120 | 0.2150 | 0.2250 | 0.1530 |
| Oct | 0.0826 | 0.1010 | 0.1040 | 0.1520 | 0.1900 | 0.2010 |
| Nov | 0.1480 | 0.1450 | 0.1030 | 0.1320 | 0.1670 | 0.1870 |
| Dec | 0.1910 | 0.1740 | 0.1990 | 0.1660 | 0.1880 | 0.2310 |

| BSS-act-75 | LT=1 | LT=2 | LT=3 | LT=4 | LT=5 | LT=6 |
|---|---|---|---|---|---|---|
| Jan | 0.0687 | -0.4706 | -0.5776 | -0.4309 | -0.7736 | -0.2727 |
| Feb | -0.2268 | -0.0341 | -0.5170 | -0.5506 | -0.5563 | -0.6610 |
| Mar | 0.0455 | -0.1004 | -0.0882 | -0.2827 | -0.7806 | -0.4636 |
| Apr | -0.1263 | 0.0495 | 0.2609 | -0.2018 | -0.0498 | -0.1750 |
| May | 0.0193 | -0.1833 | -0.2229 | 0.0206 | -0.1154 | -0.3103 |
| Jun | -0.2254 | -0.1942 | -0.1105 | -0.1657 | 0.0182 | -0.1517 |
| Jul | 0.1311 | -0.1173 | 0.0244 | 0.0529 | -0.2312 | -0.0105 |
| Aug | 0.1148 | -0.0367 | -0.0576 | -0.1615 | 0.0123 | -0.2450 |
| Sep | -0.0178 | -0.0779 | 0.1751 | -0.0287 | -0.0714 | 0.2783 |
| Oct | -0.0883 | -0.3272 | 0.0370 | 0.0000 | -0.0556 | -0.0691 |
| Nov | -0.1840 | -0.5778 | -0.0098 | 0.0365 | 0.0060 | -0.0936 |
| Dec | 0.0591 | -0.1918 | -0.2675 | -0.2296 | 0.0208 | -0.1786 |

| BS-A/T-25 | LT=1 | LT=2 | LT=3 | LT=4 | LT=5 | LT=6 |
|---|---|---|---|---|---|---|
| Jan | 16.68 | 6.87 | 21.46 | 37.66 | 60.76 | 81.89 |
| Feb | 2.48 | 14.67 | 10.78 | 27.33 | 50.99 | 72.93 |
| Mar | 13.75 | 16.70 | 35.69 | 32.98 | 37.50 | 44.64 |
| Apr | 35.20 | 35.84 | 40.29 | 59.35 | 48.97 | 62.26 |
| May | 27.78 | 25.67 | 43.59 | 48.81 | 82.82 | 81.92 |
| Jun | 9.14 | 49.76 | 48.15 | 50.20 | 62.40 | 109.09 |
| Jul | 11.35 | 39.71 | 61.97 | 68.78 | 58.82 | 73.66 |
| Aug | 21.09 | 49.21 | 107.28 | 82.24 | 89.95 | 73.78 |
| Sep | 23.97 | 48.12 | 68.46 | 95.67 | 87.38 | 81.43 |
| Oct | 34.84 | 51.36 | 61.74 | 62.99 | 87.95 | 101.49 |
| Nov | 10.00 | 37.05 | 56.33 | 61.39 | 71.77 | 90.09 |
| Dec | 10.00 | 10.33 | 31.63 | 54.19 | 66.54 | 75.00 |

| BS-A/T-75 | LT=1 | LT=2 | LT=3 | LT=4 | LT=5 | LT=6 |
|---|---|---|---|---|---|---|
| Jan | 43.91 | 40.60 | 75.96 | 92.61 | 106.91 | 122.02 |
| Feb | 42.86 | 35.97 | 41.20 | 54.29 | 78.72 | 93.88 |
| Mar | 33.33 | 51.33 | 55.60 | 58.78 | 63.77 | 97.29 |
| Apr | 29.51 | 55.73 | 105.88 | 72.76 | 70.36 | 92.34 |
| May | 24.90 | 54.46 | 67.00 | 84.74 | 68.53 | 59.87 |
| Jun | 61.49 | 103.61 | 88.63 | 120.59 | 127.16 | 123.90 |
| Jul | 42.96 | 96.69 | 103.75 | 136.87 | 100.00 | 109.90 |
| Aug | 36.53 | 57.08 | 82.18 | 116.04 | 137.27 | 135.64 |
| Sep | 61.63 | 29.44 | 83.02 | 72.09 | 78.22 | 143.79 |
| Oct | 19.37 | 111.88 | 134.62 | 123.68 | 122.63 | 121.39 |
| Nov | 34.80 | 91.72 | 158.25 | 122.73 | 110.78 | 125.67 |
| Dec | 61.78 | 44.02 | 42.06 | 92.77 | 74.47 | 87.01 |

**Table 14: Skill scores in the Volga**





| BS-theo-25 | LT=1 | LT=2 | LT=3 | LT=4 | LT=5 | LT=6 |
|---|---|---|---|---|---|---|
| Jan | 0.1130 | 0.1860 | 0.1750 | 0.1690 | 0.2090 | 0.2060 |
| Feb | 0.0775 | 0.0675 | 0.1480 | 0.2090 | 0.1990 | 0.1900 |
| Mar | 0.1120 | 0.0524 | 0.1070 | 0.1850 | 0.2090 | 0.2290 |
| Apr | 0.1290 | 0.1080 | 0.1410 | 0.1850 | 0.2180 | 0.2250 |
| May | 0.0658 | 0.0457 | 0.0975 | 0.1610 | 0.1880 | 0.2200 |
| Jun | 0.0190 | 0.1290 | 0.0623 | 0.1640 | 0.1660 | 0.2080 |
| Jul | 0.0556 | 0.0620 | 0.1120 | 0.0934 | 0.1640 | 0.1640 |
| Aug | 0.0559 | 0.0441 | 0.0551 | 0.0939 | 0.1040 | 0.1530 |
| Sep | 0.0493 | 0.0813 | 0.0804 | 0.0837 | 0.1230 | 0.1000 |
| Oct | 0.0664 | 0.0978 | 0.1020 | 0.0895 | 0.1040 | 0.0893 |
| Nov | 0.0658 | 0.1390 | 0.1880 | 0.2040 | 0.1870 | 0.2050 |
| Dec | 0.1290 | 0.2030 | 0.1820 | 0.1950 | 0.2070 | 0.1930 |

| BSS-theo-25 | LT=1 | LT=2 | LT=3 | LT=4 | LT=5 | LT=6 |
|---|---|---|---|---|---|---|
| Jan | -0.1142 | -0.3461 | -0.0064 | -0.0036 | -0.1821 | -0.0635 |
| Feb | -1.0939 | 0.3248 | -0.0337 | -0.3067 | -0.0889 | -0.0214 |
| Mar | -5.8539 | 0.2502 | 0.2599 | -0.1674 | -0.1932 | -0.2283 |
| Apr | -0.5877 | 0.2241 | -0.2465 | -0.2529 | -0.2990 | -0.1068 |
| May | -1.4894 | 0.4736 | 0.1705 | -0.3364 | -0.2125 | -0.4716 |
| Jun | 0.2102 | -0.7030 | 0.4925 | -0.2146 | -0.2099 | -0.2977 |
| Jul | -4.5636 | -0.8746 | -0.3868 | 0.2167 | -0.2018 | -0.1818 |
| Aug | -1.0866 | -2.0039 | -0.4059 | -0.1146 | 0.1895 | -0.0169 |
| Sep | 0.2698 | -0.0067 | -0.0762 | 0.0264 | -0.1052 | 0.1838 |
| Oct | -0.3358 | -0.3738 | -0.2361 | -0.1705 | -0.0551 | 0.1874 |
| Nov | 0.4681 | 0.0222 | -0.2047 | -0.3695 | -0.3041 | -0.3320 |
| Dec | -0.0251 | -0.6203 | -0.1153 | -0.0686 | -0.1567 | -0.1161 |

| BS-theo-75 | LT=1 | LT=2 | LT=3 | LT=4 | LT=5 | LT=6 |
|---|---|---|---|---|---|---|
| Jan | 0.0501 | 0.1090 | 0.1310 | 0.1410 | 0.1460 | 0.1920 |
| Feb | 0.0855 | 0.0556 | 0.1230 | 0.1470 | 0.1310 | 0.1660 |
| Mar | 0.1340 | 0.0610 | 0.0829 | 0.1360 | 0.1760 | 0.1650 |
| Apr | 0.0989 | 0.0377 | 0.0767 | 0.0818 | 0.1460 | 0.1420 |
| May | 0.0510 | 0.0521 | 0.0711 | 0.0801 | 0.0857 | 0.1560 |
| Jun | 0.0468 | 0.0760 | 0.0716 | 0.0854 | 0.1070 | 0.1150 |
| Jul | 0.0336 | 0.0190 | 0.0598 | 0.0576 | 0.1020 | 0.1230 |
| Aug | 0.0333 | 0.0336 | 0.0105 | 0.0510 | 0.0471 | 0.0904 |
| Sep | 0.0438 | 0.0350 | 0.0281 | 0.0399 | 0.0584 | 0.0441 |
| Oct | 0.0501 | 0.0612 | 0.0490 | 0.0529 | 0.0408 | 0.0780 |
| Nov | 0.0537 | 0.1060 | 0.1160 | 0.0876 | 0.0871 | 0.0758 |
| Dec | 0.1020 | 0.1410 | 0.1310 | 0.1250 | 0.1410 | 0.1400 |

| BSS-theo-75 | LT=1 | LT=2 | LT=3 | LT=4 | LT=5 | LT=6 |
|---|---|---|---|---|---|---|
| Jan | -0.1208 | -0.1720 | 0.0368 | 0.0600 | 0.1152 | -0.2308 |
| Feb | -3.9709 | 0.3638 | -0.0250 | -0.0970 | 0.1863 | 0.0621 |
| Mar | -1.5769 | -0.1531 | -0.0467 | -0.0794 | -0.1812 | -0.0577 |
| Apr | -10.2514 | -0.0590 | -0.3362 | 0.0757 | -0.0977 | 0.1013 |
| May | -0.2531 | -0.4472 | -0.0676 | -0.2438 | 0.0941 | -0.0833 |
| Jun | -0.3970 | -0.5574 | -0.1205 | -0.1178 | -0.3789 | -0.1058 |
| Jul | -0.0735 | 0.4461 | -0.1865 | -0.2608 | | -0.5649 |
| Aug | -1.8220 | -0.1237 | 0.6729 | 0.0538 | 0.2547 | -0.1942 |
| Sep | -0.1806 | 0.0141 | 0.1831 | -0.1433 | -0.0998 | -0.0232 |
| Oct | -0.2748 | -0.5896 | -0.1639 | -0.3812 | 0.0286 | -0.4885 |
| Nov | 0.4213 | -0.3469 | -0.6315 | -0.3153 | -0.1899 | -0.1001 |
| Dec | -0.3878 | -0.1750 | -0.1391 | 0.0458 | -0.0763 | 0.0000 |

| BS-act-25 | LT=1 | LT=2 | LT=3 | LT=4 | LT=5 | LT=6 |
|---|---|---|---|---|---|---|
| Jan | 0.3340 | 0.4120 | 0.3440 | 0.3320 | 0.2280 | 0.2380 |
| Feb | 0.4160 | 0.3070 | 0.3680 | 0.2970 | 0.3180 | 0.2340 |
| Mar | 0.4250 | 0.3910 | 0.2980 | 0.3290 | 0.2900 | 0.3170 |
| Apr | 0.4230 | 0.4320 | 0.3160 | 0.2730 | 0.2870 | 0.2630 |
| May | 0.5050 | 0.4370 | 0.4160 | 0.3110 | 0.2820 | 0.3270 |
| Jun | 0.4550 | 0.4020 | 0.3770 | 0.3640 | 0.3200 | 0.2810 |
| Jul | 0.4560 | 0.4260 | 0.3850 | 0.3900 | 0.3640 | 0.3110 |
| Aug | 0.4500 | 0.4380 | 0.4190 | 0.3850 | 0.3830 | 0.3590 |
| Sep | 0.3950 | 0.3900 | 0.4020 | 0.3810 | 0.3830 | 0.4280 |
| Oct | 0.4090 | 0.3310 | 0.3170 | 0.3290 | 0.3380 | 0.3470 |
| Nov | 0.4260 | 0.3480 | 0.2640 | 0.2920 | 0.3170 | 0.2870 |
| Dec | 0.4840 | 0.3940 | 0.3540 | 0.2710 | 0.2520 | 0.2870 |

| BSS-act-25 | LT=1 | LT=2 | LT=3 | LT=4 | LT=5 | LT=6 |
|---|---|---|---|---|---|---|
| Jan | 0.0997 | -0.2189 | -0.1467 | -0.1293 | 0.1972 | 0.1678 |
| Feb | -0.0532 | 0.1127 | -0.1871 | -0.0569 | -0.0821 | 0.0640 |
| Mar | 0.0406 | -0.1534 | 0.0570 | -0.0511 | -0.0821 | -0.2431 |
| Apr | 0.0804 | -0.0800 | 0.0156 | 0.0930 | 0.0772 | 0.0075 |
| May | -0.0120 | 0.0500 | -0.1183 | 0.0281 | 0.0342 | -0.0757 |
| Jun | -0.0631 | -0.0075 | 0.0208 | -0.0931 | -0.1228 | -0.0181 |
| Jul | -0.1287 | -0.0597 | 0.0052 | -0.0317 | -0.1235 | -0.1107 |
| Aug | -0.0638 | -0.0950 | -0.0423 | 0.0351 | 0.0353 | -0.0685 |
| Sep | -0.0340 | 0.0176 | -0.0334 | 0.0499 | -0.0609 | -0.1413 |
| Oct | -0.0568 | 0.0543 | 0.0593 | 0.0237 | -0.0060 | -0.1490 |
| Nov | -0.2456 | -0.0774 | 0.1200 | 0.0426 | -0.0359 | -0.0070 |
| Dec | -0.2804 | -0.1420 | -0.1879 | 0.0491 | 0.0903 | -0.0513 |

| BS-act-75 | LT=1 | LT=2 | LT=3 | LT=4 | LT=5 | LT=6 |
|---|---|---|---|---|---|---|
| Jan | 0.1350 | 0.1780 | 0.1750 | 0.1780 | 0.1770 | 0.1990 |
| Feb | 0.1540 | 0.1650 | 0.1860 | 0.2030 | 0.2120 | 0.1720 |
| Mar | 0.2000 | 0.1800 | 0.1680 | 0.1860 | 0.2010 | 0.2210 |
| Apr | 0.2750 | 0.2440 | 0.2020 | 0.1970 | 0.1960 | 0.2110 |
| May | 0.2690 | 0.2100 | 0.1920 | 0.1370 | 0.1460 | 0.1560 |
| Jun | 0.2470 | 0.2310 | 0.2290 | 0.1880 | 0.1570 | 0.1690 |
| Jul | 0.2340 | 0.2310 | 0.2230 | 0.2210 | 0.2230 | 0.2050 |
| Aug | 0.2330 | 0.2280 | 0.2290 | 0.2090 | 0.2050 | 0.2120 |
| Sep | 0.2380 | 0.2290 | 0.2100 | 0.2160 | 0.1920 | 0.2020 |
| Oct | 0.2200 | 0.2010 | 0.1940 | 0.1740 | 0.1740 | 0.1870 |
| Nov | 0.1870 | 0.1660 | 0.2130 | 0.1850 | 0.1720 | 0.1480 |
| Dec | 0.1930 | 0.2010 | 0.2090 | 0.1670 | 0.1720 | 0.1890 |

| BSS-act-75 | LT=1 | LT=2 | LT=3 | LT=4 | LT=5 | LT=6 |
|---|---|---|---|---|---|---|
| Jan | 0.0146 | -0.1484 | 0.0223 | -0.0230 | 0.0000 | -0.1706 |
| Feb | -0.2031 | -0.1301 | -0.0751 | -0.1154 | -0.1217 | 0.1134 |
| Mar | 0.1266 | 0.0217 | -0.0307 | -0.1071 | -0.0865 | -0.1693 |
| Apr | 0.1158 | -0.0252 | -0.1099 | -0.1257 | 0.0101 | 0.0580 |
| May | -0.1070 | 0.0583 | 0.0495 | 0.1104 | 0.2108 | 0.2353 |
| Jun | -0.0786 | -0.0932 | -0.1117 | 0.0553 | -0.0129 | 0.0611 |
| Jul | -0.0130 | -0.0221 | 0.0176 | -0.0676 | -0.1040 | -0.2349 |
| Aug | 0.0251 | -0.0179 | -0.0223 | 0.0413 | -0.0099 | -0.1522 |
| Sep | -0.0818 | -0.0651 | -0.0145 | -0.0485 | 0.0254 | -0.1038 |
| Oct | -0.0092 | -0.0924 | -0.0430 | 0.0114 | 0.0645 | 0.0650 |
| Nov | -0.0163 | 0.0621 | -0.3396 | -0.1709 | -0.2286 | -0.0571 |
| Dec | -0.2532 | -0.1618 | -0.1943 | 0.0000 | -0.0818 | -0.1887 |

| BS-A/T-25 | LT=1 | LT=2 | LT=3 | LT=4 | LT=5 | LT=6 |
|---|---|---|---|---|---|---|
| Jan | 33.83 | 45.15 | 50.87 | 50.90 | 91.67 | 86.55 |
| Feb | 18.63 | 21.99 | 40.22 | 70.37 | 62.58 | 81.20 |
| Mar | 26.35 | 13.40 | 35.91 | 56.23 | 72.07 | 72.24 |
| Apr | 30.50 | 25.00 | 44.62 | 67.77 | 75.96 | 85.55 |
| May | 13.03 | 24.40 | 23.44 | 51.77 | 66.67 | 67.28 |
| Jun | 4.18 | 32.09 | 16.53 | 45.05 | 51.88 | 74.02 |
| Jul | 12.19 | 14.55 | 29.09 | 23.95 | 45.05 | 52.73 |
| Aug | 12.42 | 10.07 | 13.15 | 24.39 | 27.15 | 42.62 |
| Sep | 12.48 | 20.85 | 20.00 | 21.97 | 32.11 | 23.36 |
| Oct | 16.23 | 29.55 | 32.18 | 27.20 | 30.77 | 25.73 |
| Nov | 15.45 | 39.94 | 71.21 | 69.86 | 58.99 | 71.43 |
| Dec | 26.65 | 51.52 | 51.41 | 71.96 | 82.14 | 67.25 |

| BS-A/T-75 | LT=1 | LT=2 | LT=3 | LT=4 | LT=5 | LT=6 |
|---|---|---|---|---|---|---|
| Jan | 37.11 | 61.24 | 74.86 | 79.21 | 82.49 | 96.48 |
| Feb | 55.52 | 33.70 | 66.13 | 72.41 | 61.79 | 96.51 |
| Mar | 67.00 | 33.89 | 49.35 | 73.12 | 87.56 | 74.66 |
| Apr | 35.96 | 15.45 | 37.97 | 41.52 | 74.49 | 67.30 |
| May | 18.96 | 24.81 | 37.03 | 58.47 | 58.70 | 100.00 |
| Jun | 18.95 | 29.46 | 31.27 | 45.43 | 68.15 | 68.05 |
| Jul | 14.36 | 8.23 | 26.82 | 26.06 | 45.74 | 64.00 |
| Aug | 14.29 | 14.74 | 4.59 | 24.40 | 22.98 | 42.64 |
| Sep | 18.40 | 15.28 | 13.38 | 18.47 | 30.42 | 21.83 |
| Oct | 22.77 | 30.45 | 25.26 | 30.40 | 23.45 | 41.71 |
| Nov | 28.72 | 63.86 | 54.46 | 47.35 | 50.64 | 51.22 |
| Dec | 52.85 | 70.15 | 62.68 | 74.85 | 81.98 | 74.07 |

**Table 15: Skill scores in the Columbia**



| BS-theo-25 | LT=1 | LT=2 | LT=3 | LT=4 | LT=5 | LT=6 |
|---|---|---|---|---|---|---|
| Jan | 0.0278 | 0.0553 | 0.0442 | 0.0547 | 0.1010 | 0.0972 |
| Feb | 0.0425 | 0.0992 | 0.0844 | 0.0972 | 0.0889 | 0.1080 |
| Mar | 0.0689 | 0.0650 | 0.1400 | 0.1420 | 0.1270 | 0.1050 |
| Apr | 0.0647 | 0.0612 | 0.0567 | 0.1330 | 0.0812 | 0.1400 |
| May | 0.0664 | 0.1430 | 0.1020 | 0.0969 | 0.1340 | 0.1270 |
| Jun | 0.0843 | 0.0548 | 0.0708 | 0.1470 | 0.1150 | 0.1060 |
| Jul | 0.1170 | 0.0937 | 0.1050 | 0.1530 | 0.1840 | 0.1570 |
| Aug | 0.0377 | 0.0793 | 0.0920 | 0.1140 | 0.1750 | 0.2030 |
| Sep | 0.0165 | 0.0391 | 0.0419 | 0.0534 | 0.1310 | 0.2040 |
| Oct | 0.0052 | 0.0160 | 0.0419 | 0.0416 | 0.0623 | 0.1190 |
| Nov | 0.0474 | 0.0661 | 0.0554 | 0.0945 | 0.0837 | 0.0857 |
| Dec | 0.0769 | 0.0736 | 0.0317 | 0.0576 | 0.0364 | 0.0972 |

| BSS-theo-25 | LT=1 | LT=2 | LT=3 | LT=4 | LT=5 | LT=6 |
|---|---|---|---|---|---|---|
| Jan | -0.1830 | 0.0659 | 0.3818 | 0.4384 | 0.0288 | 0.1164 |
| Feb | -0.5125 | -0.3626 | -0.1004 | 0.1321 | 0.3162 | 0.2230 |
| Mar | -5.3211 | -0.1168 | -0.4523 | -0.3524 | -0.2841 | 0.1393 |
| Apr | 0.1125 | 0.3598 | 0.2201 | 0.0000 | 0.2877 | 0.0000 |
| May | -0.7946 | -0.3883 | 0.0097 | 0.0310 | -0.1453 | -0.1441 |
| Jun | -2.0879 | 0.3492 | 0.3679 | -0.2895 | -0.0952 | -0.1289 |
| Jul | -1.8676 | -0.5016 | 0.0000 | -0.1504 | -0.2349 | -0.1805 |
| Aug | -0.3810 | -0.6184 | -0.2568 | 0.1915 | -0.1364 | -0.1871 |
| Sep | 0.6108 | -0.0568 | 0.3274 | 0.5189 | 0.2061 | -0.0909 |
| Oct | 0.5372 | 0.4326 | -0.2972 | 0.2877 | 0.4178 | 0.2372 |
| Nov | 0.4716 | 0.0970 | 0.3719 | -0.1553 | 0.0279 | -0.0793 |
| Dec | 0.1549 | -0.7864 | 0.4162 | 0.1312 | 0.4663 | 0.0376 |

| BS-theo-75 | LT=1 | LT=2 | LT=3 | LT=4 | LT=5 | LT=6 |
|---|---|---|---|---|---|---|
| Jan | 0.0667 | 0.0499 | 0.0584 | 0.0601 | 0.0866 | 0.0826 |
| Feb | 0.0348 | 0.0355 | 0.0450 | 0.0536 | 0.0487 | 0.0832 |
| Mar | 0.0344 | 0.0191 | 0.0300 | 0.0590 | 0.0462 | 0.0522 |
| Apr | 0.0336 | 0.0499 | 0.0388 | 0.0226 | 0.0550 | 0.0576 |
| May | 0.0344 | 0.0736 | 0.0446 | 0.0359 | 0.0493 | 0.0678 |
| Jun | 0.0879 | 0.0212 | 0.0708 | 0.0576 | 0.0408 | 0.0507 |
| Jul | 0.0895 | 0.0782 | 0.0669 | 0.0515 | 0.0609 | 0.0795 |
| Aug | 0.1000 | 0.0515 | 0.0697 | 0.0725 | 0.0661 | 0.0683 |
| Sep | 0.1010 | 0.0559 | 0.0562 | 0.0829 | 0.0879 | 0.0713 |
| Oct | 0.0504 | 0.0328 | 0.0741 | 0.0532 | 0.1020 | 0.1020 |
| Nov | 0.0884 | 0.0446 | 0.0584 | 0.0890 | 0.1180 | 0.1110 |
| Dec | 0.0556 | 0.0388 | 0.0397 | 0.0631 | 0.0821 | 0.0939 |

| BSS-theo-75 | LT=1 | LT=2 | LT=3 | LT=4 | LT=5 | LT=6 |
|---|---|---|---|---|---|---|
| Jan | -0.9503 | -0.0267 | -0.0523 | -0.1426 | -0.2146 | -0.0631 |
| Feb | -20.7500 | -0.1993 | -0.0465 | -0.0268 | 0.1209 | -0.1886 |
| Mar | -6.7130 | -1.3012 | 0.0323 | -0.3470 | 0.1554 | 0.1046 |
| Apr | -2.5707 | -1.3649 | 0.1397 | 0.2207 | 0.1115 | 0.0448 |
| May | 0.0601 | -0.1429 | 0.1373 | 0.0452 | 0.1228 | -0.0813 |
| Jun | -5.7099 | 0.0093 | -0.1185 | 0.0695 | -0.1271 | 0.1348 |
| Jul | -2.3902 | -0.9309 | -0.7605 | -0.0981 | -0.5457 | -0.3452 |
| Aug | -12.6612 | -2.1212 | -0.9201 | -1.4167 | -0.5301 | -0.7927 |
| Sep | -618.6319 | -4.3750 | -2.4061 | -1.6742 | -1.4349 | -0.8142 |
| Oct | -60.9165 | -7.8410 | -3.9400 | -1.4860 | -2.1875 | -2.1003 |
| Nov | -0.7931 | -1.6707 | -0.3550 | -0.4883 | -0.4824 | -0.3105 |
| Dec | -3.3780 | -0.4370 | -1.7762 | -0.4180 | -0.4454 | -0.1578 |

| BS-act-25 | LT=1 | LT=2 | LT=3 | LT=4 | LT=5 | LT=6 |
|---|---|---|---|---|---|---|
| Jan | 0.2160 | 0.1340 | 0.1730 | 0.1640 | 0.1410 | 0.1760 |
| Feb | 0.1430 | 0.2330 | 0.1820 | 0.2190 | 0.1990 | 0.1990 |
| Mar | 0.1360 | 0.1590 | 0.2130 | 0.1860 | 0.2020 | 0.1930 |
| Apr | 0.1190 | 0.1400 | 0.1510 | 0.1760 | 0.1720 | 0.1940 |
| May | 0.2000 | 0.2280 | 0.1800 | 0.1970 | 0.2440 | 0.2180 |
| Jun | 0.2060 | 0.1760 | 0.1740 | 0.1900 | 0.2280 | 0.2150 |
| Jul | 0.2140 | 0.1480 | 0.1900 | 0.2130 | 0.2570 | 0.2640 |
| Aug | 0.1100 | 0.1340 | 0.1220 | 0.1690 | 0.2110 | 0.2450 |
| Sep | 0.1440 | 0.1360 | 0.1570 | 0.1810 | 0.2030 | 0.2340 |
| Oct | 0.1810 | 0.1430 | 0.1510 | 0.1260 | 0.1410 | 0.1740 |
| Nov | 0.1750 | 0.1810 | 0.1640 | 0.1850 | 0.1630 | 0.1770 |
| Dec | 0.1310 | 0.1400 | 0.1830 | 0.1420 | 0.1760 | 0.1640 |

| BSS-act-25 | LT=1 | LT=2 | LT=3 | LT=4 | LT=5 | LT=6 |
|---|---|---|---|---|---|---|
| Jan | -0.1192 | 0.1779 | 0.0495 | -0.0513 | 0.2034 | 0.0112 |
| Feb | 0.0272 | -0.1095 | 0.0421 | -0.0046 | 0.0149 | 0.1459 |
| Mar | -0.1826 | 0.0185 | -0.1833 | -0.0276 | 0.0049 | 0.0493 |
| Apr | 0.1185 | -0.2281 | -0.0067 | -0.1069 | -0.1391 | 0.0443 |
| May | -0.1765 | -0.1232 | 0.0217 | 0.1005 | -0.1675 | -0.0187 |
| Jun | -0.2956 | -0.0353 | 0.1386 | 0.1775 | -0.0179 | 0.0271 |
| Jul | -0.2022 | 0.2952 | 0.2607 | 0.2022 | 0.0000 | 0.0737 |
| Aug | 0.1912 | 0.0360 | 0.3297 | 0.2747 | 0.0944 | -0.0251 |
| Sep | 0.1724 | 0.0621 | 0.1130 | 0.1773 | 0.1748 | 0.0290 |
| Oct | -0.1242 | 0.1488 | 0.0904 | 0.1000 | 0.2079 | 0.0279 |
| Nov | 0.0331 | 0.0055 | 0.0787 | -0.0632 | 0.0739 | -0.1346 |
| Dec | 0.1088 | 0.1463 | -0.2039 | 0.0897 | -0.1139 | -0.0186 |

| BS-act-75 | LT=1 | LT=2 | LT=3 | LT=4 | LT=5 | LT=6 |
|---|---|---|---|---|---|---|
| Jan | 0.1330 | 0.1440 | 0.1150 | 0.1730 | 0.1560 | 0.1700 |
| Feb | 0.1320 | 0.1080 | 0.1390 | 0.0912 | 0.1370 | 0.1650 |
| Mar | 0.0890 | 0.0912 | 0.1030 | 0.1280 | 0.0901 | 0.1270 |
| Apr | 0.0336 | 0.0499 | 0.0670 | 0.0529 | 0.0863 | 0.0764 |
| May | 0.0890 | 0.0433 | 0.0567 | 0.0579 | 0.0614 | 0.0929 |
| Jun | 0.1120 | 0.1180 | 0.0890 | 0.1000 | 0.1130 | 0.0992 |
| Jul | 0.0956 | 0.0722 | 0.0791 | 0.0758 | 0.0609 | 0.0826 |
| Aug | 0.1000 | 0.0636 | 0.0636 | 0.0725 | 0.1090 | 0.0804 |
| Sep | 0.1860 | 0.1950 | 0.2080 | 0.1980 | 0.2090 | 0.2290 |
| Oct | 0.1530 | 0.1720 | 0.1350 | 0.1990 | 0.1870 | 0.2540 |
| Nov | 0.1610 | 0.1900 | 0.1740 | 0.1800 | 0.2340 | 0.1470 |
| Dec | 0.1340 | 0.1300 | 0.1730 | 0.1900 | 0.1610 | 0.2150 |

| BSS-act-75 | LT=1 | LT=2 | LT=3 | LT=4 | LT=5 | LT=6 |
|---|---|---|---|---|---|---|
| Jan | 0.1133 | 0.0270 | 0.1288 | -0.0549 | 0.0602 | -0.0119 |
| Feb | 0.0222 | -0.0693 | -0.0221 | 0.2762 | 0.1329 | -0.0443 |
| Mar | 0.2261 | 0.2930 | -0.1294 | -0.0240 | 0.2961 | 0.1699 |
| Apr | 0.1744 | 0.1485 | 0.2796 | 0.1049 | 0.2295 | 0.2724 |
| May | -0.1680 | 0.0069 | -0.0967 | 0.2290 | -0.2899 | 0.0981 |
| Jun | 0.3293 | 0.0084 | -0.1125 | -0.2092 | -0.0561 | -0.0420 |
| Jul | -0.2867 | -0.4765 | -0.0777 | -0.0542 | 0.1109 | 0.1449 |
| Aug | -0.3514 | -0.1778 | -0.6563 | -0.2697 | -0.6490 | -0.2331 |
| Sep | 0.2874 | 0.1631 | 0.0415 | -0.0588 | -0.0097 | -0.2378 |
| Oct | 0.1947 | 0.0899 | 0.2582 | -0.0815 | -0.0747 | -0.3298 |
| Nov | 0.1658 | 0.0952 | 0.1792 | 0.0769 | -0.1415 | 0.2304 |
| Dec | 0.1879 | 0.1216 | 0.0226 | -0.0326 | 0.1250 | -0.0437 |

| BS-A/T-25 | LT=1 | LT=2 | LT=3 | LT=4 | LT=5 | LT=6 |
|---|---|---|---|---|---|---|
| Jan | 12.87 | 41.27 | 25.55 | 33.35 | 71.63 | 55.23 |
| Feb | 29.72 | 42.58 | 46.37 | 44.38 | 44.67 | 54.27 |
| Mar | 50.66 | 40.88 | 65.73 | 76.34 | 62.87 | 54.40 |
| Apr | 54.37 | 43.71 | 37.55 | 75.57 | 47.21 | 72.16 |
| May | 33.20 | 62.72 | 56.67 | 49.19 | 54.92 | 58.26 |
| Jun | 40.92 | 31.14 | 40.69 | 77.37 | 50.44 | 49.30 |
| Jul | 54.67 | 63.31 | 55.26 | 71.83 | 71.60 | 59.47 |
| Aug | 34.27 | 59.18 | 75.41 | 67.46 | 82.94 | 82.86 |
| Sep | 11.46 | 28.75 | 26.69 | 29.50 | 64.53 | 87.18 |
| Oct | 2.89 | 11.19 | 27.75 | 33.02 | 44.18 | 68.39 |
| Nov | 27.09 | 36.52 | 33.78 | 51.08 | 51.35 | 48.42 |
| Dec | 58.70 | 52.57 | 17.32 | 40.56 | 20.68 | 59.27 |

| BS-A/T-75 | LT=1 | LT=2 | LT=3 | LT=4 | LT=5 | LT=6 |
|---|---|---|---|---|---|---|
| Jan | 50.15 | 34.65 | 50.78 | 34.74 | 55.51 | 48.59 |
| Feb | 26.36 | 32.87 | 32.37 | 58.77 | 35.55 | 50.42 |
| Mar | 38.65 | 20.94 | 29.13 | 46.09 | 51.28 | 41.10 |
| Apr | 100.00 | 100.00 | 57.91 | 42.72 | 63.73 | 75.39 |
| May | 38.65 | 169.98 | 78.66 | 62.00 | 80.29 | 72.98 |
| Jun | 78.48 | 17.97 | 79.55 | 57.60 | 36.11 | 51.11 |
| Jul | 93.62 | 108.31 | 84.58 | 67.94 | 100.00 | 96.25 |
| Aug | 100.00 | 80.97 | 109.59 | 100.00 | 60.64 | 84.95 |
| Sep | 54.30 | 28.67 | 27.02 | 41.87 | 42.06 | 31.14 |
| Oct | 32.94 | 19.07 | 54.89 | 26.73 | 54.55 | 40.16 |
| Nov | 54.91 | 23.47 | 33.56 | 49.44 | 50.43 | 75.51 |
| Dec | 41.49 | 29.85 | 22.95 | 33.21 | 50.99 | 43.67 |

**Table 16: Skill scores in the St. Lawrence**



| BS-theo-25 | LT=1 | LT=2 | LT=3 | LT=4 | LT=5 | LT=6 |
|---|---|---|---|---|---|---|
| Jan | 0.0187 | 0.0653 | 0.1310 | 0.1150 | 0.1980 | 0.1600 |
| Feb | 0.0205 | 0.0488 | 0.0886 | 0.1390 | 0.1530 | 0.1900 |
| Mar | 0.0435 | 0.0408 | 0.0832 | 0.0980 | 0.1430 | 0.2050 |
| Apr | 0.0736 | 0.1350 | 0.1620 | 0.1470 | 0.1330 | 0.2070 |
| May | 0.0702 | 0.1720 | 0.1700 | 0.1720 | 0.1600 | 0.1440 |
| Jun | 0.0804 | 0.1230 | 0.1020 | 0.1490 | 0.1970 | 0.2010 |
| Jul | 0.1060 | 0.0901 | 0.1250 | 0.1360 | 0.1610 | 0.2070 |
| Aug | 0.1360 | 0.1170 | 0.1970 | 0.2060 | 0.1700 | 0.2290 |
| Sep | 0.1200 | 0.2520 | 0.2500 | 0.2800 | 0.2360 | 0.2150 |
| Oct | 0.1650 | 0.1600 | 0.2000 | 0.1470 | 0.1720 | 0.1760 |
| Nov | 0.0364 | 0.1230 | 0.1280 | 0.1340 | 0.1910 | 0.1180 |
| Dec | 0.0758 | 0.1390 | 0.1240 | 0.1520 | 0.1300 | 0.1300 |

| BSS-theo-25 | LT=1 | LT=2 | LT=3 | LT=4 | LT=5 | LT=6 |
|---|---|---|---|---|---|---|
| Jan | 0.4228 | 0.1199 | -0.0565 | 0.2333 | -0.1928 | 0.0476 |
| Feb | 0.5675 | 0.3232 | 0.0978 | 0.0733 | -0.0268 | -0.2102 |
| Mar | 0.4689 | 0.1840 | 0.1449 | -0.0392 | -0.1172 | -0.4138 |
| Apr | 0.2405 | -0.1947 | -0.0872 | 0.0068 | 0.1988 | -0.1829 |
| May | 0.2618 | -0.1467 | -0.1258 | -0.0750 | 0.0698 | 0.1111 |
| Jun | 0.2269 | -0.0885 | 0.2555 | 0.0067 | -0.1657 | -0.1356 |
| Jul | -0.3135 | 0.2299 | 0.1071 | 0.1338 | 0.0585 | -0.1250 |
| Aug | -0.9825 | 0.0565 | -0.4173 | -0.2561 | 0.0341 | -0.1393 |
| Sep | 0.3782 | -0.1831 | -0.0730 | -0.1715 | -0.0351 | 0.0000 |
| Oct | -0.0855 | -0.1594 | -0.3514 | 0.0000 | -0.1467 | -0.1139 |
| Nov | 0.6018 | 0.1800 | 0.1847 | 0.1184 | -0.1863 | 0.2805 |
| Dec | 0.3466 | -0.0221 | 0.1895 | 0.0952 | 0.2073 | 0.1925 |

| BS-theo-75 | LT=1 | LT=2 | LT=3 | LT=4 | LT=5 | LT=6 |
|---|---|---|---|---|---|---|
| Jan | 0.0581 | 0.1200 | 0.1010 | 0.1680 | 0.1240 | 0.1320 |
| Feb | 0.1150 | 0.1330 | 0.1900 | 0.1680 | 0.2040 | 0.1350 |
| Mar | 0.0799 | 0.1250 | 0.1440 | 0.2020 | 0.1650 | 0.1810 |
| Apr | 0.0446 | 0.0997 | 0.1320 | 0.1130 | 0.1440 | 0.1240 |
| May | 0.0782 | 0.1590 | 0.2230 | 0.1730 | 0.1730 | 0.2170 |
| Jun | 0.1440 | 0.1160 | 0.1430 | 0.2010 | 0.1760 | 0.2260 |
| Jul | 0.1630 | 0.1530 | 0.1280 | 0.1550 | 0.2000 | 0.1830 |
| Aug | 0.1060 | 0.2020 | 0.1850 | 0.1860 | 0.1590 | 0.1720 |
| Sep | 0.1460 | 0.1990 | 0.2160 | 0.1910 | 0.2420 | 0.1820 |
| Oct | 0.0744 | 0.1810 | 0.1610 | 0.1800 | 0.1750 | 0.2070 |
| Nov | 0.0532 | 0.1250 | 0.1560 | 0.1890 | 0.1990 | 0.1910 |
| Dec | 0.0898 | 0.1960 | 0.1780 | 0.1420 | 0.1920 | 0.2360 |

| BSS-theo-75 | LT=1 | LT=2 | LT=3 | LT=4 | LT=5 | LT=6 |
|---|---|---|---|---|---|---|
| Jan | 0.2465 | -0.0426 | 0.2808 | -0.1213 | 0.1237 | 0.1170 |
| Feb | -0.1102 | 0.2362 | 0.0715 | 0.1271 | -0.2615 | 0.0098 |
| Mar | 0.0766 | -0.0243 | -0.0820 | -0.1220 | -0.1225 | -0.1247 |
| Apr | 0.1328 | 0.0570 | -0.0368 | 0.0460 | 0.0477 | 0.3242 |
| May | 0.3252 | 0.0419 | -0.2572 | -0.0096 | 0.0033 | -0.1704 |
| Jun | -0.5637 | 0.2542 | 0.1647 | -0.0211 | 0.1573 | -0.1058 |
| Jul | -0.7845 | -0.3406 | -0.1679 | -0.0172 | -0.1863 | -0.1923 |
| Aug | -1.2787 | -0.5281 | -0.0986 | -0.0769 | 0.0735 | 0.0679 |
| Sep | 0.0796 | -0.0351 | 0.0028 | 0.1571 | -0.0695 | 0.1338 |
| Oct | 0.4470 | -0.0306 | 0.0844 | 0.0203 | 0.0549 | -0.1355 |
| Nov | 0.1293 | 0.1318 | 0.1074 | -0.1032 | -0.0855 | -0.0269 |
| Dec | 0.2625 | -0.0468 | -0.0796 | 0.2913 | 0.0337 | -0.2007 |

| BS-act-25 | LT=1 | LT=2 | LT=3 | LT=4 | LT=5 | LT=6 |
|---|---|---|---|---|---|---|
| Jan | 0.3850 | 0.3040 | 0.2810 | 0.2720 | 0.2360 | 0.2170 |
| Feb | 0.3370 | 0.2580 | 0.2770 | 0.2710 | 0.2780 | 0.2460 |
| Mar | 0.2340 | 0.2510 | 0.2440 | 0.2170 | 0.2750 | 0.2800 |
| Apr | 0.2040 | 0.2470 | 0.2590 | 0.2220 | 0.2150 | 0.2450 |
| May | 0.2550 | 0.1820 | 0.2520 | 0.2690 | 0.2110 | 0.2260 |
| Jun | 0.2650 | 0.1990 | 0.1710 | 0.2310 | 0.2500 | 0.2220 |
| Jul | 0.2720 | 0.2390 | 0.1950 | 0.2180 | 0.2610 | 0.2730 |
| Aug | 0.2050 | 0.2470 | 0.2360 | 0.1730 | 0.1970 | 0.2620 |
| Sep | 0.2740 | 0.1700 | 0.2410 | 0.2340 | 0.2090 | 0.2000 |
| Oct | 0.2590 | 0.2350 | 0.1850 | 0.1920 | 0.2120 | 0.2030 |
| Nov | 0.2330 | 0.2470 | 0.2400 | 0.2280 | 0.2720 | 0.2420 |
| Dec | 0.2850 | 0.2390 | 0.2720 | 0.2460 | 0.2180 | 0.2480 |

| BSS-act-25 | LT=1 | LT=2 | LT=3 | LT=4 | LT=5 | LT=6 |
|---|---|---|---|---|---|---|
| Jan | -0.2031 | -0.1515 | -0.1020 | -0.2035 | -0.0926 | -0.0188 |
| Feb | -0.1013 | 0.0618 | -0.0778 | -0.1245 | -0.1830 | -0.1884 |
| Mar | 0.1460 | -0.0203 | 0.0279 | 0.1033 | -0.1044 | -0.1667 |
| Apr | 0.2444 | -0.0786 | -0.1881 | -0.0091 | -0.0047 | -0.0987 |
| May | -0.2687 | 0.0990 | -0.2990 | -0.3058 | -0.0603 | -0.1024 |
| Jun | -0.1134 | -0.1180 | 0.1047 | -0.3158 | -0.1907 | -0.1503 |
| Jul | -0.0924 | -0.1950 | -0.0428 | -0.0900 | -0.2794 | -0.3188 |
| Aug | 0.0238 | -0.1706 | -0.1800 | 0.0281 | -0.0535 | -0.2780 |
| Sep | -0.1138 | 0.0503 | -0.2684 | -0.2251 | -0.2011 | -0.0870 |
| Oct | -0.0117 | -0.1990 | 0.0054 | 0.0634 | -0.0600 | 0.0333 |
| Nov | 0.0754 | -0.0601 | -0.0959 | 0.0044 | -0.1878 | -0.1000 |
| Dec | -0.3318 | 0.0245 | -0.1982 | -0.1442 | 0.0000 | -0.0736 |

| BS-act-75 | LT=1 | LT=2 | LT=3 | LT=4 | LT=5 | LT=6 |
|---|---|---|---|---|---|---|
| Jan | 0.1790 | 0.2960 | 0.2460 | 0.2810 | 0.2300 | 0.2200 |
| Feb | 0.2500 | 0.2900 | 0.3020 | 0.2810 | 0.2600 | 0.2470 |
| Mar | 0.1650 | 0.1500 | 0.1620 | 0.2210 | 0.2210 | 0.2500 |
| Apr | 0.0810 | 0.1250 | 0.1570 | 0.1380 | 0.1500 | 0.1740 |
| May | 0.0843 | 0.1410 | 0.2600 | 0.2140 | 0.1730 | 0.2300 |
| Jun | 0.1870 | 0.1590 | 0.1980 | 0.2310 | 0.2140 | 0.2500 |
| Jul | 0.1810 | 0.2310 | 0.2010 | 0.1850 | 0.2790 | 0.2490 |
| Aug | 0.1370 | 0.2379 | 0.1910 | 0.2400 | 0.1950 | 0.2450 |
| Sep | 0.1340 | 0.2420 | 0.2160 | 0.2820 | 0.2600 | 0.2180 |
| Oct | 0.1530 | 0.1750 | 0.1970 | 0.1800 | 0.2180 | 0.2010 |
| Nov | 0.1320 | 0.1500 | 0.1440 | 0.2190 | 0.1930 | 0.1660 |
| Dec | 0.1020 | 0.1840 | 0.1720 | 0.1360 | 0.1920 | 0.2300 |

| BSS-act-75 | LT=1 | LT=2 | LT=3 | LT=4 | LT=5 | LT=6 |
|---|---|---|---|---|---|---|
| Jan | 0.2694 | 0.1084 | 0.1246 | -0.0849 | -0.0222 | 0.0135 |
| Feb | 0.0157 | 0.0169 | -0.1353 | -0.2165 | -0.3684 | -0.1875 |
| Mar | -0.1224 | 0.1071 | 0.1980 | -0.1106 | -0.0728 | -0.3736 |
| Apr | 0.2358 | -0.0350 | -0.0195 | 0.0548 | 0.1525 | 0.1212 |
| May | 0.1816 | 0.1076 | -0.4130 | -0.1383 | 0.0170 | -0.1386 |
| Jun | -0.3551 | 0.2330 | 0.0435 | 0.0435 | -0.1608 | -0.2755 |
| Jul | -0.3407 | -0.4808 | -0.1292 | 0.0657 | -0.2857 | -0.1318 |
| Aug | -0.0379 | -0.1559 | 0.0905 | -0.1111 | 0.0930 | -0.0938 |
| Sep | 0.2071 | -0.1805 | 0.0886 | -0.1605 | -0.0526 | 0.0480 |
| Oct | -0.2143 | -0.0479 | -0.1939 | -0.0286 | -0.1658 | -0.1044 |
| Nov | 0.0000 | -0.0563 | 0.1111 | -0.3273 | -0.0663 | 0.0879 |
| Dec | 0.2093 | 0.0160 | -0.0424 | 0.3131 | 0.0000 | -0.2105 |

| BS-A/T-25 | LT=1 | LT=2 | LT=3 | LT=4 | LT=5 | LT=6 |
|---|---|---|---|---|---|---|
| Jan | 4.86 | 21.48 | 46.62 | 42.28 | 83.90 | 73.73 |
| Feb | 6.08 | 18.91 | 31.99 | 51.29 | 55.04 | 77.24 |
| Mar | 18.59 | 16.25 | 34.10 | 45.16 | 52.00 | 73.21 |
| Apr | 36.08 | 54.66 | 62.55 | 66.22 | 61.86 | 84.49 |
| May | 27.53 | 67.46 | 63.94 | 75.83 | 63.72 | |
| Jun | 30.34 | 61.81 | 59.65 | 64.50 | 78.80 | 90.54 |
| Jul | 38.97 | 37.70 | 64.10 | 62.39 | 61.69 | 75.82 |
| Aug | 66.34 | 47.37 | 83.47 | 119.08 | 86.29 | 87.40 |
| Sep | 43.80 | 148.24 | 103.73 | 119.66 | 112.92 | 107.50 |
| Oct | 63.71 | 68.09 | 108.11 | 76.56 | 81.13 | 86.70 |
| Nov | 15.62 | 49.80 | 53.33 | 58.77 | 70.22 | 48.76 |
| Dec | 26.60 | 58.16 | 45.59 | 61.79 | 59.63 | 52.42 |

| BS-A/T-75 | LT=1 | LT=2 | LT=3 | LT=4 | LT=5 | LT=6 |
|---|---|---|---|---|---|---|
| Jan | 32.46 | 40.54 | 41.06 | 59.79 | 53.91 | 60.00 |
| Feb | 46.00 | 45.86 | 62.91 | 59.79 | 78.46 | 54.66 |
| Mar | 48.42 | 83.33 | 88.89 | 91.40 | 74.66 | 72.40 |
| Apr | 55.06 | 67.36 | 84.08 | 81.88 | 96.00 | 71.26 |
| May | 92.76 | 112.77 | 85.77 | 80.84 | 100.00 | 94.35 |
| Jun | 77.01 | 73.42 | 72.22 | 87.01 | 82.24 | 90.40 |
| Jul | 90.06 | 66.23 | 63.68 | 83.78 | 71.68 | 73.49 |
| Aug | 77.37 | 93.95 | 96.86 | 77.50 | 81.54 | 70.20 |
| Sep | 108.96 | 82.23 | 100.00 | 67.73 | 93.08 | 83.49 |
| Oct | 48.63 | 103.43 | 81.73 | 100.00 | 80.28 | 102.99 |
| Nov | 40.30 | 83.33 | 108.33 | 86.30 | 103.11 | 115.06 |
| Dec | 88.04 | 106.52 | 103.49 | 104.41 | 100.00 | 102.61 |

**Table 17: Skill scores in the Mississippi**





| BS-theo-25 | LT=1 | LT=2 | LT=3 | LT=4 | LT=5 | LT=6 |
|---|---|---|---|---|---|---|
| Jan | 0.0683 | 0.1330 | 0.1580 | 0.1530 | 0.1480 | 0.1720 |
| Feb | 0.0607 | 0.1370 | 0.1670 | 0.1860 | 0.1820 | 0.1840 |
| Mar | 0.0824 | 0.1400 | 0.1180 | 0.1230 | 0.1580 | 0.1500 |
| Apr | 0.0529 | 0.2070 | 0.1940 | 0.1470 | 0.2020 | 0.1450 |
| May | 0.0931 | 0.2300 | 0.2170 | 0.1890 | 0.1870 | 0.1950 |
| Jun | 0.0678 | 0.1910 | 0.2060 | 0.2560 | 0.2070 | 0.2230 |
| Jul | 0.0821 | 0.1470 | 0.1770 | 0.2690 | 0.2010 | 0.2480 |
| Aug | 0.0289 | 0.1190 | 0.1500 | 0.1280 | 0.1930 | 0.1800 |
| Sep | 0.0521 | 0.0898 | 0.1310 | 0.1520 | 0.1610 | 0.1710 |
| Oct | 0.0322 | 0.0804 | 0.0813 | 0.1250 | 0.1580 | 0.1380 |
| Nov | 0.0725 | 0.1140 | 0.0972 | 0.1020 | 0.1730 | 0.1630 |
| Dec | 0.0873 | 0.1330 | 0.1510 | 0.1360 | 0.1350 | 0.1340 |

| BSS-theo-25 | LT=1 | LT=2 | LT=3 | LT=4 | LT=5 | LT=6 |
|---|---|---|---|---|---|---|
| Jan | 0.3791 | -0.0556 | -0.3621 | -0.1679 | -0.1563 | -0.3231 |
| Feb | 0.2892 | 0.0284 | -0.0570 | -0.1553 | -0.1447 | -0.1572 |
| Mar | 0.2226 | 0.0141 | 0.2133 | 0.1087 | -0.1049 | -0.0274 |
| Apr | 0.3906 | -0.3800 | -0.1279 | 0.0329 | -0.4638 | -0.0284 |
| May | 0.3350 | -0.2921 | -0.1244 | 0.1870 | 0.0053 | -0.0104 |
| Jun | 0.3604 | 0.0591 | 0.0096 | -0.2308 | -0.0561 | -0.0985 |
| Jul | 0.1951 | 0.1503 | 0.0635 | -0.3317 | 0.0000 | 0.0374 |
| Aug | 0.6745 | 0.0985 | -0.0638 | 0.2644 | -0.0782 | 0.0374 |
| Sep | 0.2058 | 0.3145 | 0.2108 | 0.1111 | 0.1006 | 0.0904 |
| Oct | 0.5069 | 0.1057 | 0.3978 | 0.2604 | 0.0971 | 0.2541 |
| Nov | 0.1686 | -0.0556 | 0.3057 | 0.3333 | -0.0812 | 0.0355 |
| Dec | 0.0385 | 0.0634 | 0.0382 | 0.1807 | 0.2241 | 0.2872 |

| BS-theo-75 | LT=1 | LT=2 | LT=3 | LT=4 | LT=5 | LT=6 |
|---|---|---|---|---|---|---|
| Jan | 0.1310 | 0.1910 | 0.2170 | 0.2560 | 0.2640 | 0.2720 |
| Feb | 0.0969 | 0.1820 | 0.1800 | 0.1790 | 0.1420 | 0.1980 |
| Mar | 0.1180 | 0.1710 | 0.1710 | 0.1900 | 0.1760 | 0.1980 |
| Apr | 0.1340 | 0.1610 | 0.1420 | 0.1810 | 0.1670 | 0.1660 |
| May | 0.0485 | 0.1660 | 0.2060 | 0.2320 | 0.2010 | 0.2570 |
| Jun | 0.1360 | 0.1770 | 0.2240 | 0.2270 | 0.2350 | 0.2280 |
| Jul | 0.0931 | 0.1450 | 0.1640 | 0.1680 | 0.2010 | 0.1890 |
| Aug | 0.0105 | 0.0862 | 0.1280 | 0.1240 | 0.1340 | 0.1990 |
| Sep | 0.0413 | 0.0945 | 0.1150 | 0.0821 | 0.1010 | 0.1390 |
| Oct | 0.0488 | 0.1120 | 0.1570 | 0.2140 | 0.2070 | 0.2050 |
| Nov | 0.0548 | 0.0931 | 0.1380 | 0.1300 | 0.1740 | 0.1010 |
| Dec | 0.1710 | 0.2180 | 0.1990 | 0.2310 | 0.2670 | 0.2370 |

| BSS-theo-75 | LT=1 | LT=2 | LT=3 | LT=4 | LT=5 | LT=6 |
|---|---|---|---|---|---|---|
| Jan | 0.1208 | 0.0683 | -0.0333 | -0.1907 | -0.2512 | -0.3333 |
| Feb | 0.1191 | -0.0581 | 0.0674 | 0.0725 | 0.2604 | -0.0313 |
| Mar | 0.1511 | -0.0667 | 0.1231 | 0.0404 | 0.0928 | 0.0149 |
| Apr | -0.5691 | 0.0852 | 0.1932 | -0.0056 | 0.1480 | 0.1354 |
| May | 0.5198 | 0.1616 | -0.0404 | -0.1485 | -0.0308 | -0.2786 |
| Jun | -0.3333 | -0.0473 | -0.0980 | -0.0861 | -0.1353 | -0.1400 |
| Jul | -0.0751 | -0.0902 | -0.0380 | 0.1472 | -0.0579 | -0.0161 |
| Aug | 0.5977 | 0.3930 | 0.0725 | 0.1842 | 0.2910 | -0.0311 |
| Sep | 0.5130 | -0.0407 | 0.1353 | 0.3158 | 0.2463 | 0.2235 |
| Oct | 0.4948 | 0.2911 | 0.0925 | -0.0918 | -0.1374 | -0.0963 |
| Nov | 0.1397 | 0.4358 | 0.1534 | 0.2614 | -0.0235 | 0.3804 |
| Dec | -0.1250 | -0.1295 | 0.0701 | -0.0452 | -0.2361 | -0.1505 |

| BS-act-25 | LT=1 | LT=2 | LT=3 | LT=4 | LT=5 | LT=6 |
|---|---|---|---|---|---|---|
| Jan | 0.6740 | 0.6530 | 0.6280 | 0.5730 | 0.6060 | 0.6040 |
| Feb | 0.6410 | 0.6220 | 0.6180 | 0.6130 | 0.6580 | 0.5660 |
| Mar | 0.7490 | 0.5880 | 0.5850 | 0.5750 | 0.6220 | 0.5830 |
| Apr | 0.7130 | 0.6190 | 0.6300 | 0.5900 | 0.5940 | 0.5870 |
| May | 0.6510 | 0.6240 | 0.6350 | 0.5840 | 0.5870 | 0.6340 |
| Jun | 0.7280 | 0.5790 | 0.5330 | 0.6260 | 0.6530 | 0.6530 |
| Jul | 0.7000 | 0.6200 | 0.5770 | 0.5480 | 0.6010 | 0.6870 |
| Aug | 0.6470 | 0.6200 | 0.6110 | 0.5760 | 0.6050 | 0.6890 |
| Sep | 0.7790 | 0.6290 | 0.6400 | 0.6070 | 0.5670 | 0.5590 |
| Oct | 0.7050 | 0.6680 | 0.6330 | 0.6220 | 0.6630 | 0.6170 |
| Nov | 0.7690 | 0.6710 | 0.6610 | 0.5990 | 0.5790 | 0.5870 |
| Dec | 0.7660 | 0.6600 | 0.6180 | 0.6150 | 0.5900 | 0.5590 |

| BSS-act-25 | LT=1 | LT=2 | LT=3 | LT=4 | LT=5 | LT=6 |
|---|---|---|---|---|---|---|
| Jan | 0.0729 | -0.0722 | -0.0572 | -0.0035 | -0.0236 | -0.0523 |
| Feb | -0.0423 | -0.0780 | -0.0249 | 0.0103 | 0.0053 | 0.0358 |
| Mar | -0.1031 | 0.0652 | 0.0051 | 0.0103 | -0.1068 | -0.0697 |
| Apr | -0.0439 | -0.0351 | -0.0518 | 0.0000 | 0.0034 | 0.0034 |
| May | 0.0327 | -0.0487 | -0.0781 | -0.0069 | -0.0389 | -0.1590 |
| Jun | -0.0722 | 0.0810 | 0.0550 | -0.1041 | -0.0543 | -0.2274 |
| Jul | -0.0703 | -0.0197 | 0.0303 | 0.0036 | -0.1028 | -0.2817 |
| Aug | 0.0137 | -0.0578 | -0.0719 | 0.0287 | -0.0558 | -0.2831 |
| Sep | -0.1966 | -0.0501 | -0.1111 | -0.0520 | 0.0374 | 0.0261 |
| Oct | -0.0633 | -0.0969 | -0.1086 | -0.1289 | -0.0698 | -0.0768 |
| Nov | -0.1461 | -0.1000 | -0.1203 | -0.0509 | -0.0585 | -0.0156 |
| Dec | -0.0698 | -0.1149 | -0.0748 | -0.0866 | -0.0573 | 0.0000 |

| BS-act-75 | LT=1 | LT=2 | LT=3 | LT=4 | LT=5 | LT=6 |
|---|---|---|---|---|---|---|
| Jan | 0.6890 | 0.6580 | 0.5340 | 0.6100 | 0.5810 | 0.6010 |
| Feb | 0.7020 | 0.5880 | 0.6160 | 0.5710 | 0.5900 | 0.5330 |
| Mar | 0.7300 | 0.6060 | 0.6010 | 0.5500 | 0.5490 | 0.5650 |
| Apr | 0.7890 | 0.5790 | 0.6090 | 0.5630 | 0.5340 | 0.5640 |
| May | 0.7270 | 0.6450 | 0.5940 | 0.6180 | 0.5520 | 0.5550 |
| Jun | 0.7420 | 0.6680 | 0.5630 | 0.5720 | 0.5890 | 0.5370 |
| Jul | 0.6990 | 0.6360 | 0.6480 | 0.5620 | 0.6010 | 0.5430 |
| Aug | 0.7010 | 0.6620 | 0.6920 | 0.6570 | 0.5710 | 0.5690 |
| Sep | 0.7140 | 0.6100 | 0.6840 | 0.6340 | 0.6650 | 0.6120 |
| Oct | 0.6790 | 0.6540 | 0.6440 | 0.6370 | 0.6530 | 0.6530 |
| Nov | 0.6850 | 0.6390 | 0.6110 | 0.6090 | 0.6100 | 0.6400 |
| Dec | 0.7110 | 0.6180 | 0.6300 | 0.5950 | 0.5940 | 0.5640 |

| BSS-act-75 | LT=1 | LT=2 | LT=3 | LT=4 | LT=5 | LT=6 |
|---|---|---|---|---|---|---|
| Jan | -0.0600 | -0.1729 | 0.0549 | -0.1031 | -0.0525 | -0.1007 |
| Feb | -0.0541 | 0.0167 | -0.0903 | -0.0178 | -0.0926 | -0.0038 |
| Mar | -0.3082 | -0.0341 | 0.0653 | 0.0143 | 0.0000 | -0.0386 |
| Apr | -0.2977 | -0.0017 | -0.0760 | 0.0770 | 0.0309 | -0.0387 |
| May | -0.2097 | -0.1857 | -0.0259 | -0.0528 | 0.0195 | 0.0704 |
| Jun | -0.1346 | -0.2101 | 0.0035 | 0.0272 | -0.0120 | 0.0709 |
| Jul | -0.0704 | -0.0325 | -0.1250 | -0.0108 | -0.0562 | 0.0638 |
| Aug | -0.0370 | -0.0541 | -0.1002 | -0.1005 | -0.0555 | -0.0179 |
| Sep | -0.1591 | 0.0033 | -0.0979 | -0.0428 | -0.1139 | -0.0373 |
| Oct | -0.0104 | -0.0615 | -0.0864 | -0.0842 | -0.0983 | -0.1873 |
| Nov | -0.1230 | -0.0424 | -0.0286 | -0.0235 | -0.0445 | -0.0959 |
| Dec | -0.1694 | -0.0880 | -0.1190 | -0.0420 | -0.0224 | 0.0521 |

| BS-A/T-25 | LT=1 | LT=2 | LT=3 | LT=4 | LT=5 | LT=6 |
|---|---|---|---|---|---|---|
| Jan | 10.13 | 20.37 | 25.16 | 26.70 | 24.42 | 28.48 |
| Feb | 9.47 | 22.03 | 27.02 | 30.34 | 27.66 | 32.51 |
| Mar | 11.00 | 23.81 | 20.17 | 21.39 | 25.40 | 25.73 |
| Apr | 7.42 | 33.44 | 30.79 | 24.92 | 34.01 | 24.70 |
| May | 14.30 | 36.86 | 34.17 | 32.36 | 31.86 | 30.30 |
| Jun | 9.31 | 32.99 | 38.65 | 40.89 | 34.39 | 34.15 |
| Jul | 11.73 | 23.71 | 30.68 | 49.09 | 33.44 | 34.10 |
| Aug | 4.47 | 19.13 | 24.55 | 22.22 | 31.90 | 26.12 |
| Sep | 6.69 | 14.28 | 20.47 | 25.04 | 28.40 | 30.59 |
| Oct | 4.57 | 12.04 | 12.84 | 20.10 | 25.77 | 22.37 |
| Nov | 9.43 | 16.99 | 14.70 | 17.03 | 29.88 | 27.77 |
| Dec | 11.40 | 20.15 | 24.43 | 22.11 | 22.88 | 23.97 |

| BS-A/T-75 | LT=1 | LT=2 | LT=3 | LT=4 | LT=5 | LT=6 |
|---|---|---|---|---|---|---|
| Jan | 19.01 | 29.03 | 40.64 | 41.97 | 45.44 | 45.26 |
| Feb | 13.80 | 30.95 | 29.22 | 31.35 | 24.07 | 37.15 |
| Mar | 16.16 | 31.68 | 28.45 | 34.55 | 32.06 | 35.04 |
| Apr | 16.98 | 27.81 | 23.32 | 32.15 | 31.27 | 29.43 |
| May | 6.67 | 25.74 | 34.68 | 37.54 | 36.41 | 46.31 |
| Jun | 18.33 | 26.50 | 39.79 | 39.69 | 39.90 | 42.46 |
| Jul | 13.32 | 22.80 | 25.31 | 29.89 | 33.44 | 34.81 |
| Aug | 1.50 | 13.02 | 18.50 | 18.87 | 23.47 | 34.97 |
| Sep | 5.78 | 15.49 | 16.81 | 12.95 | 15.19 | 22.71 |
| Oct | 7.19 | 17.53 | 24.01 | 33.23 | 32.50 | 31.39 |
| Nov | 8.00 | 14.57 | 22.59 | 21.35 | 28.52 | 15.78 |
| Dec | 24.05 | 35.28 | 31.59 | 38.82 | 44.95 | 42.02 |

**Table 18: Skill scores in the Murray**



| BS-theo-25 | LT=1 | LT=2 | LT=3 | LT=4 | LT=5 | LT=6 |
|---|---|---|---|---|---|---|
| Jan | 0.1350 | 0.1750 | 0.1920 | 0.1610 | 0.2070 | 0.2080 |
| Feb | 0.1410 | 0.1300 | 0.1170 | 0.2120 | 0.1630 | 0.1470 |
| Mar | 0.0758 | 0.1870 | 0.1890 | 0.1840 | 0.2290 | 0.2200 |
| Apr | 0.0777 | 0.1240 | 0.1580 | 0.1990 | 0.1740 | 0.1780 |
| May | 0.0642 | 0.0755 | 0.0835 | 0.1390 | 0.1820 | 0.1740 |
| Jun | 0.0727 | 0.0468 | 0.0741 | 0.0777 | 0.1260 | 0.1710 |
| Jul | 0.0636 | 0.0468 | 0.0325 | 0.0309 | 0.0584 | 0.1100 |
| Aug | 0.0091 | 0.0435 | 0.0526 | 0.0366 | 0.0262 | 0.0730 |
| Sep | 0.0405 | 0.0639 | 0.0694 | 0.0689 | 0.0747 | 0.0829 |
| Oct | 0.0449 | 0.0730 | 0.1360 | 0.0851 | 0.1030 | 0.1130 |
| Nov | 0.1160 | 0.1480 | 0.1830 | 0.1780 | 0.1730 | 0.1710 |
| Dec | 0.0741 | 0.1120 | 0.2220 | 0.2250 | 0.2310 | 0.1780 |

| BSS-theo-25 | LT=1 | LT=2 | LT=3 | LT=4 | LT=5 | LT=6 |
|---|---|---|---|---|---|---|
| Jan | -0.7419 | -0.4344 | -0.2152 | 0.0747 | -0.0952 | -0.0612 |
| Feb | -0.4491 | 0.2073 | 0.3237 | -0.1648 | 0.1189 | 0.2423 |
| Mar | -0.0813 | -0.1131 | 0.0597 | 0.0754 | -0.1744 | -0.1765 |
| Apr | 0.1170 | -0.0420 | -0.1367 | -0.1637 | -0.0807 | -0.1410 |
| May | -0.0078 | 0.1251 | 0.1650 | 0.1094 | -0.0643 | 0.0169 |
| Jun | -0.0491 | -0.0935 | 0.0738 | 0.1150 | -0.1351 | -0.0755 |
| Jul | -1.8649 | -0.4579 | -0.4509 | 0.2463 | -0.0523 | -0.2009 |
| Aug | 0.6305 | -0.4846 | -0.9627 | -1.5068 | 0.3515 | -0.3201 |
| Sep | -0.4311 | 0.0289 | -0.0420 | 0.0826 | -0.0905 | -0.2281 |
| Oct | 0.3288 | 0.2091 | -0.3204 | 0.2600 | 0.1488 | -0.0180 |
| Nov | -0.2554 | -0.0068 | -0.1091 | -0.0595 | -0.0176 | 0.0229 |
| Dec | 0.4344 | 0.2583 | -0.3962 | -0.3804 | -0.3750 | -0.0533 |

| BS-theo-75 | LT=1 | LT=2 | LT=3 | LT=4 | LT=5 | LT=6 |
|---|---|---|---|---|---|---|
| Jan | 0.1390 | 0.1930 | 0.1930 | 0.2280 | 0.2300 | 0.2060 |
| Feb | 0.1280 | 0.1890 | 0.2090 | 0.1880 | 0.2280 | 0.2410 |
| Mar | 0.0961 | 0.1410 | 0.1420 | 0.1800 | 0.2090 | 0.1990 |
| Apr | 0.0298 | 0.1440 | 0.1910 | 0.1990 | 0.2190 | 0.2030 |
| May | 0.0248 | 0.0815 | 0.1230 | 0.1710 | 0.1960 | 0.1940 |
| Jun | 0.0482 | 0.0441 | 0.0562 | 0.0829 | 0.1360 | 0.1830 |
| Jul | 0.0537 | 0.0369 | 0.0485 | 0.0601 | 0.0994 | 0.1420 |
| Aug | 0.0295 | 0.0534 | 0.0339 | 0.0529 | 0.0570 | 0.0981 |
| Sep | 0.0752 | 0.0989 | 0.1090 | 0.0826 | 0.1120 | 0.0898 |
| Oct | 0.0449 | 0.1110 | 0.1040 | 0.1390 | 0.1310 | 0.1460 |
| Nov | 0.0766 | 0.1150 | 0.1740 | 0.1510 | 0.2010 | 0.1470 |
| Dec | 0.0906 | 0.1150 | 0.1210 | 0.1800 | 0.1770 | 0.2050 |

| BSS-theo-75 | LT=1 | LT=2 | LT=3 | LT=4 | LT=5 | LT=6 |
|---|---|---|---|---|---|---|
| Jan | 0.0671 | -0.1221 | -0.0663 | -0.0317 | -0.0748 | -0.0049 |
| Feb | 0.0448 | -0.0161 | -0.0773 | 0.0600 | -0.1400 | -0.2172 |
| Mar | -0.0031 | 0.0140 | 0.2487 | 0.0754 | -0.0773 | -0.0311 |
| Apr | 0.1791 | -0.2414 | -0.0106 | -0.0258 | -0.1231 | -0.0150 |
| May | -1.9880 | 0.1519 | -0.0424 | 0.1094 | -0.0370 | 0.0490 |
| Jun | -7.2253 | -0.2857 | 0.2248 | 0.1471 | 0.1338 | -0.0055 |
| Jul | -0.3392 | -0.5061 | 0.1019 | 0.3396 | 0.2412 | 0.1598 |
| Aug | 0.1424 | -0.2654 | 0.0960 | 0.2277 | 0.4673 | 0.2891 |
| Sep | -0.2071 | -0.0334 | -0.0187 | 0.1902 | 0.0894 | 0.3145 |
| Oct | 0.2554 | 0.0000 | 0.2121 | -0.0611 | 0.0775 | 0.0068 |
| Nov | 0.0065 | 0.1353 | -0.1226 | -0.0134 | -0.2722 | 0.0392 |
| Dec | -0.1828 | 0.0873 | 0.1933 | -0.0112 | -0.0351 | -0.1988 |

| BS-act-25 | LT=1 | LT=2 | LT=3 | LT=4 | LT=5 | LT=6 |
|---|---|---|---|---|---|---|
| Jan | 0.1950 | 0.1880 | 0.1920 | 0.1540 | 0.1570 | 0.1390 |
| Feb | 0.2790 | 0.1780 | 0.1990 | 0.2120 | 0.2200 | 0.2100 |
| Mar | 0.2030 | 0.1620 | 0.1470 | 0.1590 | 0.1980 | 0.1950 |
| Apr | 0.1560 | 0.1420 | 0.1110 | 0.1570 | 0.1520 | 0.1690 |
| May | 0.0763 | 0.0391 | 0.0410 | 0.1200 | 0.1330 | 0.1480 |
| Jun | 0.0848 | 0.0711 | 0.0499 | 0.0656 | 0.1200 | 0.1650 |
| Jul | 0.1850 | 0.1560 | 0.1290 | 0.1220 | 0.1190 | 0.1470 |
| Aug | 0.1850 | 0.1590 | 0.1440 | 0.1520 | 0.1290 | 0.1340 |
| Sep | 0.1010 | 0.1180 | 0.1180 | 0.1050 | 0.1050 | 0.1130 |
| Oct | 0.1660 | 0.1700 | 0.1960 | 0.1640 | 0.1690 | 0.1670 |
| Nov | 0.1830 | 0.2210 | 0.2200 | 0.2390 | 0.2280 | 0.2250 |
| Dec | 0.2500 | 0.1670 | 0.2100 | 0.2130 | 0.1820 | 0.1420 |

| BSS-act-25 | LT=1 | LT=2 | LT=3 | LT=4 | LT=5 | LT=6 |
|---|---|---|---|---|---|---|
| Jan | -0.1471 | -0.4574 | -0.3427 | -0.2833 | -0.1056 | 0.0795 |
| Feb | -0.8477 | 0.0481 | 0.0149 | -0.1459 | -0.1765 | -0.0714 |
| Mar | -0.5263 | -0.1329 | 0.1600 | 0.1167 | -0.1250 | -0.0955 |
| Apr | -0.1818 | -0.1008 | 0.1190 | -0.0064 | 0.0732 | -0.0120 |
| May | -1.2116 | 0.1988 | 0.3212 | -0.2793 | 0.0952 | 0.0000 |
| Jun | -0.0928 | -0.2387 | 0.0567 | 0.1094 | -0.3001 | -0.1957 |
| Jul | -0.4453 | -0.1304 | -0.1518 | -0.0702 | 0.0916 | -0.0068 |
| Aug | -0.0221 | -0.1197 | -0.0511 | -0.0556 | -0.1835 | -0.3522 |
| Sep | -0.1663 | -0.0536 | -0.1346 | -0.0096 | -0.0096 | -0.1449 |
| Oct | -0.0921 | -0.0692 | -0.2327 | -0.0380 | -0.0497 | -0.0774 |
| Nov | 0.0804 | -0.2278 | -0.1702 | -0.2646 | -0.2459 | -0.1968 |
| Dec | -0.9084 | 0.0402 | -0.4384 | -0.3654 | -0.1818 | 0.1013 |

| BS-act-75 | LT=1 | LT=2 | LT=3 | LT=4 | LT=5 | LT=6 |
|---|---|---|---|---|---|---|
| Jan | 0.3000 | 0.2840 | 0.2650 | 0.2880 | 0.1950 | 0.2030 |
| Feb | 0.2940 | 0.2340 | 0.2940 | 0.2480 | 0.2880 | 0.2500 |
| Mar | 0.2480 | 0.2600 | 0.2390 | 0.2430 | 0.2590 | 0.2550 |
| Apr | 0.3630 | 0.3130 | 0.2350 | 0.2660 | 0.2690 | 0.2660 |
| May | 0.4520 | 0.3570 | 0.3260 | 0.2240 | 0.2420 | 0.2540 |
| Jun | 0.3880 | 0.3200 | 0.3230 | 0.3370 | 0.2300 | 0.2440 |
| Jul | 0.3900 | 0.3790 | 0.3790 | 0.3480 | 0.3020 | 0.2210 |
| Aug | 0.3540 | 0.3290 | 0.3160 | 0.3290 | 0.2900 | 0.2770 |
| Sep | 0.3510 | 0.3320 | 0.3480 | 0.3460 | 0.3210 | 0.2930 |
| Oct | 0.3210 | 0.2960 | 0.3070 | 0.2760 | 0.3040 | 0.2700 |
| Nov | 0.3710 | 0.3060 | 0.2610 | 0.2630 | 0.2290 | 0.2530 |
| Dec | 0.3570 | 0.3030 | 0.2370 | 0.2100 | 0.2550 | 0.1990 |

| BSS-act-75 | LT=1 | LT=2 | LT=3 | LT=4 | LT=5 | LT=6 |
|---|---|---|---|---|---|---|
| Jan | -0.0135 | -0.0717 | -0.1422 | -0.4328 | -0.0052 | 0.0146 |
| Feb | -0.3674 | -0.0174 | -0.2353 | -0.0598 | -0.2308 | -0.0638 |
| Mar | -0.2277 | -0.1304 | 0.0324 | -0.0083 | -0.0747 | -0.0494 |
| Apr | 0.0902 | -0.0981 | -0.1667 | -0.2396 | -0.2036 |  |
| May | 0.0850 | 0.0165 | -0.2636 | -0.1667 | -0.1635 | -0.2959 |
| Jun | -0.0184 | -0.0428 | 0.0415 | -0.2481 | -0.1111 | -0.1091 |
| Jul | -0.0456 | -0.0053 | -0.1590 | -0.2297 | -0.4245 | -0.2416 |
| Aug | -0.0663 | -0.0444 | 0.0336 | -0.1544 | -0.1647 | -0.3781 |
| Sep | -0.0541 | -0.0850 | -0.2518 | -0.2862 | -0.3777 | -0.3952 |
| Oct | 0.0881 | -0.1608 | -0.1370 | -0.1084 | -0.2358 | -0.1203 |
| Nov | -0.1075 | -0.1250 | -0.1348 | -0.0913 | 0.0172 | -0.0720 |
| Dec | -0.0259 | -0.1181 | 0.0207 | 0.0625 | -0.0625 | 0.1459 |

| BS-A/T-25 | LT=1 | LT=2 | LT=3 | LT=4 | LT=5 | LT=6 |
|---|---|---|---|---|---|---|
| Jan | 69.23 | 93.09 | 100.00 | 104.55 | 131.85 | 149.64 |
| Feb | 50.54 | 73.03 | 58.79 | 100.00 | 74.09 | 70.00 |
| Mar | 37.34 | 115.43 | 128.57 | 115.72 | 115.66 | 112.82 |
| Apr | 49.81 | 87.32 | 142.34 | 126.75 | 114.47 | 105.33 |
| May | 84.14 | 193.09 | 203.66 | 115.83 | 136.84 | 117.57 |
| Jun | 85.73 | 65.82 | 148.50 | 118.45 | 105.00 | 103.64 |
| Jul | 34.38 | 30.00 | 25.19 | 25.33 | 49.08 | 74.83 |
| Aug | 4.91 | 27.36 | 36.53 | 24.08 | 20.31 | 54.48 |
| Sep | 40.10 | 54.15 | 58.81 | 65.62 | 71.14 | 73.36 |
| Oct | 27.05 | 42.94 | 69.39 | 51.89 | 60.95 | 67.66 |
| Nov | 63.39 | 66.97 | 83.18 | 74.48 | 75.88 | 76.00 |
| Dec | 29.64 | 67.07 | 105.71 | 105.63 | 126.92 | 125.35 |

| BS-A/T-75 | LT=1 | LT=2 | LT=3 | LT=4 | LT=5 | LT=6 |
|---|---|---|---|---|---|---|
| Jan | 46.33 | 67.96 | 72.83 | 79.17 | 117.95 | 101.48 |
| Feb | 43.54 | 80.77 | 71.09 | 75.81 | 79.17 | 96.40 |
| Mar | 38.75 | 54.23 | 59.41 | 74.07 | 80.69 | 78.04 |
| Apr | 8.21 | 46.01 | 81.28 | 74.81 | 81.41 | 76.32 |
| May | 5.49 | 22.83 | 37.73 | 76.34 | 80.99 | 76.38 |
| Jun | 12.42 | 11.31 | 17.40 | 24.60 | 59.13 | 75.00 |
| Jul | 13.77 | 9.74 | 12.80 | 17.27 | 32.91 | 64.25 |
| Aug | 8.33 | 16.23 | 10.73 | 16.08 | 19.66 | 35.42 |
| Sep | 21.42 | 29.79 | 31.32 | 23.87 | 34.89 | 30.65 |
| Oct | 13.99 | 37.50 | 33.88 | 50.36 | 43.09 | 54.07 |
| Nov | 20.65 | 37.58 | 66.67 | 57.41 | 87.77 | 58.10 |
| Dec | 25.38 | 37.95 | 51.05 | 85.71 | 69.41 | 103.02 |

**Table 19: Skill scores in the Orange River**



| BS-theo-25 | LT=1 | LT=2 | LT=3 | LT=4 | LT=5 | LT=6 |
|---|---|---|---|---|---|---|
| Jan | 0.1110 | 0.2310 | 0.2490 | 0.2500 | 0.1860 | 0.2670 |
| Feb | 0.1240 | 0.1870 | 0.2050 | 0.2220 | 0.2390 | 0.2820 |
| Mar | 0.0493 | 0.1060 | 0.2110 | 0.1940 | 0.1740 | 0.2100 |
| Apr | 0.1120 | 0.1340 | 0.1580 | 0.2400 | 0.2120 | 0.2030 |
| May | 0.0000 | 0.0821 | 0.1660 | 0.1560 | 0.2630 | 0.2250 |
| Jun | 0.0333 | 0.0138 | 0.0994 | 0.1630 | 0.1760 | 0.2580 |
| Jul | 0.0000 | 0.0072 | 0.0168 | 0.0840 | 0.1500 | 0.1690 |
| Aug | 0.0044 | 0.0223 | 0.0278 | 0.0322 | 0.0683 | 0.1400 |
| Sep | 0.0419 | 0.0179 | 0.0132 | 0.0262 | 0.0242 | 0.0868 |
| Oct | 0.1220 | 0.1670 | 0.1390 | 0.1750 | 0.1670 | 0.1690 |
| Nov | 0.0956 | 0.1550 | 0.2320 | 0.2080 | 0.1950 | 0.1580 |
| Dec | 0.1530 | 0.1830 | 0.2360 | 0.2420 | 0.2180 | 0.1720 |

| BSS-theo-25 | LT=1 | LT=2 | LT=3 | LT=4 | LT=5 | LT=6 |
|---|---|---|---|---|---|---|
| Jan | -0.1235 | -0.1667 | -0.2388 | -0.2438 | 0.0837 | -0.3088 |
| Feb | -0.2157 | -0.3551 | -0.2349 | -0.0882 | -0.1602 | -0.3756 |
| Mar | 0.4750 | 0.3072 | -0.2197 | 0.0443 | 0.1212 | -0.0345 |
| Apr | -1.2266 | -0.1261 | 0.0000 | -0.3408 | -0.0547 | -0.0305 |
| May | 1.0000 | 0.2106 | -0.1690 | 0.0983 | -0.2956 | -0.0817 |
| Jun | -1020.472 | 0.1429 | 0.0158 | -0.0724 | -0.0173 | -0.2900 |
| Jul | NA | -2.3774 | 0.2466 | 0.1845 | 0.1071 | 0.0611 |
| Aug | 0.5840 | -0.2816 | -0.0296 | 0.0000 | 0.3369 | 0.1954 |
| Sep | -0.7605 | 0.1864 | 0.5200 | 0.1027 | 0.1626 | 0.0258 |
| Oct | 0.1701 | -0.1438 | 0.0974 | -0.1513 | -0.0987 | -0.1267 |
| Nov | -0.0876 | 0.0882 | -0.3034 | -0.1429 | -0.0714 | 0.1366 |
| Dec | -0.2047 | 0.0985 | -0.1626 | -0.2161 | -0.0955 | 0.1357 |

| BS-theo-75 | LT=1 | LT=2 | LT=3 | LT=4 | LT=5 | LT=6 |
|---|---|---|---|---|---|---|
| Jan | 0.1420 | 0.2060 | 0.2030 | 0.2370 | 0.1960 | 0.2310 |
| Feb | 0.1040 | 0.1470 | 0.2140 | 0.2030 | 0.2270 | 0.2090 |
| Mar | 0.0983 | 0.1230 | 0.1330 | 0.1300 | 0.1560 | 0.1430 |
| Apr | 0.0518 | 0.1200 | 0.1820 | 0.2310 | 0.2280 | 0.1850 |
| May | 0.0275 | 0.1230 | 0.1260 | 0.2070 | 0.2400 | 0.2030 |
| Jun | 0.0000 | 0.0512 | 0.1670 | 0.1290 | 0.2230 | 0.2210 |
| Jul | 0.0556 | 0.0967 | 0.0788 | 0.1420 | 0.1330 | 0.2210 |
| Aug | 0.0769 | 0.0339 | 0.0209 | 0.0576 | 0.0860 | 0.0975 |
| Sep | 0.0554 | 0.0413 | 0.0375 | 0.0433 | 0.0521 | 0.0975 |
| Oct | 0.1160 | 0.1600 | 0.1700 | 0.1650 | 0.1600 | 0.1330 |
| Nov | 0.0986 | 0.1490 | 0.2060 | 0.2210 | 0.1920 | 0.2180 |
| Dec | 0.1550 | 0.1220 | 0.2140 | 0.2090 | 0.2540 | 0.2020 |

| BSS-theo-75 | LT=1 | LT=2 | LT=3 | LT=4 | LT=5 | LT=6 |
|---|---|---|---|---|---|---|
| Jan | -0.5760 | -0.0457 | -0.0201 | -0.1850 | 0.0200 | -0.1436 |
| Feb | -0.2322 | -0.2049 | 0.0000 | -0.0201 | -0.1019 | -0.0195 |
| Mar | -0.2396 | 0.0081 | 0.1133 | 0.2216 | 0.1429 | 0.2353 |
| Apr | 0.0460 | -0.3970 | -0.0833 | -0.3125 | -0.2527 | 0.0054 |
| May | NA | -1.0000 | -0.1053 | -0.2176 | -0.2308 | -0.0573 |
| Jun | 1.0000 | -1.3486 | -0.7199 | -0.0157 | -0.1737 | -0.1333 |
| Jul | -0.9786 | -0.9301 | -0.7395 | -0.5011 | -0.0556 | -0.1755 |
| Aug | -24.3795 | -0.4125 | -0.0097 | -1.5830 | -0.4007 | -0.3265 |
| Sep | -4.8501 | -0.0248 | 0.0602 | -0.0561 | -0.1655 | 0.0152 |
| Oct | -0.0841 | 0.0909 | 0.1005 | 0.1270 | 0.1579 | 0.3073 |
| Nov | 0.0140 | 0.2240 | -0.0564 | -0.1218 | 0.0254 | -0.1010 |
| Dec | -0.3136 | 0.2948 | -0.0863 | -0.0609 | -0.2893 | -0.0254 |

| BS-act-25 | LT=1 | LT=2 | LT=3 | LT=4 | LT=5 | LT=6 |
|---|---|---|---|---|---|---|
| Jan | 0.2620 | 0.2370 | 0.2050 | 0.2250 | 0.2240 | 0.2110 |
| Feb | 0.3900 | 0.4960 | 0.5720 | 0.5570 | 0.5310 | 0.5920 |
| Mar | 0.4640 | 0.4220 | 0.5350 | 0.5080 | 0.4680 | 0.4550 |
| Apr | 0.6270 | 0.5400 | 0.5150 | 0.4880 | 0.4660 | 0.4310 |
| May | 0.6000 | 0.6030 | 0.5290 | 0.5140 | 0.5540 | 0.5730 |
| Jun | 0.3000 | 0.2990 | 0.1960 | 0.2230 | 0.1970 | 0.1910 |
| Jul | 0.4000 | 0.3710 | 0.3500 | 0.2420 | 0.2350 | 0.1910 |
| Aug | 0.3800 | 0.3680 | 0.3610 | 0.3410 | 0.2680 | 0.1940 |
| Sep | 0.2900 | 0.2970 | 0.2980 | 0.2810 | 0.2670 | 0.2440 |
| Oct | 0.2130 | 0.2220 | 0.1570 | 0.1870 | 0.1610 | 0.2000 |
| Nov | 0.2470 | 0.1910 | 0.1530 | 0.1770 | 0.1950 | 0.1640 |
| Dec | 0.2260 | 0.2080 | 0.1570 | 0.1870 | 0.2060 | 0.2260 |

| BSS-act-25 | LT=1 | LT=2 | LT=3 | LT=4 | LT=5 | LT=6 |
|---|---|---|---|---|---|---|
| Jan | -0.1963 | -0.1970 | -0.0459 | -0.1421 | -0.1256 | -0.0603 |
| Feb | 0.1410 | -0.1022 | -0.1327 | -0.0881 | -0.1642 | -0.0364 |
| Mar | -0.0265 | 0.0474 | -0.2442 | -0.1651 | -0.1642 | -0.0364 |
| Apr | -0.1318 | -0.1465 | -0.1444 | -0.1116 | -0.0762 | -0.0564 |
| May | -0.0733 | -0.0415 | -0.0115 | -0.0554 | -0.0573 | -0.0495 |
| Jun | 0.0937 | -0.0136 | 0.2000 | -0.1799 | -0.1257 | 0.0000 |
| Jul | 0.0000 | 0.0313 | 0.0278 | -0.0083 | -0.2176 | -0.0437 |
| Aug | -0.0243 | -0.0308 | -0.0618 | -0.0149 | 0.0759 | -0.0486 |
| Sep | 0.0938 | -0.0839 | -0.0916 | -0.0293 | 0.0361 | 0.0543 |
| Oct | -0.2384 | -0.4051 | 0.0063 | -0.1987 | -0.0321 | -0.2658 |
| Nov | -0.1932 | -0.0977 | 0.0671 | -0.0473 | -0.1471 | 0.0353 |
| Dec | -0.0971 | -0.2023 | 0.1278 | -0.0219 | -0.1135 | -0.2086 |

| BS-act-75 | LT=1 | LT=2 | LT=3 | LT=4 | LT=5 | LT=6 |
|---|---|---|---|---|---|---|
| Jan | 0.4510 | 0.3350 | 0.2880 | 0.3470 | 0.3430 | 0.3150 |
| Feb | 0.6370 | 0.5500 | 0.4650 | 0.4730 | 0.4900 | 0.4790 |
| Mar | 0.2680 | 0.3170 | 0.3030 | 0.3080 | 0.2600 | 0.2470 |
| Apr | 0.2700 | 0.2840 | 0.2890 | 0.2910 | 0.3190 | 0.2640 |
| May | 0.5610 | 0.5770 | 0.4410 | 0.5120 | 0.4940 | 0.3840 |
| Jun | 0.5330 | 0.5420 | 0.5430 | 0.3770 | 0.3770 | 0.3960 |
| Jul | 0.5890 | 0.5450 | 0.5450 | 0.4580 | 0.3510 | 0.3000 |
| Aug | 0.5680 | 0.5240 | 0.5240 | 0.5610 | 0.4440 | 0.3220 |
| Sep | 0.5010 | 0.4810 | 0.4950 | 0.5010 | 0.4670 | 0.4100 |
| Oct | 0.4070 | 0.3910 | 0.3940 | 0.3950 | 0.3900 | 0.3570 |
| Nov | 0.3930 | 0.3700 | 0.4030 | 0.3700 | 0.3470 | 0.3540 |
| Dec | 0.4760 | 0.3520 | 0.4200 | 0.4030 | 0.3690 | 0.4020 |

| BSS-act-75 | LT=1 | LT=2 | LT=3 | LT=4 | LT=5 | LT=6 |
|---|---|---|---|---|---|---|
| Jan | 0.0196 | -0.0737 | 0.1479 | 0.0388 | 0.0311 | 0.1152 |
| Feb | -0.1136 | -0.0742 | -0.0242 | 0.0227 | 0.0485 | 0.0735 |
| Mar | 0.0037 | -0.3664 | -0.1976 | -0.2369 | -0.2381 | -0.1932 |
| Apr | 0.1892 | -0.2348 | -0.4098 | -0.2227 | -0.3292 | -0.3005 |
| May | -0.0525 | -0.2277 | 0.1018 | -0.1302 | 0.0536 | 0.0922 |
| Jun | -0.0211 | -0.0304 | -0.2867 | -0.0525 | 0.0284 | 0.0458 |
| Jul | -0.1284 | 0.0127 | 0.0073 | -0.2181 | 0.0763 | 0.1870 |
| Aug | -0.0717 | -0.0219 | 0.0368 | -0.0525 | -0.2065 | 0.2165 |
| Sep | 0.0079 | 0.0103 | -0.0102 | -0.0204 | 0.0311 | -0.0074 |
| Oct | -0.0383 | -0.0182 | -0.0181 | -0.0207 | -0.0052 | 0.0775 |
| Nov | 0.0076 | -0.0278 | -0.1226 | -0.0364 | 0.0280 | 0.0056 |
| Dec | -0.3077 | 0.0613 | -0.0633 | -0.0203 | 0.0658 | -0.0177 |

| BS-A/T-25 | LT=1 | LT=2 | LT=3 | LT=4 | LT=5 | LT=6 |
|---|---|---|---|---|---|---|
| Jan | 42.37 | 97.47 | 121.46 | 111.11 | 83.04 | 126.54 |
| Feb | 31.79 | 37.70 | 35.84 | 39.86 | 45.01 | 47.64 |
| Mar | 10.63 | 25.12 | 39.44 | 38.19 | 37.18 | 46.15 |
| Apr | 17.86 | 24.81 | 30.68 | 49.18 | 45.49 | 47.10 |
| May | 0.00 | 13.62 | 31.38 | 30.35 | 47.47 | 39.27 |
| Jun | 11.10 | 4.62 | 50.71 | 73.09 | 89.34 | 135.08 |
| Jul | 0.00 | 1.93 | 4.80 | 34.71 | 63.83 | 88.48 |
| Aug | 1.16 | 6.06 | 7.70 | 9.44 | 25.49 | 72.16 |
| Sep | 14.45 | 6.03 | 4.43 | 9.32 | 9.06 | 35.57 |
| Oct | 57.28 | 75.23 | 88.54 | 93.58 | 103.73 | 84.50 |
| Nov | 38.70 | 81.15 | 151.63 | 117.51 | 100.00 | 96.34 |
| Dec | 67.70 | 87.98 | 150.32 | 129.41 | 105.83 | 76.11 |

| BS-A/T-75 | LT=1 | LT=2 | LT=3 | LT=4 | LT=5 | LT=6 |
|---|---|---|---|---|---|---|
| Jan | 31.49 | 61.49 | 70.49 | 68.30 | 57.14 | 73.33 |
| Feb | 16.33 | 26.73 | 46.02 | 42.92 | 46.33 | 43.63 |
| Mar | 36.68 | 38.80 | 43.89 | 42.21 | 60.00 | 57.89 |
| Apr | 19.19 | 42.25 | 62.98 | 79.38 | 71.47 | 70.08 |
| May | 4.90 | 21.32 | 28.57 | 40.43 | 48.58 | 52.86 |
| Jun | 0.00 | 9.45 | 30.76 | 34.22 | 59.15 | 55.81 |
| Jul | 9.44 | 17.74 | 14.46 | 31.00 | 37.89 | 73.67 |
| Aug | 13.54 | 6.04 | 3.99 | 10.27 | 19.37 | 30.28 |
| Sep | 11.06 | 8.59 | 7.58 | 8.64 | 11.16 | 23.78 |
| Oct | 28.50 | 40.92 | 43.15 | 41.77 | 41.03 | 37.25 |
| Nov | 25.09 | 40.27 | 51.12 | 59.73 | 55.33 | 61.58 |
| Dec | 32.56 | 34.66 | 50.95 | 51.86 | 68.83 | 50.25 |

**Table 20: Skill scores in the Zambezi**





| BS-theo-25 | LT=1 | LT=2 | LT=3 | LT=4 | LT=5 | LT=6 |
|---|---|---|---|---|---|---|
| Jan | 0.0000 | 0.0000 | 0.0000 | 0.0901 | 0.1030 | 0.1450 |
| Feb | 0.0000 | 0.0000 | 0.0000 | 0.0048 | 0.0901 | 0.0949 |
| Mar | 0.0000 | 0.0000 | 0.0000 | 0.0011 | 0.0308 | 0.0955 |
| Apr | 0.0000 | 0.0000 | 0.0000 | 0.0011 | 0.0031 | 0.0482 |
| May | 0.0000 | 0.0000 | 0.0006 | 0.0074 | 0.0033 | 0.0174 |
| Jun | 0.0044 | 0.0234 | 0.0298 | 0.0366 | 0.0467 | 0.0521 |
| Jul | 0.0667 | 0.0320 | 0.0174 | 0.0488 | 0.1350 | 0.1740 |
| Aug | 0.0813 | 0.0537 | 0.0380 | 0.1020 | 0.1220 | 0.1560 |
| Sep | 0.1220 | 0.0551 | 0.0521 | 0.0837 | 0.1180 | 0.1460 |
| Oct | 0.1000 | 0.1000 | 0.0380 | 0.0931 | 0.1420 | 0.1620 |
| Nov | 0.1020 | 0.1000 | 0.0766 | 0.1020 | 0.1410 | 0.2160 |
| Dec | 0.0832 | 0.0482 | 0.0678 | 0.0873 | 0.1500 | 0.1960 |

| BSS-theo-25 | LT=1 | LT=2 | LT=3 | LT=4 | LT=5 | LT=6 |
|---|---|---|---|---|---|---|
| Jan | 1.0000 | 1.0000 | 1.0000 | -0.3391 | -0.1377 | -0.3049 |
| Feb | 1.0000 | 1.0000 | 1.0000 | 0.2130 | -0.2881 | -0.1053 |
| Mar | 1.0000 | 1.0000 | 0.9511 | | -0.9410 | -0.3272 |
| Apr | NA | 1.0000 | 1.0000 | 0.9207 | 0.8872 | -2.0854 |
| May | 1.0000 | 1.0000 | -0.5388 | -3.3776 | 0.5156 | 0.3832 |
| Jun | 0.8677 | 0.2980 | 0.2682 | 0.3394 | 0.1337 | 0.1427 |
| Jul | -3.1988 | -0.8905 | 0.5768 | 0.1501 | -0.6037 | -1.0255 |
| Aug | -1.4390 | -2.2032 | 0.1442 | -0.1784 | 0.0834 | -0.0944 |
| Sep | -2.6600 | -3.8921 | 0.0807 | -0.3049 | -0.3713 | -0.0090 |
| Oct | -3.5579 | -13.0917 | -0.3418 | -0.6362 | -0.9992 | -0.5417 |
| Nov | NA | -2.6099 | -6.6900 | -1.6827 | -1.0966 | -1.6279 |
| Dec | NA | -1215.08 | -0.2513 | -0.6231 | -0.8039 | -0.9152 |

| BS-theo-75 | LT=1 | LT=2 | LT=3 | LT=4 | LT=5 | LT=6 |
|---|---|---|---|---|---|---|
| Jan | 0.1000 | 0.1174 | 0.0863 | 0.0567 | 0.0459 | 0.1188 |
| Feb | 0.0690 | 0.0667 | 0.0690 | 0.0265 | 0.0388 | 0.0712 |
| Mar | 0.0344 | 0.0348 | 0.0433 | 0.0772 | 0.0048 | 0.0427 |
| Apr | 0.0333 | 0.0333 | 0.0348 | 0.0190 | 0.0792 | 0.0633 |
| May | 0.0336 | 0.0099 | 0.0080 | 0.0046 | 0.0094 | 0.0026 |
| Jun | 0.0711 | 0.0609 | 0.0069 | 0.0105 | 0.0137 | 0.0364 |
| Jul | 0.0802 | 0.0868 | 0.0769 | 0.0545 | 0.0606 | 0.0368 |
| Aug | 0.1333 | 0.1231 | 0.1041 | 0.1091 | 0.1212 | 0.1262 |
| Sep | 0.1333 | 0.1160 | 0.1273 | 0.1022 | 0.1110 | 0.1350 |
| Oct | 0.1667 | 0.1278 | 0.0848 | 0.1011 | 0.1025 | 0.1182 |
| Nov | 0.1333 | 0.1275 | 0.0854 | 0.0656 | 0.1022 | 0.1331 |
| Dec | 0.1333 | 0.1275 | 0.0736 | 0.0347 | 0.0890 | 0.1212 |

| BSS-theo-75 | LT=1 | LT=2 | LT=3 | LT=4 | LT=5 | LT=6 |
|---|---|---|---|---|---|---|
| Jan | NA | NA | -3.6285 | -0.7118 | -0.2573 | -0.6999 |
| Feb | -2.3898 | -13.1324 | -21.7297 | 0.1427 | 0.2925 | -0.2420 |
| Mar | -0.0290 | -0.0067 | -0.7081 | -29.3800 | 0.8210 | 0.2161 |
| Apr | -0.2176 | -0.1265 | -0.4571 | -0.8703 | -0.5747 | 0.0557 |
| May | -2.5725 | -0.3540 | -4.0086 | 0.4528 | 0.0145 | 0.9676 |
| Jun | -1.1322 | -49.5481 | 0.0160 | -3.5941 | -2.3088 | -0.6909 |
| Jul | -0.7294 | -0.8000 | -1.9888 | -0.2107 | -1.7140 | -0.6731 |
| Aug | -2.0096 | -1.2331 | -0.4049 | -0.5573 | -0.6918 | -1.3113 |
| Sep | -39.9600 | -3.7315 | -1.0964 | -0.3049 | -0.4216 | -0.8554 |
| Oct | NA | -244.4215 | -1.0316 | -0.0233 | 0.0896 | 0.0070 |
| Nov | NA | -978.5703 | -86.4490 | -0.4134 | 0.0517 | -0.0779 |
| Dec | NA | -10.1351 | -1.6740 | -1.6946 | -0.4209 | -0.0384 |

| BS-act-25 | LT=1 | LT=2 | LT=3 | LT=4 | LT=5 | LT=6 |
|---|---|---|---|---|---|---|
| Jan | 0.7330 | 0.7240 | 0.7240 | 0.6760 | 0.7200 | 0.8010 |
| Feb | 0.7590 | 0.7330 | 0.7240 | 0.7290 | 0.7010 | 0.7560 |
| Mar | 0.7330 | 0.7590 | 0.7330 | 0.7250 | 0.7490 | 0.7260 |
| Apr | 0.7330 | 0.7330 | 0.7590 | 0.7340 | 0.7270 | 0.7530 |
| May | 0.7330 | 0.7330 | 0.7340 | 0.7660 | 0.7370 | 0.7420 |
| Jun | 0.7380 | 0.7450 | 0.7020 | 0.7150 | 0.7300 | 0.7190 |
| Jul | 0.6670 | 0.7170 | 0.6900 | 0.7210 | 0.7230 | 0.7450 |
| Aug | 0.6270 | 0.6360 | 0.7170 | 0.7440 | 0.7340 | 0.7070 |
| Sep | 0.6010 | 0.6430 | 0.6880 | 0.7320 | 0.7540 | 0.7520 |
| Oct | 0.6330 | 0.6330 | 0.6870 | 0.7170 | 0.7600 | 0.8110 |
| Nov | 0.6180 | 0.6330 | 0.6280 | 0.7630 | 0.7840 | 0.8530 |
| Dec | 0.6350 | 0.6540 | 0.6680 | 0.6990 | 0.8050 | 0.8440 |

| BSS-act-25 | LT=1 | LT=2 | LT=3 | LT=4 | LT=5 | LT=6 |
|---|---|---|---|---|---|---|
| Jan | 0.0443 | 0.0372 | 0.0082 | -0.0196 | -0.0496 | -0.1728 |
| Feb | -0.0066 | 0.0214 | 0.0109 | 0.0014 | -0.0463 | -0.0941 |
| Mar | 0.0108 | -0.0341 | 0.0054 | 0.0295 | -0.0122 | -0.0917 |
| Apr | 0.0000 | 0.0014 | -0.0341 | 0.0174 | 0.0332 | -0.0217 |
| May | 0.0214 | 0.0027 | 0.0000 | -0.0422 | 0.0041 | 0.0133 |
| Jun | 0.0378 | 0.0287 | 0.0475 | 0.0124 | -0.0069 | 0.0110 |
| Jul | 0.1095 | 0.0311 | 0.0129 | -0.0315 | -0.0478 | -0.0892 |
| Aug | 0.1825 | 0.1080 | 0.0028 | -0.1171 | -0.1362 | -0.1222 |
| Sep | 0.2164 | 0.1168 | -0.0029 | -0.0796 | -0.1930 | -0.2129 |
| Oct | 0.0996 | 0.1281 | 0.0432 | -0.0513 | -0.1293 | -0.2997 |
| Nov | 0.1569 | 0.0691 | 0.1229 | -0.0671 | -0.1754 | -0.2808 |
| Dec | 0.1337 | 0.1078 | 0.0234 | 0.0237 | -0.1370 | -0.3005 |

| BS-act-75 | LT=1 | LT=2 | LT=3 | LT=4 | LT=5 | LT=6 |
|---|---|---|---|---|---|---|
| Jan | 0.8330 | 0.8420 | 0.8100 | 0.7240 | 0.6200 | 0.4790 |
| Feb | 0.7930 | 0.7930 | 0.7930 | 0.7380 | 0.7000 | 0.5700 |
| Mar | 0.7680 | 0.7590 | 0.7770 | 0.7570 | 0.7040 | 0.6670 |
| Apr | 0.7670 | 0.7670 | 0.7530 | 0.7400 | 0.7340 | 0.6870 |
| May | 0.7670 | 0.7430 | 0.7410 | 0.7290 | 0.7250 | 0.7270 |
| Jun | 0.8040 | 0.7940 | 0.7160 | 0.7130 | 0.6560 | 0.6360 |
| Jul | 0.8130 | 0.8200 | 0.7800 | 0.6670 | 0.5760 | 0.5670 |
| Aug | 0.8670 | 0.8500 | 0.7650 | 0.6300 | 0.5390 | 0.4350 |
| Sep | 0.8670 | 0.8370 | 0.7450 | 0.6170 | 0.5230 | 0.4380 |
| Oct | 0.9000 | 0.8610 | 0.7580 | 0.6220 | 0.4960 | 0.4520 |
| Nov | 0.8670 | 0.8610 | 0.8070 | 0.6660 | 0.5390 | 0.4360 |
| Dec | 0.8670 | 0.8610 | 0.7950 | 0.6830 | 0.5560 | 0.4480 |

| BSS-act-75 | LT=1 | LT=2 | LT=3 | LT=4 | LT=5 | LT=6 |
|---|---|---|---|---|---|---|
| Jan | -0.1364 | -0.1630 | -0.1571 | -0.0343 | 0.1610 | 0.3329 |
| Feb | -0.0517 | -0.0840 | -0.0938 | -0.0438 | -0.0448 | 0.1972 |
| Mar | -0.0013 | 0.0117 | -0.0237 | -0.0485 | -0.0100 | -0.0214 |
| Apr | -0.0079 | -0.0027 | 0.0000 | -0.0237 | -0.0237 | -0.0148 |
| May | -0.0323 | -0.0027 | -0.0082 | 0.0175 | 0.0014 | 0.0041 |
| Jun | -0.0482 | 0.0219 | -0.0803 | 0.0219 | 0.0952 | 0.1323 |
| Jul | -0.0423 | -0.0581 | -0.1337 | 0.0104 | 0.1616 | 0.1818 |
| Aug | -0.1144 | -0.1518 | -0.0566 | 0.0172 | 0.1552 | 0.3182 |
| Sep | -0.1764 | -0.1235 | -0.0735 | 0.0736 | 0.1565 | 0.2901 |
| Oct | -0.2278 | -0.1730 | -0.0369 | 0.0504 | 0.2114 | 0.2441 |
| Nov | -0.1828 | -0.1811 | -0.1055 | 0.0659 | 0.1512 | 0.2864 |
| Dec | -0.1828 | -0.2195 | -0.1088 | 0.0707 | 0.2202 | 0.2855 |

| BS-A/T-25 | LT=1 | LT=2 | LT=3 | LT=4 | LT=5 | LT=6 |
|---|---|---|---|---|---|---|
| Jan | 0.00 | 0.00 | 0.00 | 13.33 | 14.31 | 18.10 |
| Feb | 0.00 | 0.00 | 0.00 | 0.66 | 12.85 | 12.55 |
| Mar | 0.00 | 0.00 | 0.00 | 0.16 | 4.11 | 13.15 |
| Apr | 0.00 | 0.00 | 0.00 | 0.15 | 0.43 | 6.40 |
| May | 0.00 | 0.00 | 0.08 | 0.97 | 0.45 | 2.35 |
| Jun | 0.60 | 3.14 | 4.25 | 5.12 | 6.40 | 7.25 |
| Jul | 10.00 | 4.46 | 2.52 | 6.77 | 18.67 | 23.36 |
| Aug | 12.97 | 8.44 | 5.30 | 13.71 | 16.62 | 22.07 |
| Sep | 20.30 | 8.57 | 7.57 | 11.43 | 15.65 | 19.41 |
| Oct | 15.80 | 15.80 | 5.53 | 12.98 | 18.68 | 19.98 |
| Nov | 16.50 | 15.80 | 12.20 | 13.37 | 17.98 | 25.32 |
| Dec | 13.10 | 7.37 | 10.15 | 12.49 | 18.63 | 23.22 |

| BS-A/T-75 | LT=1 | LT=2 | LT=3 | LT=4 | LT=5 | LT=6 |
|---|---|---|---|---|---|---|
| Jan | 12.00 | 13.94 | 10.66 | 7.83 | 7.40 | 24.81 |
| Feb | 8.70 | 8.33 | 8.70 | 3.59 | 5.54 | 12.50 |
| Mar | 4.48 | 4.58 | 5.57 | 10.20 | 0.69 | 6.41 |
| Apr | 4.35 | 4.35 | 4.62 | 2.57 | 10.79 | 9.21 |
| May | 4.38 | 1.33 | 1.08 | 0.63 | 1.29 | 0.35 |
| Jun | 8.84 | 7.67 | 0.96 | 1.47 | 2.09 | 5.72 |
| Jul | 9.86 | 9.85 | 8.18 | 10.52 | 6.48 | |
| Aug | 15.38 | 14.49 | 13.61 | 17.32 | 22.49 | 29.00 |
| Sep | 15.38 | 17.08 | 13.86 | 16.56 | 21.23 | 30.82 |
| Oct | 18.52 | 14.85 | 11.19 | 16.25 | 20.66 | 26.15 |
| Nov | 15.38 | 14.81 | 10.58 | 9.84 | 18.96 | 30.52 |
| Dec | 15.38 | 14.81 | 9.25 | 5.08 | 16.00 | 27.06 |

**Table 21: Skill scores in the Nile**





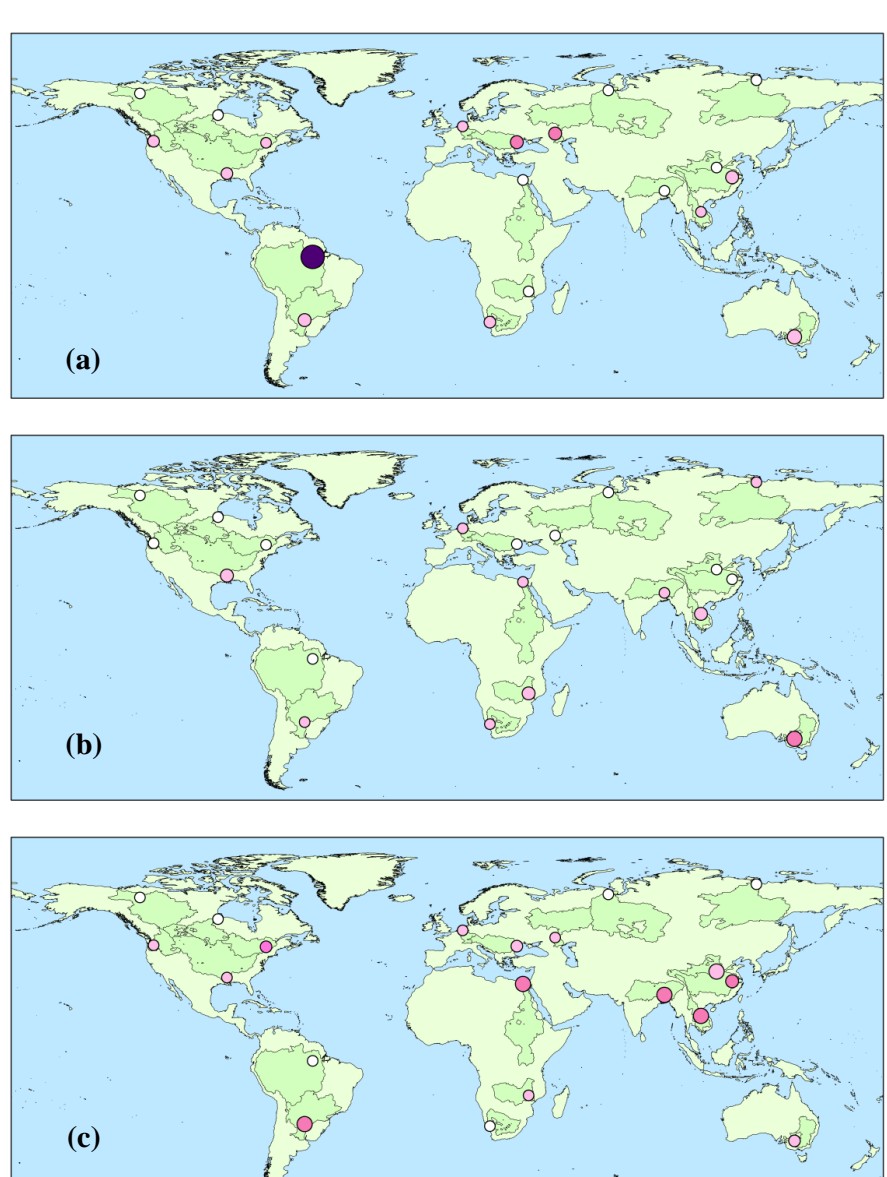





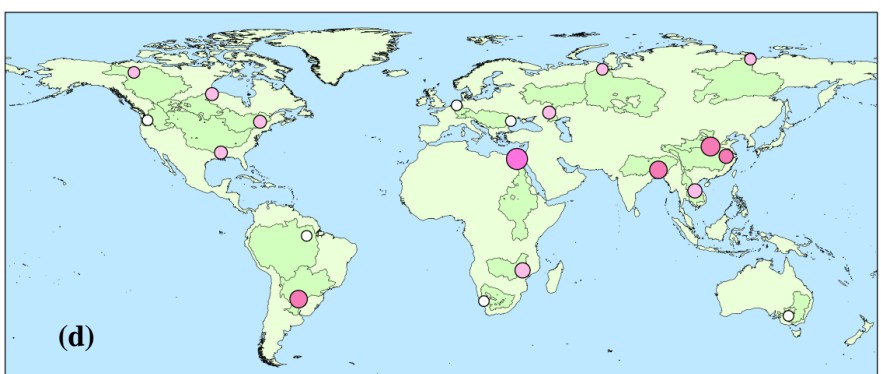

Number of skilful forecasts (months per year)

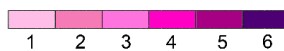

Maximum lead time (month)

**Figure 1: Global overview of basins with improved forecast skill**

  (a)  **theoretical skill in low flows**

  (b)  **theoretical skill in high flows**

  (c)  **actual skill in low flows**

  (d)  **actual skill in high flows**