# Peer review of "Skill of a global forecasting system in seasonal ensemble streamflow prediction"

_Hydrology and Earth System Sciences, 2016_

## Referee Comment (RC1) · Anonymous Referee #1 · 16 Jan 2017

Review of "Skill of a global forecasting system in seasonal ensemble streamflow prediction" by Candogan Yossef et al.

General Comments:

This paper uses a global hydrological model to produce seasonal forecasts for 20 large rivers across the globe, using both the ECMWF System3 forecasts as forcing, and the ESP technique. The 'theoretical' and 'actual' skill of the two approaches is assessed for high and low flows using the Brier Score.

I read this paper with great interest, and liked the comparison of the S3 forecasts with the ESP. The paper is very well written throughout, the methodology is sound and I

believe the results and conclusions are of interest to the hydrometeorology community and relevant for publication in HESS.

My only concern with this paper is the authors' decision to use ECMWF's System 3 forecasts, rather than the current System 4 which has been in operation since 2011. This is not currently justified in the manuscript - as such it may seem that the results could become outdated due to the existence of newer, potentially more skilful seasonal meteorologial forecast.

Specific Comments & Corrections:

Overall, the text is very well written, but I have a few comments:

Page 2 Line 13: The authors state that medium-range NWP is based on atmosphere-only integrations; however, many NWP systems are now coupled atmosphere-ocean models. For example, ECMWF's medium-range ensemble forecasts are coupled, and the authors indeed later state that S3 is a coupled model (which not medium-range, this is still a NWP system). The explanation of NWP vs GCM should be amended as such and clarified, as this is not the key difference between the two.

Page 2 Line 31: While I have no issue with the number of rivers or their distribution, I don't believe the authors provide an explanation of why they chose to conduct the study in this way, and why 20 rivers were chosen. Is this a good representation of global rivers? Some further explanation with regards to this choice would be helpful.

Page 5 Line 13: I would ask the authors to please explain the reason for choosing to use ECMWF System 3 forecasts rather than the current version, System 4, which has been in operation since November 2011.

Page 7, BS: Having read the full paper, my understanding that the authors have calculated the BS for both the ESP and ECMWF S3 forecasts, however this is not clear from the explanation provided here, please could the authors clarify this.

Page 6/7: For the skill assessment, it is not clear exactly what data are used and

there is very little information provided about the actual discharge observations. Is the forecast discharge used the basin-average discharge? Or the discharge at the basin outlet? Where exactly are the GRDC stations located, at the basin outlet also? Is just one GRDC station used per river? How long are the GRDC observation records and does this affect the skill assessment if they are not of the same length as the control, or contain missing data? More information on this is required, and in fact in general there is little discussion of the limitations of the datasets used.

Page 8, Line 1: With regards to the tables, 20 8-panel tables is rather too much to include in the main manuscript. I would recommend choosing one, or part of one, table to use as an example in the main text, and move the remaining tables to an appendix or supplementary material to which the reader can refer.

Page 8, Line 5: I presume here that the BS presented are those for the ECMWF S3 forecast, not the ESP, but I am unsure of this - please clarify when introducing the tables, which forecast the BS is shown for, and that the BSS compares this to the other forecast to indicate the improvement.

Figures: I would recommend adding an additional map showing the basin names, to avoid the reader needing to look up those they are not familiar with.

On the global maps (figure 1), it is not clear what a white circle indicates - I presume that these are the basins which do not see an improved forecast skill for that scenario - I would recommend either not showing a circle at all for those which do not see improved skill, or clarifying this in the legend.

---

## Referee Comment (RC2) · Anonymous Referee #2 · 18 Jan 2017

**Review of the paper entitled "Skill of a global forecasting system in seasonal ensemble streamflow prediction" by Candogan Yossef et al., submitted to HESS in October 2016**

This manuscript evaluates the benefits of using ECMWF S3 bias-corrected seasonal ensemble forecasts for predicting high and low monthly flows compared with the Ensemble Streamflow Prediction (ESP) flow ensembles for 20 large river basins in all continents. The evaluation of the quality of ensemble forecasts is based on the Brier Score (BS) and the Brier Skill Score (BSS) for 2 binary events defined by the 75% and 25% probability thresholds, for forecast lead times up to 6 months. The relative contribution of the meteorological forcing uncertainty and the total contribution of both the meteorological and hydrological uncertainties are evaluated by verifying with simulated flow and observed flow. This is an interesting research topic area since probabilistic seasonal forecasts could potentially support various critical applications of global hydrometeorological ensemble prediction systems.

The paper is clearly organized and generally well written, although the terminology regarding forecast verification should be improved. It includes appropriate references and substantial evaluation results. However the process to define the benchmark forecasts in this study and the verification scores being used need to be clarified. The presentation of the evaluation results in the different figures and tables needs to be more synthetic and more easily understandable.

I recommend this paper to be published after the authors have addressed the following general comments and specific comments to help improve the quality of the manuscript.

**General comments**

The verification terminology used in the paper should be improved. When referring to forecast *skill*, in most part of the paper, the authors mean forecast *quality*, one aspect of the forecast quality being the forecast skill when using a given verification metric and a specific reference forecast. In this paper, the authors should refer to the evaluation of the forecast *accuracy* with one metric, the Brier Score (which does include different aspects of the forecast quality attributes, see the decomposition of BS for example). Then the authors used its associated skill score, the BSS, the ESP flow forecasts being the reference forecast, to evaluate the gain *in terms of the Brier Score* by integrating the seasonal forecasts.

Also the terminology used by the authors regarding the "theoretical skill" vs. the "actual skill" of the flow forecasts is not widely used in the literature and may be misleading (again, the *skill* term should only be used when a benchmark forecast is defined). It needs to be clarified in the paper (and in the interpretation of the results) that, when comparing forecast flows with simulated flows, the hydrological errors are cancelled out, which leads to assess the contribution of the forcing uncertainty only to the forecast flows; the verification with the observed flows leads to assess the total contribution of the forcing uncertainty and the hydrologic uncertainty. It may be more appropriate to refer to MF uncertainty vs. total uncertainty in the BS notations and interpretation of results. The equation 2 on page 7 should be corrected ($BS_{theo}*100/BS_{act}$) and the obtained ratio

results should be discussed in terms of the relative contribution of these 2 sources of uncertainty.

Besides, the reforecasting process and the benchmark forecasts used in this study need to be clarified. As the authors mentioned, one has to first run the hydrological model with the observed forcing values during a spin up time period to define appropriate initial states to then start reforecasting. This spin up period should be excluded from the analysis of the reforecast dataset. In the paper, it is not clear whether the spin up period covers the period of 1979-1984 (p. 5) or 1979-1980 (p. 6) and has been excluded from the forecast dataset for the verification analysis. The authors then integrated retroactively the ECMWF S3 seasonal forecasts and the historical observed forcing values to produce, respectively, the S3-based flow forecasts and the ESP flow forecasts from 1981 to 2010. In the ESP flow ensembles, based on observed forcing values from all years in 1981-2010, one member corresponds to the simulated flow (e.g., the run with forcings from 1981 initiated in 1981). This member should not be part of the ESP flow ensembles since, in real-time forecasting, all ESP ensembles use past historical years of forcing as possible future outcomes (considering that the climate is stationary and repeated itself). Including the simulated flow in the ESP ensembles will lead to artificially increase the accuracy of these forecasts in the BS values, which will then decrease the skill of the S3-based flow forecasts in the BSS values. It seems that the authors did include the simulated flow (or control run) in the ESP members, which may explain why there is almost no gain in using the seasonal forecasts. If they didn't include that member in the ESP ensembles, this should be clarified in the paper.

The evaluation results are presented for 20 basins, using too many tables (10 tables for each basin). The BS values do not seem to be essential for the evaluation study since the main point of the authors concerns the benefits of seasonal forecasts compared to ESP. Having a common benchmark with the BSS score for all 20 basins makes it easier to compare the results among the different river basins. The authors could include only the BSS results (even if the BS values are mentioned in the text for specific aspects), as well as the ratio describing the relative contribution of the 2 sources of uncertainty. Also, all the tables could be turned into grid figures, using a color scale with more color categories than the ones currently used in the tables, to facilitate the interpretation of the results. To further reduce the number of figures, some of them could be included in appendices if the results for basins from similar climatic zones are the same.

The authors need to include more information about the selection of the 20 test basins, with a table describing the basins in terms of basin size, average flow, and a corresponding map of the rivers and outlet locations with names (see the material included in Candogan Yossef et al. 2012). Since the authors referred quite extensively to 2 past studies with the same hydrologic model and the analysis of the forcing uncertainty and initial conditions uncertainty (Candogan Yossef et al. 2012 and 2013), it would be necessary to mention whether the same 20 basins were used in all these studies. The impact of flow regulations needs also to be discussed here since this impact could dominate the hydrologic uncertainty as well as the forcing uncertainty (cf. discussion in the 2012 paper).

**Specific comments**

- Abstract, p. 1 line 12: please specify that the PCR-GLOBWB hydrologic model is distributed (maybe better to spell out the acronym of the model name). It would be better to mention "ensemble reforecasts" (as the forecasts are produced retroactively) and indicate the reforecast period.

- Abstract, p. 1 line 15: please remove "the skill from" (see general comment about the use of the term "skill").

- Abstract, p.1 line 18: please change "skill" to "forecast accuracy".

- Abstract, p. 1 line 19-22: consider clarifying that the analysis concerns the relative contribution of the forcing uncertainty and the hydrologic uncertainty to flow predictions when verifying with both simulated flow and observed flow (see general comment).

- Page 3, lines 2-6: please consider including more specifics about the 2012 study (period of evaluation, test basins, simulated and/or observed flows used for evaluation, possible differences with the presented work) since the authors referred to the study results quite extensively in the paper (see general comment).

- Page 3, line 18: consider adding a short description of the ESP and reverse ESP approach since the conclusions about the relative importance of the forcing uncertainty and the initial conditions uncertainty is one of the main points of discussion in this paper and results from the 2013 study are mentioned quite extensively in the results section.

- Page 4, line 27: clarify what DDM30 is (source of dataset? Please spell out the acronym).

- Section 2.2, page 5: please specify the spatial and temporal resolution of the forcings for both the observation/reanalysis and reforecast datasets (spell out the acronyms only if necessary; WCRP is not needed); is there any change of spatial resolution from the original dataset to the forcing inputs for the reforecast runs?

- Page 5, line 14: please explain why the authors used the ECMWF S3 seasonal forecasts when the S4 forecasts are operational since November 2011 (and being evaluated in the GLOWASIS project); how the S3 and S4 seasonal forecast datasets compare to each other (number of members; is the same bias correction procedure also included in the S4 forecasts?); see also the comment for the conclusion section.

- Section 2.3, page 5, starting at line 35: please clarify the reforecast process, which starts with the spin up run of the hydrologic model and needs to exclude the spin up period (1979-1980 to start reforecasting in 1981?) from the reforecasting and verification period (see general comment).

- Page 6, lines 5-9: please clarify the ESP members being used, excluding the member corresponding to the control run (see general comment).

- Section 2.4, page 6: please consider adding a reference to the forecast accuracy when using the Brier Score, and the forecast skill when using the BSS (see general comment).

- Page 6, line 21: please specify that GRDC is the source of the flow observations (spell out the acronym).

- Page 6, lines 24-29: please include that the BS is the mean squared error of probabilistic forecasts for a given dichotomous event; a probability threshold is used to define the binary event to be observed and forecasted. The authors should point out that the BS is a relevant metric for analyzing the performance of a forecast system for specific categories of flow (in this case, with a high flow threshold and a low flow threshold). The authors should clarify why they selected the 75% and 25% flow thresholds (user requirements? large enough sample sizes?)

- Page 6, lines 30-35: please clarify how the threshold values for the BS computation are defined using the simulated flow values vs. forecast flow values (using which forcing forecasts?).

- Page 7, lines 19-28: please refer to the relative contribution of the forcing uncertainty and the hydrologic uncertainty to flow predictions when verifying with both simulated flow and observed flow (see general comment). Please correct equation 2 and comment on the variations of the ratio (what if the denominator has a value close to 0?).

- Section 3.1, page 8: please see the general comment about including only figures with the BSS results (not the BS values) with a smaller number of color-coded figures.

- Page 10, lines 14-15: clarify what the authors mean by "the skill of the ESP is below the climatology" (referring to the unconditional climatological record of observed flow?).

- Section 5, Conclusion, page 13: please comment on the potential use of the S4 seasonal forecasts in a similar study and the potential gain in forecast quality due to the seasonal system enhancements. The authors should also mention that the use of a single verification metric (the BS and its associated skill score) could be complement by other verification metrics, such as the ROC score (to characterize the event discrimination of the forecasts) and the BS decomposition (to evaluate in more details the conditional and unconditional biases in the forecasts). The sampling uncertainty of the verification metrics should also be evaluated (for example with a bootstrapping technique), especially if the verification analysis is also conducted with higher probability thresholds for the BS computation. Finally the authors could include some comments about the user requirements for seasonal probabilistic flow forecast systems (and the evaluation the system performance) and collaborations between forecasters and end users (for example in the GLOWASIS project) to further improve the usefulness of such systems.

- Figure 1, pages 38-39: please clarify what the white circles mean; it would be better to use more different colors in the color scale. Please refer to the BSS and the ESP reference forecasts in the legend (forecast skill being specific to a given verification metric and a benchmark).

---

## Referee Comment (RC3) · Anonymous Referee #3 · 19 Jan 2017

General comments

The objective is to investigate the total skill of seasonal streamflow forecasts for 20 of the largest rivers in the world, produced by a global hydrological model (PCR-GLOBWB) forced with the ECMWF S3 system. This manuscript got my attention because I think that ensemble streamflow prediction, as a product of coupling GCMs to Global Hydrological Models, is very promising. In addition, initiatives that can operationally provide seasonal forecasts and conditions about water shortages, occurrence of floods and energy production, in a global scale, are valuable especially for data scarce regions and countries where river basins extend beyond the political boundaries. In this context, I believe that the scope of the study is of interest of the scientific

community. In general, the methodology is sound and the manuscript is well written, providing an extension of previous studies about the skill of the PCR-GLOBWB to predict streamflows (Candogan Yossef et al. 2012, Candogan Yossef et al. 2013). Although I am not familiar enough with the state-of-the-art research regarding seasonal hydrologic forecasting, some questions arose after reading the manuscript and I was not fully convinced by the conclusions obtained in the study. Therefore, in an attempt to motivate further discussion, I have made some appointments that should be addressed/answered by the authors in order to the article be acceptable for publication.

Specific Comments

Firstly, I missed a better support of literature in the introduction, and authors should make an exercise of providing a suitable number of references. For instance, at page 2, lines 11-29, there is a long explanation about climate models, drivers of global climate patterns, atmospheric chaos and the use of ensembles for streamflow forecasts, supported by a single reference. You have mentioned that the capability of the global models to simulate streamflow was demonstrated in several cases, but what has been done in respect to [seasonal] streamflow forecasting? The study should be positioned among other existing research in this field, especially for continental (e.g. Zhao et al. 2016) and global scales (e.g. Yuan et al. 2015a). Did the experiments of coupling GCMs to hydrological models show skill on seasonal streamflow forecasting when compared to the ESP approach on those scales? It gives the reader an overview about the subject and allows to verify if the conclusions of your work agree to the existing studies or are pointing to the opposite direction. Maybe the article of Yuan et al. 2015b can be a good start to carry out this exercise.

Despite of using a specific workflow of the Delft-FEWS, I believe that the ESP generation procedure should be better clarified in text. In my understanding, the authors are producing a 32-member ensemble for each calendar month based on meteorological forcing data between years 1979-2010. This means that the ESP ensembles contain all the information of the observed meteorology (bias corrected ERA-Interim reanalysis with GPCP monthly rainfall observations) of the 'almost' same hindcasting period (1981-2010). In this sense, the comparison of ESP ensembles to ECMWF S3 seems to be unfair, because in a real situation we do not have perfect information about meteorology in the future. Did the authors leave the target year out when producing the ESP ensembles? The forecast years resampled from the historical record are required to be independent from the forecast target year, in order to avoid the inclusion of a perfect forecast. Maybe it is more convenient to generate ESP ensembles by selecting a random number of years from the historical period, i.e., not the entire period, excluding the target year (e.g. Mo et al. 2012; Yuan et al. 2016).

Another issue that I am concerned about is the definition of low and high flows. Thresholds corresponding to the 25th and 75th percentiles of the observed values and for both control and forecasting runs were respectively selected, meaning that streamflows are reaching values outside the "normal" conditions during 50% of the time. On the other hand, one of the advantages of climate models in respect to statistical methods is that the former is most suitable to predict extreme (rare) events, since the predictability of large-scale climate drivers such as the ENSO can be improved by assimilation of real time observed data in the numerical models. Especially for development countries, knowledge about the possibility of extreme droughts and floods are, indeed, much more important to predict than a simple detection of high or low flows. Would the results be the same if thresholds corresponding to 10th and 90th percentiles (still conservative for rare events, but enough for statistical significance) were selected?

Is there a specific reason for choosing the BSS to assess skill of ECMWF S3 over ESP method? Since BS is a metric for discrete events (two categories), the problem is that it does not account for the distance of ensemble members to the threshold used for low or high flows (BS values are too much dependent of those thresholds). Thus, why not including performance metrics such as the Continuous Ranked Probability Skill Score (CRPSS), which has been adopted in several recent studies involving seasonal streamflow forecasts (e.g. Arnal et al. 2015, Zao et al. 2016, Yuan et al. 2015)?

Particularly, I am very opposite to manuscripts that use an excessive number of tables to present results. The way you presented here sounds more like a report (10 tables x 20 rivers = 200 tables) than a scientific paper to me, which can discourage people to continue reading. I think the authors must demonstrate the ability to summarize the information produced and present the results in an appropriate manner; otherwise, it can be very difficult to draw conclusions. So, I am wondering: are all those information really necessary? If your objective is to assess the skill of the forecasting system, perhaps is better to focus in your Skill Score (BS can be moved to supplementary data). One possibility is to constrain your assessment to dry and wet seasons instead of all months. Conversely, if the authors really want to show results for each calendar month, I suggest to take a look at papers such as Schepen et al. (2016) and Yuan et al (2016), which are also handling results for many rivers. Finally, all those tables comprising results of BS and BSS for each month of the year can be moved to supplementary data.

Page 13, lines 34-36: One of the findings is that: "the apparent potential for improvement in seasonal hydrological forecasts by using better meteorological forecasts cannot be realized as yet with the model PCR-GLOBWB and the ECMWF S3 dataset". However, there is only a short description about the quality of meteorological forecasts, regarding some verifications conducted with the ECMWF S3 seasonal forecasting system (page 5, lines 16-24, without listing any references). I think that more information is needed to provide a clear understanding about the improvement of the ECMWF S3 over the ensembles of meteorological forcing used in the ESP. Thus, it would be better to verify the skill of ECMWF S3 (precipitation / temperature) forecasts prior to the hydrological assessments as it can be helpful to support your conclusions.

Technical issues

There is a misunderstanding about skill and Brier Score (BS) in the manuscript, so terms should be revised to avoid confusions. The Brier Score measures the magnitude of the probability forecast errors (authors are referring this to be skill), which is strongly

influenced by the climatological frequency of the event in the hindcasting sample. For instance, if thresholds are selected for rare events, a good performance for the Brier Score will be obtained no matter if the forecasting system is less or more conservative, so low values of BS not necessarily implies in existence of forecasting skill. Conversely, skill must reflect the relative accuracy of the forecast over a reference forecast. You can do this by comparing a score (like BS) of a given forecasting system relatively to the BS of an unskilled forecast (conditional persistence, climatological forecast) or to another forecasting system, as done when computing Brier Skill Score (BSS) for BS ECMWF S3 / BS ESP. Therefore, you cannot say that skill is obtained through a comparison of forecasts to observed values or a control simulation (because these variables are used to compute the Brier Score, not skill). Authors should make clear that "Skill assessment was conducted in terms of Brier Skill Score" and also make the necessary adaptations.

I guess the term "actualized" is not well suitable in: pg. 10, line 35 "the percentage of theoretical skill actualized". I would recommend changing to "ratio of actual to theoretical skill". (There are other occurrences in the manuscript that should be changed)

Some parts of the text are too much didactic and do not fit well in a scientific paper. For instance, page 8, line 30 "In Table 2 for the Amazon, the color-coded first part which presents the theoretical skill for low flow shows that most of the BS values are coloured blue. This indicates that the ECMWF S3 forecasts are significantly more skilful than the ESP forecasts, i.e., the difference between the BS values is higher than 0.05." However, after handling the excessive number of tables, I guess this technical comment will be addressed.

Page 12, lines 34-35: "is the lowest in this continent", I could not understand if you are referring to semi-arid regions or to Murray-Darling basin. Also, add "-Darling" after "Murray" in line 36.

In the abstract, authors should include more details about results and conclusions. The current form is too vague to be presented.

References

Arnal, Louise, Fredrik Wetterhall, and Florian Pappenberger. "Seasonal hydrological ensemble forecasts over Europe." EGU General Assembly Conference Abstracts. Vol. 17. 2015.

Candogan Yossef, N., van Beek, L. P. H., Kwadijk, J. C. J. and Bierkens, M. F. P.: Assessment of the potential forecasting skill of a global hydrological model in reproducing the occurrence of monthly flow extremes, Hydrol. Earth Syst. Sci., 16, 4233–4246, doi:10.5194/hess-16-4233-2012, 2012.

Candogan Yossef, N., Winsemius, H., van Beek, L. P. H., Weerts, A. and Bierkens, M. F. P.: Skill of a global seasonal streamflow forecasting system, relative roles of initial conditions and meteorological forcing, Water Resour. Res., 49/8 4687–4699, doi:10.1002/wrcr.20350, 2013.

Mo, K. C., Shukla, S., Lettenmaier, D. P., & Chen, L. C. Do Climate Forecast System (CFSv2) forecasts improve seasonal soil moisture prediction? Geophysical Research Letters, 39(23). 2012

Schepen, A., Zhao, T., Wang, Q. J., Zhou, S., and Feikema, P.: Optimising seasonal streamflow forecast lead time for operational decision making in Australia, Hydrol. Earth Syst. Sci., 20, 4117-4128, doi:10.5194/hess-20-4117-2016, 2016.

Yuan, X., Roundy, J. K., Wood, E. F., & Sheffield, J. Seasonal forecasting of global hydrologic extremes: system development and evaluation over GEWEX basins. Bulletin of the American Meteorological Society, 96(11), 1895-1912. 2015a

Yuan, X., Ma, F., Wang, L., Zheng, Z., Ma, Z., Ye, A., and Peng, S.: An experimental seasonal hydrological forecasting system over the Yellow River basin – Part 1: Understanding the role of initial hydrological conditions, Hydrol. Earth Syst. Sci., 20, 2437-2451, doi:10.5194/hess-20-2437-2016, 2016.

Yuan, X.; Wood, E.; Ma, Z. A review on climate-model-based seasonal hydrologic

forecasting: physical understanding and system development. WIREs Water 2015, 2:523–536. doi: 10.1002/wat2.1088, 2015b

Zao, T.; Schepen, S. and Wang, Q. J. Ensemble forecasting of sub-seasonal to seasonal streamflow by a Bayesian joint probability modelling approach. Journal of Hydrology, v.541, Part B, p.839-849, 2016.

---

## Author Comment (AC1) · 14 Feb 2017

Anonymous Referee #1:

We thank Referee #1 for her/his kind words and valuable comments.

With regards to the specific comments and corrections, we understand that we intermingle the terminology of GMC and NWP. We will clarify this accordingly and provide a consistent description of the meteorological forecasting system used.

The choice of the 20 major river basins is a continuation of earlier analysis (Candogan Yossef et al., 2012, 2013) and this selection covers a wide range of hydro-climatic conditions. We will provide a short explanation. In the original paper, also the location

of the GRDC stations is given. We will summarize this information here and include a brief description of the nature of the datasets used. We will clarify that we compare discharge values at the station location in order to allow us to determine both the theoretical and actual skill from in our analysis.

The referee rightly asks why we have used the ECWF S3 forecast and not the newer, potentially more skilful S4. The only reason is that S4 was not available when the project started. Still, one would expect the S3 temperature and precipitation to be sufficiently accurate. While the forecast skill might improve using S4, we believe that the current study using S3 provides important insights.

We also thank Referee #1 for her/his suggestions to improve the tables and figures in the manuscript. We will clarify in the text and in the table captions which BS is presented. In the main text, we will summarize the tabular information to its essentials and relegate the full tables to an appendix. We will clarify the legend of the global map (Figure 1) as suggested by the referee.

Anonymous Referee #2:

We thank Referee #2 for her/his kind words on the content and organization of our manuscript and the suggestions to improve its formulation. We understand that one of the major comments concerns the terminology used re verification. We value the suggestion to further extricate the BS and interpret it in terms of the forcing and hydrological uncertainty.

In our simulations we used the observed forcing over the period 1979-1984 to spin up the model and then ran the model starting from these initial states with the ECMWF S3 seasonal forecasts for the period 1979-2010. For each of these simulations, we excluded the first two years and limited the subsequent analysis to the period 1981-2010 in order to avoid any further bias. We did not include the 1981 member in the ESP ensembles as suspected by the reviewer and we will clarify this in the revised manuscript. We will clarify this better in the text.

We understand that the number of tables is too large for the main body of the text and we will change it accordingly. The suggestion to relegate the BS values to an appendix may be considered, and the same holds for the different basins. In the design of the figures we experimented with different lay-outs, colour-codings etc. to present the material concisely. We will re-evaluate this and try and consolidate the information to its essentials.

We will comprise a brief description of the 20 major river basins used and underline the continuity with our earlier work. We wished not to dwell too much on points raised in earlier studies and therefore focused on the skill assessment here. However, in light of the distinction between forcing and hydrological uncertainty suggested by Referee #2, aspects of flow regulation can be covered in the discussion. The specific comments raised by Referee #2 will help to improve the clarity of the paper and we will accommodate them within the revised manuscript.

Anonymous Referee #3:

We thank Referee #3 for her/his suggestions and comments that certainly will help us to improve the quality of our manuscript. We will address these comments here and hope to make a stronger case for the conclusions we had drawn originally.

We thank Referee #3 for the number of relevant and interesting articles and we will include these with other references in our introduction.

In the workflow of the ESP we indeed excluded the year under consideration out of the ensembles. It is unfortunate that this is not completely clear from our description and we will improve this in our revision. We acknowledge that the selected 25th and 75th percentiles of the streamflow are not very extreme. However, it should be noted that we consider monthly discharges and, therefore, using the upper and lower 10% over 32 years provide far too small number of events for the analysis.

The suggestion to use the Continuous Ranked Probability Skill Score (CRPSS) is interesting and a viable solution to the BSS. However, we deem that the differences between the thresholds used in the Brier Score (BS) are not too different, and, although the remark justified, the practical implications are not too grave.

As mentioned by the other referees, Referee #3 suggests to move the tables to the Supplementary Information. We will do this accordingly and review, in light of all comments, the possibility to update the figures and to present our findings as clear and concise as possible. We wish to stress that already in writing the current manuscript much time and effort were spent to present the material as briefly as possible without jeopardizing the transparency of our outcomes.

Concerning the verification of S3 dataset, we will include the references to the verification carried out by ECMWF. We can expect the quality of the seasonal forecast to be initially fairly well, conditioned by SSTs and will start to deteriorate later on, as seasonal forecasts start to show signs of systematic model errors after about ten days into the forecast. As mentioned in the manuscript, the ECMWF applied a daily bias-correction based on quantile-quantile transformation but did not introduce any artificial terms in the equations to reduce the drift. In order to account for drift, we applied a bias correction using 12 datasets varying per forecast month, provided by the ECMWF. Therefore, we expect the temperature and precipitation to be reasonably correct, and we believe further verification of S3 forecasts would be beyond the scope of our study.

In terms of technical comments, we will update the description of the BS throughout the manuscript. Similarly, we will remove the occurrence of "actualized" in the text and refer to the Murray-Darling as such throughout. Equally, we will update the abstract.

———————————————

---

## Author Response (AR1)

Dear Maria-Helena Ramos,

We would like to thank you for the opportunity to submit a revised version of our manuscript. We have carried out the revision according to your recommendations as well as those of the referees. The major changes to the manuscript include:

- a clarification of the differences between the systems S3 and S4, and a justification of the use of S3 in our study (Section 2.2)

- correction of terminology regarding skill (We have revised the whole text and limited the use of the term "skill" to the cases where a skill score and reference benchmark are concerned. In all other cases, we used the terms "forecast accuracy" or "performance".)

- moving the score tables to the supplementary materials

- an explanation of our reason for preferring the BS and BSS over other scores such as CRPS or ROC (Section 2.4)

- a clarification of the spin-up period and exclusion of the target year from the ESP flow ensembles (Section 2.3)

- presenting a map with the basin names and a table with basin characteristics and gauging stations

We believe the manuscript has improved significantly after these major and several other minor changes; and hope that it is now fit for publication in HESS.

looking forward to hearing from you, best regards,

Naze Candogan Yossef

[revised manuscript text omitted]

5      **Figure 1: Selected basins**

25

[Figure]

[Figure]

[Figure]

[Figure]

Number of skilful forecasts (months per year)

[Figure]

Maximum lead time (month)

[Figure]

**Figure 2: Global overview of basins with improved forecast skill**

       **(a)**   **theoretical skill in low flows**

       **(b)**   **theoretical skill in high flows**

       **(c)**   **actual skill in low flows**

       **(d)**   **actual skill in high flows**

---

## Author Response (AR2)

Dear Maria-Helena Ramos,

Thank you very much for your decision to accept our manuscript for publication in HESS, with minor corrections. Regarding your final remarks our response is as follows:

- Our past study (2012) does not include the exact same 20 rivers. While most of them are the same, there are a few different ones. In our later study (2013) we analyzed 78 basins, not all of which were discussed in detail. For the present paper, we selected 20 of those basins. The referee comment was actually very right. The paper was missing the selection criteria and basin data, and adding them definitely improved the paper. If the basins were the same as the ones for the previous studies, we should mention this, but since they are not, we think it's not really relevant to mention which basins are the same and which ones are different.

- We included a reference to Yuan et. al (2015) in the introduction and added it to our reference list. This study is indeed relevant to the subject of our paper, and we thank you for reminding this point.

- We added a line mentioning that the tables are provided as supplement material.

- We went through the discussion of results section, but we didn't really feel comfortable to remove some parts to the previous section without impairing the discussion. Also concerning the meteorological forcing uncertainty versus total uncertainty, in our discussion we make several references to our 2013 study, which is basically an investigation of the role of meteorological forcing vs. initial conditions. So that is an aspect that was covered there, and actually the present assessment is based on the results of that study.

We hope our explanations will be satisfactory. We cordially thank you for all your help to improve our manuscript.

kind regards,

Naze Candogan Yossef

[revised manuscript text omitted]

5 **Figure 1: Selected basins**

25

[Figure]

[Figure]

[Figure]

[Figure]

Number of skilful forecasts (months per year)

[Figure]

1 2 3 4 5 6 7 8 9 10 11 12

Maximum lead time (month)

[Figure]

0    1    2    3    4    5    6

**Figure 2: Global overview of basins with improved forecast skill**

5       **(a)   theoretical skill in low flows**

        **(b)   theoretical skill in high flows**

        **(c)   actual skill in low flows**

        **(d)   actual skill in high flows**